# SDEs for Adaptive Methods: The Role of Noise

## Abstract

Despite the vast empirical evidence supporting the efficacy of adaptive optimization methods in deep learning, their theoretical understanding is far from complete. In this work, we introduce novel SDEs for commonly used adaptive optimizers: SignSGD, RMSprop(W), and Adam(W). Our SDEs offer a quantitatively accurate description of these optimizers and help bring to light an intricate relationship between adaptivity, gradient noise, and curvature. Our novel analysis of SignSGD highlights a noteworthy and precise contrast to SGD in terms of convergence speed, stationary distribution, and robustness to heavy-tail noise. We extend this analysis to AdamW and RMSpropW, for which we observe that the role of noise is much more complex. Crucially, we support our theoretical analysis with experimental evidence by verifying our insights: this includes numerically integrating our SDEs using Euler-Maruyama discretization on various neural network architectures such as MLPs, CNNs, ResNets, and Transformers. Our SDEs accurately track the behavior of the respective optimizers, especially when compared to previous SDEs derived for Adam and RMSprop. We believe our approach can provide valuable insights into best training practices and novel scaling rules.

## 1 Introduction

Adaptive optimizers lay the foundation for effectively training of modern deep learning models. These methods are typically employed to optimize an objective function expressed as a sum across $N$ individual data points: $\min_{x \in \mathbb{R}^d} [f(x) := \frac{1}{N} \sum_{i=1}^{N} f_i(x)]$, where $f, f_i : \mathbb{R}^d \to \mathbb{R}, \ i = 1, \ldots, N$. Due to the practical difficulties of selecting the learning rate of stochastic gradient descent, adaptive methods have grown in popularity over the past decade. At a high level, these optimizers adjust the learning rate for each parameter based on the historical gradients. Popular optimizers that belong to this family are RMSprop (Tieleman and Hinton, 2012), Adam (Kingma and Ba, 2015), SignSGD (Bernstein et al., 2018), AdamW (Loshchilov and Hutter, 2019), and many other variants. SignSGD is often used for compressing gradients in distributed machine learning (Karimireddy et al., 2019a), but it also has gained popularity due to its connection to RMSprop and Adam (Balles and Hennig, 2018). The latter algorithms have emerged as the standard methods for training modern large language models, partly because of enhancements in signal propagation (Noci et al., 2022).

Although adaptive methods are widely favored in practice, their theoretical foundations remain enigmatic. Recent research has illuminated some of their advantages: Zhang et al. (2020b) demonstrated how gradient clipping addresses heavy-tailed gradient noise, Pan and Li (2022) related the success of Adam over SGD to sharpness, and Yang et al. (2024) showed that adaptive methods handle large gradients better than SGD. At the same time, many optimization studies focus on worst-case convergence rates: These rates (e.g., Défossez et al. (2022)) are valuable, yet they provide an incomplete depiction of algorithm behavior, showing no quantifiable advantage over standard SGD. One particular aspect still lacking clarity is the precise role of noise in the algorithm trajectory.

Our investigation aims to study how gradient noise influences the dynamics of adaptive optimizers and how it impacts their asymptotic behaviors in terms of expected loss and stationary distribution. In particular, we want to understand which algorithms are more resilient to high (possibly heavy-tailed) gradient noise levels. To do this, we rely on stochastic differential equations (SDEs) which have become popular in the literature to study the behavior of optimization algorithms (Li et al., 2017; Jastrzebski et al., 2018). These continuous-time models unlock powerful tools from Itô calculus, enabling us to establish convergence bounds, determine stationary distributions, unveil implicit regularization, and elucidate the intricate interplay between landscape and noise. Notably, SDEs facilitate direct comparisons between optimizers by explicitly illustrating how each hyperparameter and certain landscape features influence their dynamics (Compagnoni et al., 2024).

We begin by analyzing SignSGD, showing how the signal-to-noise ratio affects its dynamics and elucidating the impact of noise at convergence. After analyzing the case where the gradient noise exhibits infinite variance, we extend our analysis to Adam and RMSprop with decoupled weight decay (Loshchilov and Hutter, 2019) – i.e. AdamW and RMSpropW: for both, we refine batch size scaling rules and compare the role of noise to SignSGD. Our analysis provides some theoretical grounding for the resilience of these adaptive methods to high noise levels. Importantly, we highlight that Adam and RMSprop are byproducts of our analysis and that our novel SDEs are derived under much weaker and more realistic assumptions than those in the literature (Malladi et al., 2022).

**Contributions**   We identify our key contributions as follows:

1. We derive the first SDE for SignSGD under very general assumptions: We show that SignSGD exhibits three different phases of the dynamics and characterize the loss behavior in these phases, including the stationary distribution and asymptotic loss value.

2. We demonstrate that for SignSGD, noise inversely affects the convergence rate of both the loss and the iterates. Differently, it has a linear impact on the asymptotic expected loss and the asymptotic variance of the iterates. This is in contrast to SGD, where noise does not influence the convergence speed, but it has a quadratic effect on the loss and variance of the iterates. Finally, we show that, even if the noise has infinite variance, SignSGD is very resilient: its performance is only marginally impacted. In the same conditions, SGD would diverge.

3. We derive new, improved, SDEs for AdamW and RMSpropW and use them to (1) show a novel batch size scaling rule and (2) inspect the stationary distribution and stationary loss value in convex quadratics. In particular, we dive into the properties of weight decay: while for vanilla Adam and RMSprop the effect of noise at convergence mimics SignSGD, something different happens in AdamW and RMSpropW — Due to an intricate interaction between noise, curvature, and regularization, weight decay plays a crucial stabilization role at high noise levels near the minimizer.

4. We empirically verify every theoretical insight we derive. Importantly, we integrate our SDEs with Euler-Maruyama to confirm that our SDEs faithfully track their respective optimizers. We do so on an MLP, a CNN, a ResNet, and a Transformer. For RMSprop and Adam, our SDEs exhibit superior modeling power than the SDEs already existing in the literature.

## 2   Related work

**SDE approximations and applications.**   (Li et al., 2017) introduced a formal theoretical framework aimed at deriving SDEs that effectively model the inherent stochastic nature of optimizers. Ever since, SDEs have found several applications in the field of machine learning, for instance in connection with *stochastic optimal control* to select the stepsize (Li et al., 2017, 2019) and batch size (Zhao et al., 2022), the derivation of *convergence bounds* and *stationary distributions* (Compagnoni et al., 2023, 2024), *implicit regularization* (Smith et al., 2021), and *scaling rules* (Jastrzebski et al., 2018). Previous work by Malladi et al. (2022) has already made strides in deriving SDE models for RMSprop and Adam, albeit under certain restrictive assumptions. They establish a scaling rule which they assert remains valid throughout the entirety of the dynamics. Unfortunately, their derivation is based on the approach of Jastrzebski et al. (2018) which is problematic in the general case (See Appendix E for a detailed discussion). Indeed, we demonstrate that the SDEs derived in Malladi et al. (2022) are only accurate around minima, indicating that their scaling rule is not *globally* valid. (Zhou et al., 2020a) also claimed to have derived a Lévy SDE for Adam. Unfortunately, the quality of their SDE approximation does not come with theoretical guarantees. Additionally, their SDE has random

coefficients: an approach which is theoretically sound in very limited settings (Kohatsu-Higa et al., 1997; Bishop and Del Moral, 2019). Zhou et al. (2024) informally presented an SDE for (only) the parameters of AdamW: this is achieved under strong assumptions and various approximations, some of which are hard to motivate formally.

**Influence of noise on convergence.** Several empirical papers demonstrate that adaptive algorithms adjust better to the noise during training. Specifically, (Zhang et al., 2020b) noticed a consistent gap in the performance of SGD and Adam on language models and connected that phenomenon with heavy-tailed noise distributions. (Pascanu et al., 2013) suggests using gradient clipping to deal with heavy tail noise, and consequently several follow-up works analyzed clipped SGD under heavy-tailed noise (Zhang et al., 2020a; Mai and Johansson, 2021; Puchkin et al., 2024). Kunstner et al. (2024) present thorough numerical experiments illustrating that a significant contributor to heavy-tailed noise during language model training is class imbalance, where certain words occur much more frequently than others. They demonstrate that adaptive optimization methods such as Adam and SignSGD can better adapt to such class imbalances. However, the theoretical understanding of the influence of noise in the context of adaptive algorithms is much more limited. The first convergence results on Adam and RMSprop were derived under bounded stochastic gradients assumption (De et al., 2018; Zaheer et al., 2018; Chen et al., 2019; Défossez et al., 2022). Later, this noise model was relaxed to weak growth condition (Zhang et al., 2022; Wang et al., 2022) and its coordinate-wise version (Hong and Lin, 2023; Wang et al., 2024) and sub-gaussian noise (Li et al., 2023a). SignSGD and its momentum version Signum were originally studied as a method for compressed communication (Bernstein et al., 2018) under bounded variance assumption, but with a requirement of large batches. Several works provided counterexamples where SignSGD fails to converge if stochastic and full gradients are not correlated enough (Karimireddy et al., 2019b; Safaryan and Richtarik, 2021). In the case of AdamW, (Zhou et al., 2022, 2024) provide convergence guarantees under restrictive assumptions such as bounded gradient and bounded noise. All aforementioned results only show that SignSGD, Adam, and RMSprop at least do not perform worse than vanilla SGD. None of them studied how noise affects the dynamics of the algorithm: In this work, we attempt to close this gap.

# 3 Formal statements & insights: the SDEs

This section provides the general formulations of the SDEs of SignSGD (Theorem 3.2) and AdamW (Theorem 3.12). Due to the technical nature of the analysis, we refer the reader to the appendix for the complete formal statements and proofs.

**Assumptions and notation.** In this section, we assume that $\nabla f_\gamma(x) = \nabla f(x) + Z(x), \mathbb{E}[Z(x)] = 0$ and, unless we study the cases where the gradient variance is unbounded, we write $Cov(Z(x)) = \Sigma(x)$ where we omit the batch size unless relevant. To derive the stationary distribution around an optimum, we will approximate the loss function with a quadratic convex function $f(x) = \frac{1}{2}x^\top H x$ as commonly done in the literature (Ge et al., 2015; Levy, 2016; Jin et al., 2017; Poggio et al., 2017; Mandt et al., 2017; Compagnoni et al., 2023). Regarding the notation, $\eta > 0$ is the step size, the mini-batches $\{\gamma_k\}$ are of size $B \geq 1$ and modeled as i.i.d. random variables uniformly distributed on $\{1, \ldots, N\}$. The $\beta$ parameters refer to momentum parameters, $\gamma > 0$ is the (decoupled) $L^2$-regularization parameter, and $\epsilon > 0$ is a small scalar used for numerical stability.

The following definition formalizes the idea that an SDE can be a "good model" to describe an optimizer. It is drawn from the field of numerical analysis of SDEs (see Mil'shtein (1986)) and it quantifies the disparity between the discrete and the continuous processes.

**Definition 3.1** (Weak Approximation). A continuous-time stochastic process $\{X_t\}_{t \in [0,T]}$ is an order $\alpha$ weak approximation (or $\alpha$-order SDE) of a discrete stochastic process $\{x_k\}_{k=0}^{\lfloor T/\eta \rfloor}$ if for every polynomial growth function $g$, there exists a positive constant $C$, independent of the stepsize $\eta$, such that $\max_{k=0,\ldots,\lfloor T/\eta \rfloor} |\mathbb{E}g(x_k) - \mathbb{E}g(X_{k\eta})| \leq C\eta^\alpha$.

## 3.1 SignSGD SDE

In this section, we derive an SDE model for SignSGD, which we believe to be a novel addition to the existing literature. This derivation will reveal the unique manner in which noise influences the dynamics of SignSGD. First, we recall the update equation of SignSGD:

$$x_{k+1} = x_k - \eta \text{sign}\left(\nabla f_{\gamma_k}(x_k)\right).$$

(1)

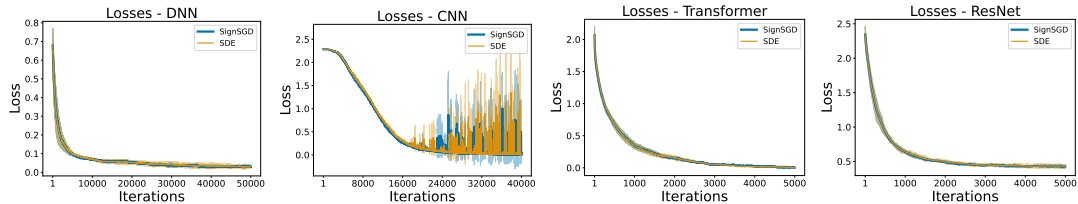

Figure 1: Comparison of SignSGD and its SDE in terms of $f(x)$: Our SDE successfully tracks the dynamics of SignSGD on several architectures: DNN on the Breast Cancer dataset (Left); CNN on MNIST (Center-Left); Transformer on MNIST (Center-Right); ResNet on CIFAR-10 (Right).

The following theorem derives a formal continuous-time model for SignSGD.

**Theorem 3.2** (Informal Statement of Theorem C.5). *Under sufficient regularity conditions, the solution of the following SDE is an order $1$ weak approximation of the discrete update of SignSGD:*

$$dX_t = -(1 - 2\mathbb{P}(\nabla f_\gamma(X_t) < 0))dt + \sqrt{\eta}\sqrt{\bar{\Sigma}(X_t)}dW_t, \tag{2}$$

*where $\bar{\Sigma}(x)$ is the noise covariance $\bar{\Sigma}(x) = \mathbb{E}[\xi_\gamma(x)\xi_\gamma(x)^\top]$ and $\xi_\gamma(x) := sign(\nabla f_\gamma(x)) - 1 + 2\mathbb{P}(\nabla f_\gamma(x) < 0)$ the noise in the sample sign $(\nabla f_\gamma(x))$.*

For didactic reasons, we next present a corollary of Theorem 3.2 that provides a more interpretable SDE. Figure 1 shows the empirical validation of this model for various neural network classes: All details are presented in Appendix F.

**Corollary 3.3** (Informal Statement of Corollary C.7). *Under the assumptions of Theorem 3.2, and that the stochastic gradient is $\nabla f_\gamma(x) = \nabla f(x) + Z$ such that $Z \sim \mathcal{N}(0, \Sigma)$, $\Sigma = \mathrm{diag}(\sigma_1^2, \cdots, \sigma_d^2)$, the following SDE provides a $1$ weak approximation of the discrete update of SignSGD*

$$dX_t = -Erf\left(\frac{\Sigma^{-\frac{1}{2}}\nabla f(X_t)}{\sqrt{2}}\right)dt + \sqrt{\eta}\sqrt{I_d - \mathrm{diag}\left(Erf\left(\frac{\Sigma^{-\frac{1}{2}}\nabla f(X_t)}{\sqrt{2}}\right)\right)^2}dW_t, \tag{3}$$

*where the error function $Erf(x)$ and the square are applied component-wise.*

While Eq. (3) may appear intricate at first glance, it becomes apparent upon closer inspection that the properties of the $\mathrm{Erf}(\cdot)$ function enable a detailed exploration of the dynamics of SignSGD. In particular, we demonstrate that the dynamics of SignSGD can be categorized into three distinct phases. The left of Figure 2 empirically verifies this result on a convex quadratic function.

**Lemma 3.4.** *Under the assumptions of Corollary 3.3 and signal-to-noise ratio $Y_t := \frac{\Sigma^{-\frac{1}{2}}\nabla f(X_t)}{\sqrt{2}}$,*

1. **Phase 1:** *If $|Y_t| > \frac{3}{2}$, the SDE coincides with the ODE of SignGD:*

$$dX_t = -sign(\nabla f(X_t))dt; \tag{4}$$

2. **Phase 2:** *If $1 < |Y_t| < \frac{3}{2}$:*[1]

   (a) $mY_t + \mathbf{q}^- \le \frac{d\mathbb{E}[X_t]}{dt} \le mY_t + \mathbf{q}^+$;

   (b) *For any $a > 0$, $\mathbb{P}\left[\|X_t - \mathbb{E}[X_t]\|_2^2 > a\right] \le \frac{\eta}{a}\left(d - \|mY_t + \mathbf{q}^-\|_2^2\right)$;*

3. **Phase 3:** *If $|Y_t| < 1$, the SDE is*

$$dX_t = -\sqrt{\frac{2}{\pi}}\Sigma^{-\frac{1}{2}}\nabla f(X_t)dt + \sqrt{\eta}\sqrt{I_d - \frac{2}{\pi}\mathrm{diag}\left(\Sigma^{-\frac{1}{2}}\nabla f(X_t)\right)^2}dW_t. \tag{5}$$

---

[1]Let $m$ and $q_1$ are the slope and intercept of the line secant to the graph of $\mathrm{Erf}(x)$ between the points $(1, \mathrm{Erf}(1))$ and $(\frac{3}{2}, \mathrm{Erf}(\frac{3}{2}))$, while $q_2$ is the intercept of the line tangent to the graph of $\mathrm{Erf}(x)$ and slope $m$, $(\mathbf{q}^+)_i := \begin{cases} q_2 & \text{if } \partial_i f(x) > 0 \\ -q_1 & \text{if } \partial_i f(x) < 0 \end{cases}$, $(\mathbf{q}^-)_i := \begin{cases} q_1 & \text{if } \partial_i f(x) > 0 \\ -q_2 & \text{if } \partial_i f(x) < 0 \end{cases}$, and $\hat{q} := \max(q_1, q_2)$.

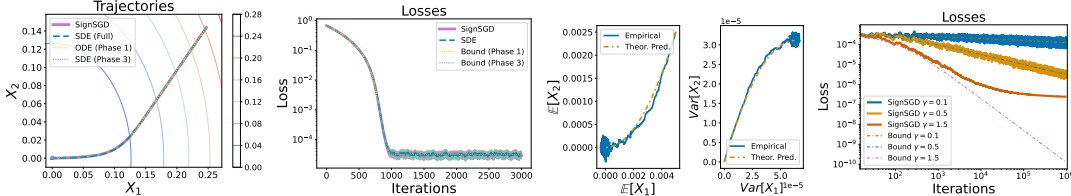

Figure 2: Phases of SignSGD: The ODE of Phase 1 and the SDE of Phase 3 overlap with the "Full" SDE as per Lemma 3.4 (Left); Phases of the Loss: The bounds derived in Lemma 3.5 for the loss during Phase 1 and Phase 3 correctly track the loss evolution (Center-Left); The dynamics of the moments of $X_t$ predicted in Lemma 3.7 track the empirical ones (Center-Right); If the schedulers satisfy the condition in Lemma 3.9, the loss decays to 0 as prescribed. Otherwise, the loss does not converge to 0 (Right).

**Remark:** The behavior of SignSGD depends on the size of the signal-to-noise ratio. In particular, the SDE itself shows that in Phase 3, the inverse of the scale of the noise $\Sigma^{-\frac{1}{2}}$ premultiplies the gradient, thus affecting the rate of descent. This is not the case for SGD where $\Sigma$ only influences the diffusion term.[2] To better understand the role of the noise, we need to study how it affects the dynamics of the loss and compare it with SGD.

**Lemma 3.5.** *Let $f$ be $\mu$-strongly convex, $Tr(\nabla^2 f(x)) \leq \mathcal{L}_\tau$, and $S_t := f(X_t) - f(X_*)$. Then, during*

> 1. *Phase 1, the loss will reach 0 before $t_* = 2\sqrt{\frac{S_0}{\mu}}$ because $S_t \leq \frac{1}{4}\left(\sqrt{\mu}t - 2\sqrt{S_0}\right)^2$;*

> 2. *Phase 2 with $\Delta := \left(\frac{m}{\sqrt{2}\sigma_{max}} + \frac{\eta\mu m^2}{4\sigma_{max}^2}\right)$: $\mathbb{E}[S_t] \leq S_0 e^{-2\mu\Delta t} + \frac{\eta}{2}\frac{(\mathcal{L}_\tau - \mu d\hat{q}^2)}{2\mu\Delta}\left(1 - e^{-2\mu\Delta t}\right)$;*

> 3. *Phase 3 with $\Delta := \left(\sqrt{\frac{2}{\pi}}\frac{1}{\sigma_{max}} + \frac{\eta}{\pi}\frac{\mu}{\sigma_{max}^2}\right)$: $\mathbb{E}[S_t] \leq S_0 e^{-2\mu\Delta t} + \frac{\eta}{2}\frac{\mathcal{L}_\tau}{2\mu\Delta}\left(1 - e^{-2\mu\Delta t}\right)$.*

In Phase 1, the signal-to-noise ratio is large, meaning that SignSGD behaves like SignGD: Consistently with the analysis of SignGD in (Ma et al., 2022), this explains the fast initial convergence of the optimizer as well as of RMSprop and Adam. In this phase, the loss undergoes a steady decrease which ensures the emergence of Phase 2 which in turn triggers that of Phase 3 which is characterized by an exponential decay to an asymptotic loss level: As a practical example, we verify the dynamics of the expected loss around a minimum in the center-left of Figure 2.

**Lemma 3.6.** *For SGD, the expected loss satisfies: $\mathbb{E}[S_t] \leq S_0 e^{-2\mu t} + \frac{\eta}{2}\frac{\mathcal{L}_\tau \sigma_{max}^2}{2\mu}\left(1 - e^{-2\mu t}\right)$.*

**Remark:** The two key observations are that:

> 1. Both in Phase 2 and Phase 3, the noise level $\sigma_{max}$ inversely affects the exponential convergence speed, while this trend is not observed with SGD;

> 2. The asymptotic loss of SignSGD is (almost) linear in $\sigma_{max}$ while that of SGD is quadratic.

Additionally, we characterize the stationary distribution of SignSGD around a minimum: Empirical validation is provided in the center-right of Figure 2.

**Lemma 3.7.** *Let $H = \text{diag}(\lambda_1, \ldots, \lambda_d)$ and $M_t := e^{-2\left(\sqrt{\frac{2}{\pi}}\Sigma^{-\frac{1}{2}}H + \frac{\eta}{\pi}\Sigma^{-\frac{1}{2}}H^2\right)t}$. Then,*

1. $\mathbb{E}[X_t] = e^{-\sqrt{\frac{2}{\pi}}\Sigma^{-\frac{1}{2}}Ht}X_0 \overset{t\to\infty}{\to} 0$;

2. $Cov[X_t] = \left(M_t - e^{-2\sqrt{\frac{2}{\pi}}\Sigma^{-\frac{1}{2}}Ht}\right)X_0^2 + \frac{\eta}{2}\left(\sqrt{\frac{2}{\pi}}I_d + \frac{\eta}{\pi}H\right)^{-1}H^{-1}\Sigma^{\frac{1}{2}}\left(I_d - M_t\right)$,

   *which as $t \to \infty$ converges to $\frac{\eta}{2}\left(\sqrt{\frac{2}{\pi}}I_d + \frac{\eta}{\pi}H\right)^{-1}H^{-1}\Sigma^{\frac{1}{2}}$.*

**Lemma 3.8.** *Under the same assumptions as Lemma 3.7, the stationary distribution for SGD is:*

$$\mathbb{E}[X_t] = e^{-Ht}X_0 \overset{t\to\infty}{\to} 0 \quad and \quad Cov[X_t] = \frac{\eta}{2}H^{-1}\Sigma\left(I_d - e^{-2Ht}\right) \overset{t\to\infty}{\to} \frac{\eta}{2}H^{-1}\Sigma.$$

---

[2]Ths SDE of SGD is $dX_t = -\nabla f(X_t)dt + \sqrt{\eta}\Sigma^{\frac{1}{2}}dW_t$.

As we observed above, the noise inversely affects the convergence rate of the iterates of SignSGD while it does not impact that of SGD. Additionally, while both covariance matrices essentially scale inversely to the hessian, that of SignSGD scales with $\Sigma^{\frac{1}{2}}$ while that of SGD scales with $\Sigma$.

We conclude this section by presenting a condition on the step size scheduler that ensures the asymptotic convergence of the expected loss to 0 in Phase 3. For general schedulers, we characterize precisely the speed of convergence and the factors influencing it. Empirical validation is provided in the right of Figure 2 for a convex quadratic.

**Lemma 3.9.** *Under the assumptions of Lemma 3.5, any step size scheduler $\eta_t$ such that*

$$\int_0^\infty \eta_s ds = \infty \ and \ \lim_{t\to\infty} \eta_t = 0 \implies \mathbb{E}[f(X_t) - f(X_*)] \overset{t\to\infty}{\lesssim} \frac{\mathcal{L}_\tau \sigma_{max}}{4\mu} \sqrt{\frac{\pi}{2}} \eta_t \overset{t\to\infty}{\to} 0. \quad (6)$$

**Remark:** Under the same conditions, SGD satisfies $\mathbb{E}[f(X_t) - f(X_*)] \overset{t\to\infty}{\lesssim} \frac{\mathcal{L}_\tau \sigma_{max}^2}{4\mu} \eta_t \overset{t\to\infty}{\to} 0$.

**Conclusion:** As noted in Bernstein et al. (2018), the signal-to-noise ratio is key in determining the dynamics of SignSGD. Our SDEs help clarify the mechanisms underlying the dynamics of SignSGD: we show that the effect of noise is radically different from SGD: 1) It affects the rate of convergence of the iterates, of the covariance of the iterates, and of the expected loss; 2) The asymptotic loss value and covariance of the iterates scale in $\Sigma^{\frac{1}{2}}$ while for SGD it does so in $\Sigma$. On the one hand, low levels of noise will ensure a faster and steadier loss decrease close to minima for SignSGD than for SGD. On the other, SGD will converge to much lower loss values. A symmetric argument holds for high levels of noise, which suggests that SignSGD is more resilient to high levels of noise.

### 3.1.1 Heavy-tailed noise

Interestingly, we can replicate the efforts above also in case the noise structure is heavy-tailed as it is distributed according to a Student's t distribution. Notably, we derive the SDE for the case where the noise has infinite variance and show how little marginal effect this has on the dynamics of SignSGD.

**Lemma 3.10.** *Under the assumptions of Corollary 3.3 but the noise on the gradients $U \sim t_\nu(0, I_d)$ where $\nu \in \mathbb{Z}^+$: The following SDE is a 1 weak approximation of the discrete update of SignSGD*

$$dX_t = -2\Xi\left(\Sigma^{-\frac{1}{2}}\nabla f(X_t)\right) dt + \sqrt{\eta}\sqrt{I_d - 4\operatorname{diag}\left(\Xi\left(\Sigma^{-\frac{1}{2}}\nabla f(X_t)\right)\right)^2} dW_t, \quad (7)$$

*where $\Xi(x)$ is defined as $\Xi(x) := x\frac{\Gamma\left(\frac{\nu+1}{2}\right)}{\sqrt{\pi\nu}\Gamma\left(\frac{\nu}{2}\right)} {}_2F_1\left(\frac{1}{2}, \frac{\nu+1}{2}; \frac{3}{2}; -\frac{x^2}{\nu}\right)$ and ${}_2F_1(a, b; c; x)$ is the hyper-geometric function. Above, the function $\Xi(x)$ and the square are applied component-wise.*

We now characterize the dynamics of SignSGD when the noise on the gradient has infinite variance.

**Corollary 3.11.** *Under the assumptions of Lemma 3.10 and $\nu = 2$, the dynamics in Phase 3 is:*

$$dX_t = -\sqrt{\frac{1}{2}}\Sigma^{-\frac{1}{2}}\nabla f(X_t)dt + \sqrt{\eta}\sqrt{I_d - \frac{1}{2}\operatorname{diag}\left(\Sigma^{-\frac{1}{2}}\nabla f(X_t)\right)^2} dW_t. \quad (8)$$

**Conclusion:** We observe that the dynamics of SignSGD when the noise is Gaussian (Eq. (5)) and when the noise is heavy-tailed with unbounded variance (Eq. (8)) are very similar: By comparing the constants in front of the drift terms $\Sigma^{-\frac{1}{2}}\nabla f(X_t)$, they are only $\sim 10\%$ apart, and the diffusion coefficients are comparable. Not only do we once more showcase the resilience of SignSGD to high levels of noise, but in alignment with (Zhang et al., 2020b), we provide theoretical support to the success of Adam in such a scenario where SGD would diverge.

All the results derived above can be extended to this setting: this is left as an exercise for the reader.

### 3.2 AdamW SDE

In the last subsection, we showcased how SDEs can serve as powerful tools to understand the dynamics of the simplest among coordinate-wise adaptive methods: SignSGD. Here, we extend the

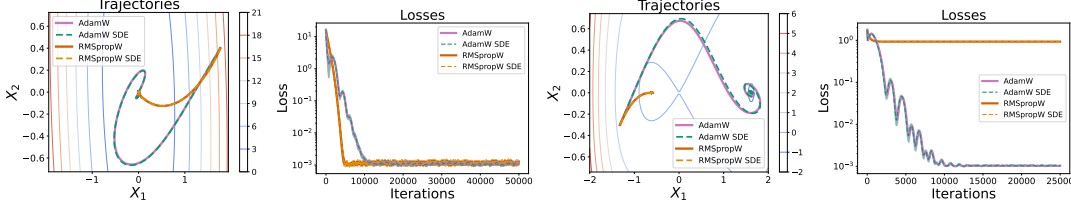

Figure 3: The first two images compare the SDEs of AdamW and RMSpropW with the respective optimizers in terms of trajectories and $f(x)$ for a convex quadratic function while the other two figures provide a comparison for an embedded saddle. In all cases, we observe good agreements.

discussion to Adam with decoupled weight decay, i.e. AdamW:

$$v_{k+1} = \beta_2 v_k + (1 - \beta_2) \left( \nabla f_{\gamma_k}(x_k) \right)^2, \quad m_{k+1} = \beta_1 m_k + (1 - \beta_1) \nabla f_{\gamma_k}(x_k),$$

$$x_{k+1} = x_k - \eta \frac{\hat{m}_{k+1}}{\sqrt{\hat{v}_{k+1}} + \epsilon} - \eta \gamma x_k, \quad \hat{m}_k = \frac{m_k}{1 - \beta_1^k}, \quad \hat{v}_k = \frac{v_k}{1 - \beta_2^k}, \tag{9}$$

which, of course, covers Adam, RMSprop, and RMSpropW depending on the values of $\gamma$ and $\beta_1$.

The following result proves the SDE of AdamW which we validate in Figure 3 for two simple landscapes and in Figure 4 for a Transformer and a ResNet.

**Theorem 3.12** (Informal Statement of Theorem C.31). *Under sufficient regularity conditions, $\rho_1 = \mathcal{O}(\eta^{-\zeta})$ s.t. $\zeta \in (0, 1)$, and $\rho_2 = \mathcal{O}(1)$, the order $1$ weak approximation of AdamW is:*

$$dX_t = -\frac{\sqrt{\gamma_2(t)}}{\gamma_1(t)} P_t^{-1} (M_t + \eta \rho_1 \left( \nabla f \left( X_t \right) - M_t \right)) dt - \gamma X_t dt \tag{10}$$

$$dM_t = \rho_1 \left( \nabla f \left( X_t \right) - M_t \right) dt + \sqrt{\eta} \rho_1 \Sigma^{1/2} \left( X_t \right) dW_t \tag{11}$$

$$dV_t = \rho_2 \left( (\nabla f(X_t))^2 + \operatorname{diag} \left( \Sigma \left( X_t \right) \right) - V_t \right) dt, \tag{12}$$

*where $\beta_i = 1 - \eta \rho_i \sim 1$, $\gamma_i(t) = 1 - e^{-\rho_i t}$, and $P_t = \operatorname{diag} \sqrt{V_t} + \epsilon \sqrt{\gamma_2(t)} I_d$.*

In contrast to *Remark 4.3* of Malladi et al. (2022), which suggests that an SDE for RMSprop and Adam is only viable if $\sigma \gg \|\nabla f(x)\|$ and $\sigma \sim \frac{1}{\eta}$, our derivation that does not need these assumptions: See Remark C.25 for a deeper discussion, the implications, and the experimental comparison.

The following result demonstrates how the asymptotic expected loss of AdamW scales with the noise level. Notably, it introduces the first scaling rule for AdamW, extending the one proposed for Adam in (Malladi et al., 2022) to include weight decay scaling. It is crucial to understand that, unlike the typical approach in the literature (see (Jastrzebski et al., 2018; Malladi et al., 2022)), our objective in deriving these rules is not to maintain the dynamics of the optimizers or the SDE unchanged. Instead, our goal is to offer a practical strategy for adjusting hyperparameters (e.g., from $\eta$ to $\tilde{\eta}$) to retain certain performance metrics or optimizer properties as the batch size increases (e.g., from $B$ to $\tilde{B}$). Therefore, in our upcoming analysis, we aim to derive scaling rules that preserve specific relevant aspects of the dynamics, such as the convergence bound on the loss or the speed. For a more detailed discussion motivating our approach, see Appendix E.

**Lemma 3.13.** *If $f$ is $\mu$-strongly convex and $L$-smooth, $\mathcal{L}_\tau := Tr(\nabla^2 f(x))$, and $(\nabla f(x))^2 = \mathcal{O}(\eta)$, $\tilde{\eta} = \kappa \eta$, $\tilde{B} = B\delta$, and $\tilde{\rho}_i = \alpha_i \rho_i$, and $\tilde{\gamma} = \xi \gamma$, AdamW satisfies*

$$\mathbb{E}[f(X_t) - f(X_*)] \overset{t \to \infty}{\leq} \frac{\eta \mathcal{L}_\tau \sigma L}{2} \frac{\kappa}{2\mu\sqrt{B\delta}L + \sigma\xi\gamma(L + \mu)}. \tag{13}$$

*We derive the novel scaling rule by 1) Preserving the upper bound, which requires that $\kappa = \sqrt{\delta}$ and $\xi = \kappa$; 2) Preserving the relative speed of $M_t$, $V_t$ and $X_t$, which requires that $\tilde{\beta}_i = 1 - \kappa^2(1 - \beta_i)$.*

The left of Figure 5 shows the empirical verification of the predicted loss value and scaling rule on a convex quadratic function.[3] Interestingly, and consistently with Lemma 3.13, such a value is not

---

[3]Table 1 in Appendix F.8 shows that our scaling rule works on DNNs: it confirms that failing to rescale the weight decay parameter is suboptimal.

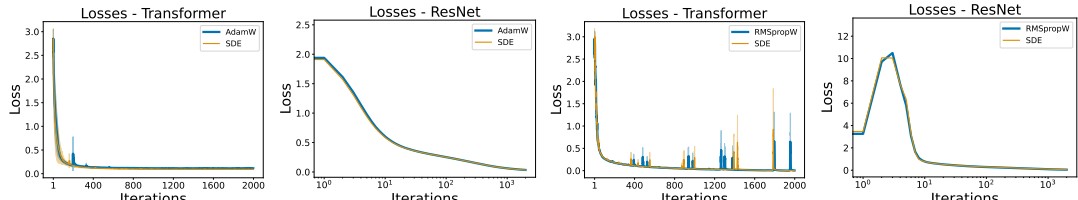

Figure 4: The first two represent the comparison between AdamW and its SDE in terms of $f(x)$. The other two do the same for RMSpropW. In both cases, the first is a Transformer on MNIST and the second a ResNet on CIFAR-10: Our SDEs match the respective optimizers.

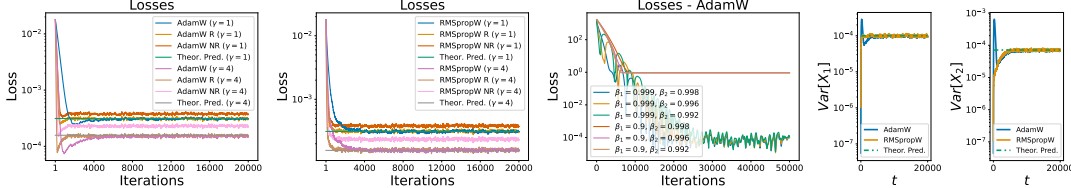

Figure 5: The loss predicted in Lemma 3.13 matches the experimental results on a convex quadratic function. *AdamW* is run with regularization parameter $\gamma = 1$. *AdamW R* (AdamW Rescaled) is run as we apply the scaling rule with $\kappa = 2$. *AdamW NR* (AdamW **Not** Rescaled) is run as we apply the scaling rule with $\kappa = 2$ on all hyperparameters but $\gamma$, which is left unchanged: Our scaling rule holds, and failing to rescale $\gamma$ leads the optimizer not to preserve the asymptotic loss level. The same happens for $\gamma = 4$ (Left); The same for RMSpropW (Center-Left); For AdamW, $\beta_1$ and $\beta_2$ influence which basin will attract the dynamics and how fast this will converge, but not the asymptotic loss level inside the basin (Center-Right). For both AdamW and RMSpropW, the variance at convergence predicted in Lemma 3.14 matches the experimental results (Right).

influenced by the choice of $\beta_i$: We argue that $\beta_i$ do not impact the asymptotic level of the loss, but rather drive the selection of the basin and speed at which AdamW converges to it — The center-right of Figure 5 exemplifies this on a simple nonconvex landscape.

We conclude this section with the stationary distribution of AdamW around a minimum which we empirically validate on the right of Figure 5.

**Lemma 3.14.** *The stationary distribution of AdamW is*

$$\left(\mathbb{E}[X_\infty], Cov[X_\infty]\right) = \left(0, \frac{\eta}{2}\left(I_d + \gamma H^{-1}\Sigma^{\frac{1}{2}}\right)^{-1} H^{-1}\Sigma^{\frac{1}{2}}\right).$$

**RMSpropW** We derived the same results for RMSprop(W) and we reported them in Appendix C.4: importantly, we validate the SDE in Figure 3 for two simple landscapes and in Figure 4 for a Transformer and a ResNet. The results regarding the asymptotic loss level and stationary distributions are validated in the center-left and right of Figure 5 for a convex quadratic function.

**Conclusion:** While for both SignSGD and Adam the asymptotic loss value and the covariance of the iterates scale linearly with $\Sigma^{\frac{1}{2}}$, we observe for AdamW this is more intricate: The interaction between curvature, noise, and regularization implies that these two quantities are upper-bounded in $\Sigma^{\frac{1}{2}}$ and increasing $\Sigma$ to infinity does not lead to their explosion: Weight decay plays a crucial stabilization role at high noise levels near the minimizer — See Figure 6 for a comparison across optimizers. Finally, we argue that $\beta_i$ play a key role in selecting the basin and the convergence speed to the asymptotic loss value rather than impacting the loss value itself.

## 4 Experiments: SDE validation

The point of our experiments is to validate the theoretical results derived from the SDEs. Therefore, we first show that our SDEs faithfully represent the dynamics of their respective optimizers. To do

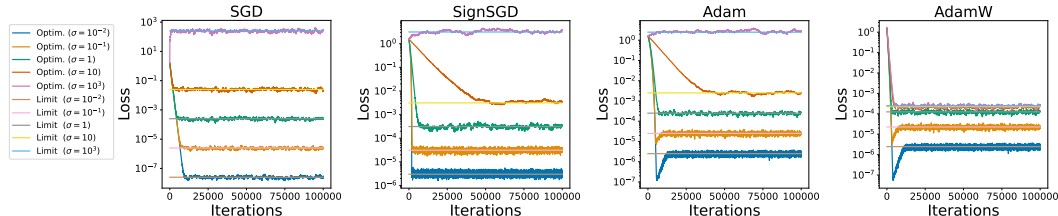

Figure 6: For SGD (Left), SignSGD (Center-Left), Adam (Center-Right), and AdamW: For each *optimizer*, we plot the loss value on a convex quadratic and compare its asymptotic value with the *limits* predicted by our theory. As we take $\Sigma = \sigma^2 I_d$, we confirm that the loss of SGD scales quadratically in $\sigma$ (Lemma 3.6), and linearly for SignSGD (Lemma 3.5) and Adam (Lemma 3.13 with $\gamma = 0$). For AdamW, the maximum asymptotic loss value is bounded in $\sigma$ (Lemma 3.13 with $\gamma > 0$). In accordance with the experiments, our theory predicts that adaptive methods are more resilient to noise.

so, we integrate the SDEs with Euler-Maruyama (Algorithm 1): This is particularly challenging and expensive as one needs to calculate the full gradients of the DNNs at each iteration.[4] We present the first set of validation experiments on a variety of architectures and datasets: An MLP on the Breast Cancer dataset, a CNN and a Transformer on MNIST, and a ResNet on CIFAR-10. All details are in Appendix F.

## 5   Conclusion

We derived the first formal SDE for SignSGD, enabling us to demonstrate its dynamics traversing three discernible phases. We characterize how the signal-to-noise ratio drives the dynamics of the loss in each of these phases, and we derive the asymptotic value of the loss function, as well as the stationary distribution. Regarding the role of noise, we draw a straightforward comparison with SGD. For SignSGD, the noise level $\sqrt{\Sigma}$ has an inverse linear effect on the convergence speed of the loss and the iterates. However, it linearly affects the asymptotic expected loss and the asymptotic variance of the iterates. In contrast, for SGD, noise does not influence the convergence speed but has a quadratic impact on the loss level and variance. We also examine the scenario where the noise has infinite variance and demonstrate the resilience of SignSGD, showing that its performance is only marginally affected. Finally, we generalize the analysis to include AdamW and RMSpropW. Specifically, we leverage our novel SDEs to derive the asymptotic value of the loss function, their stationary distribution on a convex quadratic, and a novel scaling rule. The key insight is that, similarly to SignSGD, the loss level and covariance matrix of the iterates of Adam and RMSprop scale linearly in the noise level $\Sigma^{\frac{1}{2}}$. For AdamW and RMSpropW, the complex interaction of noise, curvature, and regularization implies that these two quantities are bounded in terms of $\Sigma^{\frac{1}{2}}$, showing that weight decay plays a crucial stabilization role at high noise levels near the minimizer. Interestingly, the SDEs for Adam and RMSprop are straightforward corollary of our general results and were derived under much less restrictive and more realistic assumptions than those in the literature. Finally, we thoroughly validate all our theoretical results: We compare the dynamics of the various optimizers with the respective SDEs and find good agreement on simple landscapes and deep neural networks. For Adam and RMSprop, our SDEs track them better than those derived in (Malladi et al., 2022).

**Future work**   We believe that our results can be extended to other optimizers commonly used in practice such as Signum, AdaGrad, AdaMax, and Nadam. Additionally, inspired by the insights from our SDE analysis, there is potential for designing new optimization algorithms that combine the strengths of existing methods while mitigating their weaknesses. For example, developing hybrid optimizers that adaptively switch between different strategies based on the training phase or current state of the optimization process could offer superior performance.

---

[4]Many papers derived SDEs to model optimizers: most of them do not validate them, some do so on quadratic functions, and Paquette et al. (2021); Compagnoni et al. (2023) do it on NNs: See Appendix A for details.

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

496     stochastic gradient descent. *Communications in Mathematical Sciences*.

## A   Additional related works

498 In this section, we list some papers that derived or used SDEs to model optimizers. In particular, we
499 focus on the aspect of empirically verifying the validity of such SDEs in the sense that they indeed
500 track the respective optimizers. We divide these into three categories: Those that did not carry out
501 any type of validation, those that did it on simple landscapes (quadratic functions et similia), and
502 those that did small experiments or neural networks.

503 None of the following papers carried out any experimental validation of the approximating power of
504 the SDEs they derived. Many of them did not even validate the insights derived from the SDEs: (Liu
505 et al., 2021; Hu et al., 2019; Bercher et al., 2020; Zhu and Ying, 2021; Cui et al., 2020; Maulén Soto,
506 2021; Wang and Wu, 2020; Lanconelli and Lauria, 2022; Ayadi and Turinici, 2021; Soto et al., 2022;
507 Li and Wang, 2022; Wang and Mao, 2022; Bardi and Kouhkouh, 2022; Chen et al., 2022; Kunin
508 et al., 2023; Zhang et al., 2023; Sun et al., 2023; Li et al., 2023b; Gess et al., 2024; Dambrine et al.,
509 2024; Maulen-Soto et al., 2024).

510 The following ones carried out validation experiments on artificial landscapes, e.g. quadratic or
511 quartic function, or easy regression tasks: (Li et al., 2017, 2019; Zhou et al., 2020b; An et al., 2020;
512 Fontaine et al., 2021; Gu et al., 2021; Su and Lau, 2023; Ankirchner and Perko, 2024).

513 The following papers carried out some experiments which include neural networks: (Paquette et al.,
514 2021; Compagnoni et al., 2023). In particular, they both simulate the SDEs with a numerical integrator
515 and compare them with the respective optimizers: The first validates the SDE on a shallow MLP
516 while the second does so on a shallow and a deep MLP. Regarding (Li et al., 2021; Malladi et al.,
517 2022), they do not validate their SDEs: Rather, their approach conceptually proceeds as follows:

518       1. Derive an SDE for an optimizer which we now dub "*A*";

519      2. Notice that simulating the SDE is too expensive;

520      3. Define another discrete-time algorithm called SVAG which also has the same SDE as "*A*"
521          but does not numerically integrate the SDE as it does not even require access to it: It does
522          not need access neither to the drift nor to the diffusion term;

523      4. Simulate SVAG and show that it tracks "*A*" successfully;

524      5. Conclude that the SDE is a good approximation for "*A*".

525 However, they never validated that the SDE is a good approximation for "*A*" or for SVAG either.
526 With the same logic, they could have done the following:

527      1. Derive an SDE for "*A*";

528      2. Notice that simulating the SDE is too expensive;

529      3. Define another discrete-time algorithm called "*B*" which coincides with "*A*" and thus of
530          course shares the same SDE;

531      4. Simulate "*B*" and show that it tracks "*A*" perfectly;

532      5. Conclude that the SDE is a good approximation for "*A*".

533 In particular, the only fact they prove is that SVAG is a discrete-time optimizer that shares the same
534 SDE as "*A*" because it describes a discrete trajectory that is a 1st-order approximation of the SDE of
535 "*A*". Technically speaking, "*A*" also does the same. One cannot conclude that the SDE derived for "*A*"
536 is a good model for "*A*" by simply comparing two algorithms "*A*" and "*B*" that share the same SDE.
537 Otherwise, simply comparing an optimizer "*A*" with itself would do the trick. An SDE's empirical
538 validation can only occur if the SDE is simulated with a numerical integrator that requires access to
539 the drift and diffusion terms (Higham, 2001; Milstein, 2013).

## B   Stochastic calculus

541 In this section, we summarize some important results in the analysis of Stochastic Differential
542 Equations Mao (2007); Øksendal (1990). The notation and the results in this section will be used
543 extensively in all proofs in this paper. We assume the reader to have some familiarity with Brownian
544 motion and with the definition of stochastic integral (Ch. 1.4 and 1.5 in Mao (2007)).

### B.1   Itô's Lemma

We start with some notation: Let $(\Omega, \mathcal{F}, \{\mathcal{F}_t\}_{t\geq 0}, \mathbb{P})$ be a filtered probability space. We say that an
event $E \in \mathcal{F}$ holds almost surely (a.s.) in this space if $\mathbb{P}(E) = 1$. We call $\mathcal{L}^p([a,b], \mathbb{R}^d)$, with $p > 0$,
the family of $\mathbb{R}^d$-valued $\mathcal{F}_t$-adapted processes $\{f_t\}_{a\leq t\leq b}$ such that

$$\int_a^b \|f_t\|^p dt \leq \infty.$$

546 Moreover, we denote by $\mathcal{M}^p([a,b], \mathbb{R}^d)$, with $p > 0$, the family of $\mathbb{R}^d$-valued processes $\{f_t\}_{a\leq t\leq b}$
547 in $\mathcal{L}([a,b], \mathbb{R}^d)$ such that $\mathbb{E}\left[\int_a^b \|f_t\|^p dt\right] \leq \infty$. We will write $h \in \mathcal{L}^p(\mathbb{R}_+, \mathbb{R}^d)$, with $p > 0$, if
548 $h \in \mathcal{L}^p([0,T], \mathbb{R}^d)$ for every $T > 0$. Similar definitions hold for matrix-valued functions using the
549 Frobenius norm $\|A\| := \sqrt{\sum_{ij} |A_{ij}|^2}$.

550 Let $W = \{W_t\}_{t\geq 0}$ be a one-dimensional Brownian motion defined on our probability space and let
551 $X = \{X_t\}_{t\geq 0}$ be an $\mathcal{F}_t$-adapted process taking values on $\mathbb{R}^d$.

552 **Definition B.1.** Let the *drift* be $b \in \mathcal{L}^1(\mathbb{R}_+, \mathbb{R}^d)$ and the diffusion term be $\sigma \in \mathcal{L}^2(\mathbb{R}_+, \mathbb{R}^{d\times m})$.
553 $X_t$ is an Itô process if it takes the form

$$X_t = x_0 + \int_0^t b_s ds + \int_0^t \sigma_s dW_s.$$

554 We shall say that $X_t$ has the stochastic differential

$$dX_t = b_t dt + \sigma_t dW_t. \tag{14}$$

555

556 **Theorem B.2** (Itô's Lemma). *Let $X_t$ be an Itô process with stochastic differential $dX_t = b_t dt +$*
557 *$\sigma_t dW_t$. Let $f(x, t)$ be twice continuously differentiable in $x$ and continuously differentiable in $t$,*
558 *taking values in $\mathbb{R}$. Then $f(X_t, t)$ is again an Itô process with stochastic differential*

$$df(X_t, t) = \partial_t f(X_t, t))dt + \langle \nabla f(X_t, t), b_t \rangle dt + \frac{1}{2} Tr\left(\sigma_t \sigma_t^\top \nabla^2 f(X_t, t)\right) dt + \langle \nabla f(X_t, t), \sigma_t \rangle dW_t. \tag{15}$$

559 ## B.2 Stochastic Differential Equations

560 Stochastic Differential Equations (SDEs) are equations of the form

$$dX_t = b(X_t, t)dt + \sigma(X_t, t)dW_t.$$

561 First of all, we need to define what it means for a stochastic process $X = \{X_t\}_{t \geq 0}$ with values in $\mathbb{R}^d$
562 to solve an SDE.

563 **Definition B.3.** Let $X_t$ be as above with deterministic initial condition $X_0 = x_0$. Assume $b :$
564 $\mathbb{R}^d \times [0, T] \to \mathbb{R}^d$ and $\sigma : \mathbb{R}^d \times [0, T] \to \mathbb{R}^{d \times m}$ are Borel measurable; $X_t$ is called a solution to the
565 corresponding SDE if

566 1. $X_t$ is continuous and $\mathcal{F}_t$-adapted;

567 2. $b \in \mathcal{L}^1\left([0, T], \mathbb{R}^d\right)$;

568 3. $\sigma \in \mathcal{L}^2\left([0, T], \mathbb{R}^{d \times m}\right)$;

4. For every $t \in [0, T]$

$$X_t = x_0 + \int_0^t b(X_s, s)ds + \int_0^t \sigma(X_s, s)dW(s) \quad a.s.$$

Moreover, the solution $X_t$ is said to be unique if any other solution $X_t^\star$ is such that

$$\mathbb{P}\left\{X_t = X_t^\star, \text{ for all } 0 \leq t \leq T\right\} = 1.$$

569
570 Notice that since the solution to an SDE is an Itô process, we can use Itô's Lemma. The following
571 theorem gives a sufficient condition on $b$ and $\sigma$ for the existence of a solution to the corresponding
572 SDE.

573 **Theorem B.4.** *Assume that there exist two positive constants $\bar{K}$ and $K$ such that*

    *1. (Global Lipschitz condition) for all $x, y \in \mathbb{R}^d$ and $t \in [0, T]$*

$$\max\{\|b(x, t) - b(y, t)\|^2, \|\sigma(x, t) - \sigma(y, t)\|^2\} \leq \bar{K}\|x - y\|^2;$$

    *2. (Linear growth condition) for all $x \in \mathbb{R}^d$ and $t \in [0, T]$*

$$\max\{\|b(x, t)\|^2, \|\sigma(x, t)\|^2\} \leq K(1 + \|x\|^2).$$

574 *Then, there exists a unique solution $X_t$ to the corresponding SDE, and $X_t \in \mathcal{M}^2([0, T], \mathbb{R}^d)$.*

575 **Numerical approximation.** Often, SDEs are solved numerically. The simplest algorithm to provide
576 a sample path $(\hat{x}_k)_{k \geq 0}$ for $X_t$, so that $X_{k\Delta t} \approx \hat{x}_k$ for some small $\Delta t$ and for all $k\Delta t \leq M$ is called
577 Euler-Maruyama (Algorithm 1). For more details on this integration method and its approximation
578 properties, the reader can check Mao (2007).

---

**Algorithm 1** Euler-Maruyama Integration Method for SDEs

---

**input** The drift $b$, the volatility $\sigma$, and the initial condition $x_0$.
  Fix a stepsize $\Delta t$;
  Initialize $\hat{x}_0 = x_0$;
  $k = 0$;
  **while** $k \leq \left\lfloor \frac{T}{\Delta t} \right\rfloor$ **do**
    Sample some $d$-dimensional Gaussian noise $Z_k \sim \mathcal{N}(0, I_d)$;
    Compute $\hat{x}_{k+1} = \hat{x}_k + \Delta t \, b(\hat{x}_k, k\Delta t) + \sqrt{\Delta t} \, \sigma(\hat{x}_k, k\Delta t) Z_k$;
    $k = k + 1$;
  **end while**
**output** The approximated sample path $(\hat{x}_k)_{0 \leq k \leq \left\lfloor \frac{T}{\Delta t} \right\rfloor}$.

---

## C Theoretical framework - Weak Approximation

In this section, we introduce the theoretical framework used in the paper, together with its assumptions and notations.

First of all, many proofs will use Taylor expansions in powers of $\eta$. For ease of notation, we introduce the shorthand that whenever we write $\mathcal{O}(\eta^\alpha)$, we mean that there exists a function $K(x) \in G$ such that the error terms are bounded by $K(x)\eta^\alpha$. For example, we write

$$b(x + \eta) = b_0(x) + \eta b_1(x) + \mathcal{O}(\eta^2)$$

to mean: there exists $K \in G$ such that

$$|b(x + \eta) - b_0(x) - \eta b_1(x)| \leq K(x)\eta^2.$$

Additionally, we introduce the following shorthand:

- A multi-index is $\alpha = (\alpha_1, \alpha_2, \ldots, \alpha_n)$ such that $\alpha_j \in \{0, 1, 2, \ldots\}$;
- $|\alpha| := \alpha_1 + \alpha_2 + \cdots + \alpha_n$;
- $\alpha! := \alpha_1! \alpha_2! \cdots \alpha_n!$;
- For $x = (x_1, x_2, \ldots, x_n) \in \mathbb{R}^n$, we define $x^\alpha := x_1^{\alpha_1} x_2^{\alpha_2} \cdots x_n^{\alpha_n}$;
- For a multi-index $\beta$, $\partial_\beta^{|\beta|} f(x) := \frac{\partial^{|\beta|}}{\partial_{x_1}^{\beta_1} \partial_{x_2}^{\beta_2} \cdots \partial_{x_n}^{\beta_n}} f(x)$;
- We also denote the partial derivative with respect to $x_i$ by $\partial_{e_i}$.

**Definition C.1** (G Set). Let $G$ denote the set of continuous functions $\mathbb{R}^d \to \mathbb{R}$ of at most polynomial growth, i.e. $g \in G$ if there exists positive integers $\nu_1, \nu_2 > 0$ such that $|g(x)| \leq \nu_1 \left(1 + |x|^{2\nu_2}\right)$, for all $z \in \mathbb{R}^d$.

The next results are inspired by Theorem 1 of Li et al. (2017) and are derived under some regularity assumption on the function $f$.

**Assumption C.2.** Assume that the following conditions on $f, f_i$, and their gradients are satisfied:

- $\nabla f, \nabla f_i$ satisfy a Lipschitz condition: there exists $L > 0$ such that

$$|\nabla f(u) - \nabla f(v)| + \sum_{i=1}^{n} |\nabla f_i(u) - \nabla f_i(v)| \leq L|u - v|;$$

- $f, f_i$ and its partial derivatives up to order 7 belong to $G$;
- $\nabla f, \nabla f_i$ satisfy a growth condition: there exists $M > 0$ such that

$$|\nabla f(x)| + \sum_{i=1}^{n} |\nabla f_i(x)| \leq M(1 + |x|).$$

**Lemma C.3** (Lemma 1 Li et al. (2017)). *Let $0 < \eta < 1$. Consider a stochastic process $X_t, t \geq 0$ satisfying the SDE*

$$dX_t = b\left(X_t\right) dt + \sqrt{\eta} \sigma\left(X_t\right) dW_t$$

*with $X_0 = x \in \mathbb{R}^d$ and $b, \sigma$ together with their derivatives belong to $G$. Define the one-step difference $\Delta = X_\eta - x$, and indicate the $i$-th component of $\Delta$ with $\Delta_i$. Then we have*

1. $\mathbb{E}\Delta_i = b_i\eta + \frac{1}{2}\left[\sum_{j=1}^d b_j \partial_{e_j} b_i\right]\eta^2 + \mathcal{O}\left(\eta^3\right) \quad \forall i = 1, \ldots, d$;

2. $\mathbb{E}\Delta_i\Delta_j = \left[b_i b_j + \sigma\sigma_{(ij)}^T\right]\eta^2 + \mathcal{O}\left(\eta^3\right) \quad \forall i, j = 1, \ldots, d$;

3. $\mathbb{E}\prod_{j=1}^s \Delta_{(i_j)} = \mathcal{O}\left(\eta^3\right)$ *for all $s \geq 3, i_j = 1, \ldots, d$.*

*All functions above are evaluated at $x$.*

596

**Theorem C.4** (Theorem 2 and Lemma 5, Mil'shtein (1986)). *Let Assumption C.2 hold and let us define $\bar{\Delta} = x_1 - x$ to be the increment in the discrete-time algorithm, and indicate the $i$-th component of $\bar{\Delta}$ with $\bar{\Delta}_i$. If in addition there exists $K_1, K_2, K_3, K_4 \in G$ so that*

1. $\left|\mathbb{E}\Delta_i - \mathbb{E}\bar{\Delta}_i\right| \leq K_1(x)\eta^2, \quad \forall i = 1, \ldots, d$;

2. $\left|\mathbb{E}\Delta_i\Delta_j - \mathbb{E}\bar{\Delta}_i\bar{\Delta}_j\right| \leq K_2(x)\eta^2, \quad \forall i, j = 1, \ldots, d$;

3. $\left|\mathbb{E}\prod_{j=1}^s \Delta_{i_j} - \mathbb{E}\prod_{j=1}^s \bar{\Delta}_{i_j}\right| \leq K_3(x)\eta^2, \quad \forall s \geq 3, \quad \forall i_j \in \{1, \ldots, d\}$;

4. $\mathbb{E}\prod_{j=1}^3 \left|\bar{\Delta}_{i_j}\right| \leq K_4(x)\eta^2, \quad \forall i_j \in \{1, \ldots, d\}$.

*Then, there exists a constant $C$ so that for all $k = 0, 1, \ldots, N$ we have*

$$\left|\mathbb{E}g\left(X_{k\eta}\right) - \mathbb{E}g\left(x_k\right)\right| \leq C\eta.$$

597

## C.1 Limitations

Modeling of discrete-time algorithms using SDEs relies on Assumption C.2. As noted by Li et al. (2021), the approximation can fail when the stepsize $\eta$ is large or if certain conditions on $\nabla f$ and the noise covariance matrix are not met. Although these issues can be addressed by increasing the order of the weak approximation, we believe that the primary purpose of SDEs is to serve as simplification tools that enhance our intuition: We would not benefit significantly from added complexity.

## C.2 Formal derivation - SignSGD

In this subsection, we provide the first formal derivation of an SDE model for SignSGD. Let us consider the stochastic process $X_t \in \mathbb{R}^d$ defined as the solution of

$$dX_t = -(1 - 2\mathbb{P}(\nabla f_\gamma(X_t) < 0))dt + \sqrt{\eta}\sqrt{\bar{\Sigma}(X_t)}dW_t, \tag{16}$$

where

$$\bar{\Sigma}(x) = \mathbb{E}[\xi_\gamma(x)\xi_\gamma(x)^\top], \tag{17}$$

and $\xi_\gamma(x) := \text{sign}(\nabla f_\gamma(x)) - 1 + 2\mathbb{P}(\nabla f_\gamma(x) < 0)$ the noise in the sample sign $(\nabla f_\gamma(x))$. The following theorem guarantees that such a process is a 1-order SDE of the discrete-time algorithm of SignSGD

$$x_{k+1} = x_k - \eta\text{sign}\left(f_{\gamma_k}(x_k)\right), \tag{18}$$

with $x_0 \in \mathbb{R}^d, \eta \in \mathbb{R}^{>0}$ is the step size, the mini-batches $\{\gamma_k\}$ are modelled as i.i.d. random variables uniformly distributed on $\{1, \cdots, N\}$, and of size $B \geq 1$.

**Theorem C.5** (Stochastic modified equations). *Let $0 < \eta < 1, T > 0$ and set $N = \lfloor T/\eta \rfloor$. Let $x_k \in \mathbb{R}^d, 0 \leq k \leq N$ denote a sequence of SignSGD iterations defined by Eq. (18). Consider the stochastic process $X_t$ defined in Eq. (16) and fix some test function $g \in G$ and suppose that $g$ and its partial derivatives up to order 6 belong to $G$.*
*Then, under Assumption C.2, there exists a constant $C > 0$ independent of $\eta$ such that for all $k = 0, 1, \ldots, N$, we have*

$$|\mathbb{E}g\left(X_{k\eta}\right) - \mathbb{E}g\left(x_k\right)| \leq C\eta.$$

*That is, the SDE (16) is an order 1 weak approximation of the SignSGD iterations (18).*

613

**Lemma C.6.** *Under the assumptions of Theorem C.5, let $0 < \eta < 1$ and consider $x_k, k \geq 0$ satisfying the SignSGD iterations*

$$x_{k+1} = x_k - \eta sign\left(\nabla f_{\gamma_k}(x_k)\right)$$

*with $x_0 \in \mathbb{R}^d$. From the definition the one-step difference $\bar{\Delta} = x_1 - x$, then we have*

  1. $\mathbb{E}\bar{\Delta}_i = -\left(1 - 2\mathbb{P}\left(\partial_i f_\gamma < 0\right)\right)\eta \quad \forall i = 1, \ldots, d$;
  2. $\mathbb{E}\bar{\Delta}_i\bar{\Delta}_j = \left(\left(1 - 2\mathbb{P}\left(\partial_i f_\gamma < 0\right)\right)\left(1 - 2\mathbb{P}\left(\partial_j f_\gamma < 0\right)\right) + \bar{\Sigma}_{(ij)}\right)\eta^2 \quad \forall i, j = 1, \ldots, d$;
  3. $\mathbb{E}\prod_{j=1}^s \bar{\Delta}_{i_j} = \mathcal{O}\left(\eta^3\right) \quad \forall s \geq 3, \quad i_j \in \{1, \ldots, d\}$.

*All the functions above are evaluated at $x$.*

614

615 *Proof of Lemma C.6.* First of all, we have that by definition

$$\mathbb{E}\left[x_1^i - x^i\right] = -\eta\mathbb{E}\left[\text{sign}\left(\partial_i f_\gamma(x) < 0\right)\right], \tag{19}$$

616 which implies

$$\mathbb{E}\bar{\Delta}_i = -\left(1 - 2\mathbb{P}\left(\partial_i f_\gamma(x) < 0\right)\right)\eta \quad \forall i = 1, \ldots, d. \tag{20}$$

617 Second, we have that by definition

$$\mathbb{E}\left[\left(x_1 - x\right)\left(x_1 - x\right)^\top\right] = \mathbb{E}\Big[\left(\text{sign}\left(\partial_i f_\gamma(x) < 0\right) - 1 + 2\mathbb{P}\left(\partial_i f_\gamma(x) < 0\right)\right) \tag{21}$$

$$\left(\text{sign}\left(\partial_i f_\gamma(x) < 0\right) - 1 + 2\mathbb{P}\left(\partial_i f_\gamma(x) < 0\right)\right)^\top\Big]\eta^2, \tag{22}$$

618 which implies that

$$\mathbb{E}\bar{\Delta}_i\bar{\Delta}_j = \left(1 - 2\mathbb{P}\left(\partial_i f_\gamma < 0\right)\right)\left(1 - 2\mathbb{P}\left(\partial_j f_\gamma < 0\right)\right)\eta^2 + \bar{\Sigma}_{(ij)}\eta^2 \quad \forall i, j = 1, \ldots, d. \tag{23}$$

619 Finally, by definition

$$\mathbb{E}\prod_{j=1}^s \bar{\Delta}_{i_j} = \mathcal{O}\left(\eta^3\right) \quad \forall s \geq 3, \quad i_j \in \{1, \ldots, d\}, \tag{24}$$

620 which concludes our proof. $\qquad\square$

621 *Proof of Theorem C.5.* To prove this result, all we need to do is check the conditions in Theorem C.4.
622 As we apply Lemma C.3, we make the following choices:

623     • $b(x) = -(1 - 2\mathbb{P}\left(\nabla f_\gamma(x) < 0\right))$;

624     • $\sigma(x) = \sqrt{\bar{\Sigma}(x)}$.

First of all, we notice that $\forall i = 1, \ldots, d$, it holds that

- $\mathbb{E}\bar{\Delta}_i \overset{\text{1. Lemma C.6}}{=} - \left(1 - 2\mathbb{P}\left(\partial_i f_\gamma(x) < 0\right)\right)\eta;$

- $\mathbb{E}\Delta_i \overset{\text{1. Lemma C.3}}{=} - \left(1 - 2\mathbb{P}\left(\partial_i f_\gamma(x) < 0\right)\right)\eta + \mathcal{O}\left(\eta^2\right).$

Therefore, we have that for some $K_1(x) \in G$,

$$\left|\mathbb{E}\Delta_i - \mathbb{E}\bar{\Delta}_i\right| \leq K_1(x)\eta^2, \quad \forall i = 1, \ldots, d. \tag{25}$$

Additionally, we notice that $\forall i, j = 1, \ldots, d$, it holds that

- $\mathbb{E}\bar{\Delta}_i\bar{\Delta}_j \overset{\text{2. Lemma C.6}}{=} \left(1 - 2\mathbb{P}\left(\partial_i f_\gamma(x) < 0\right)\right)\left(1 - 2\mathbb{P}\left(\partial_j f_\gamma(x) < 0\right)\right)\eta^2 + \bar{\Sigma}_{(ij)}(x)\eta^2;$

- $\mathbb{E}\Delta_i\Delta_j \overset{\text{2. Lemma C.3}}{=} \left(\left(1 - 2\mathbb{P}\left(\partial_i f_\gamma(x) < 0\right)\right)\left(1 - 2\mathbb{P}\left(\partial_j f_\gamma(x) < 0\right)\right) + \bar{\Sigma}_{(ij)}(x)\right)\eta^2 +$
  $\mathcal{O}\left(\eta^3\right).$

Therefore, we have that for some $K_2(x) \in G$,

$$\left|\mathbb{E}\Delta_i\Delta_j - \mathbb{E}\bar{\Delta}_i\bar{\Delta}_j\right| \leq K_2(x)\eta^2, \quad \forall i, j = 1, \ldots, d. \tag{26}$$

Additionally, we notice that $\forall s \geq 3, \forall i_j \in \{1, \ldots, d\}$, it holds that

- $\mathbb{E}\prod_{j=1}^{s}\bar{\Delta}_{i_j} \overset{\text{3. Lemma C.6}}{=} \mathcal{O}\left(\eta^3\right);$

- $\mathbb{E}\prod_{j=1}^{s}\Delta_{i_j} \overset{\text{3. Lemma C.3}}{=} \mathcal{O}\left(\eta^3\right).$

Therefore, we have that for some $K_3(x) \in G$,

$$\left|\mathbb{E}\prod_{j=1}^{s}\Delta_{i_j} - \mathbb{E}\prod_{j=1}^{s}\bar{\Delta}_{i_j}\right| \leq K_3(x)\eta^2. \tag{27}$$

Additionally, for some $K_4(x) \in G$, $\forall i_j \in \{1, \ldots, d\}$,

$$\mathbb{E}\prod_{j=1}^{3}\left|\bar{\Delta}_{(i_j)}\right| \overset{\text{3. Lemma C.6}}{\leq} K_4(x)\eta^2. \tag{28}$$

To conclude, Eq. (25), Eq. (26), Eq. (27), and Eq. (28) allow us to conclude the proof. $\square$

**Corollary C.7.** *Let us take the same assumptions of Theorem C.5, and that the stochastic gradient is $\nabla f_\gamma(x) = \nabla f(x) + U$ such that $U \sim \mathcal{N}(0, \Sigma)$ that does not depend on $x$. Then, the following SDE provides a 1 weak approximation of the discrete update of SignSGD*

$$dX_t = -Erf\left(\frac{\Sigma^{-\frac{1}{2}}\nabla f(X_t)}{\sqrt{2}}\right)dt + \sqrt{\eta}\sqrt{I_d - \text{diag}\left(Erf\left(\frac{\Sigma^{-\frac{1}{2}}\nabla f(X_t)}{\sqrt{2}}\right)\right)^2}dW_t, \tag{29}$$

*where the error function $Erf(x)$ and the square are applied component-wise, and $\Sigma = \text{diag}\left(\sigma_1^2, \cdots, \sigma_d^2\right)$.*

*Proof of Corollary C.7.* First of all, we observe that

$$1 - 2\mathbb{P}\left(\nabla f_\gamma(x) < 0\right) = 1 - 2\mathbb{P}\left(\nabla f(x) + \Sigma^{\frac{1}{2}}U < 0\right) = 1 - 2\Phi\left(-\Sigma^{-\frac{1}{2}}\nabla f(x)\right), \tag{30}$$

where $\Phi$ is the cumulative distribution function of the standardized normal distribution. Remembering that

$$\Phi(x) = \frac{1}{2} \left( 1 + \mathrm{Erf} \left( \frac{x}{\sqrt{2}} \right) \right), \tag{31}$$

we have that

$$1 - 2\mathbb{P}\left( \nabla f_\gamma(x) < 0 \right) = 1 - 2\frac{1}{2} \left( 1 + \mathrm{Erf} \left( -\frac{\Sigma^{-\frac{1}{2}} \nabla f(x)}{\sqrt{2}} \right) \right) = \mathrm{Erf} \left( \frac{\Sigma^{-\frac{1}{2}} \nabla f(x)}{\sqrt{2}} \right). \tag{32}$$

Similarly, one can prove that $\bar\Sigma$ defined in (17) becomes

$$\bar\Sigma = I_d - \mathrm{diag} \left( \mathrm{Erf} \left( \frac{\Sigma^{-\frac{1}{2}} \nabla f(X_t)}{\sqrt{2}} \right) \right)^2. \tag{33}$$

$\square$

> **Corollary C.8.** *Let us take the same assumptions of Theorem C.5, and that the stochastic gradient is $\nabla f_\gamma(x) = \nabla f(x) + \sqrt{\Sigma} U$ such that $U \sim t_\nu(0, I_d)$ that does not depend on $x$ and $\nu$ is a positive integer number. Then, the following SDE provides a 1 weak approximation of the discrete update of SignSGD*
>
> $$dX_t = -2\Xi \left( \Sigma^{-\frac{1}{2}} \nabla f(X_t) \right) dt + \sqrt{\eta} \sqrt{I_d - 4 \mathrm{diag} \left( \Xi \left( \Sigma^{-\frac{1}{2}} \nabla f(X_t) \right) \right)^2} dW_t, \tag{34}$$
>
> *where $\Xi(x)$ is defined as*
>
> $$\Xi(x) := x \frac{\Gamma \left( \frac{\nu+1}{2} \right)}{\sqrt{\pi\nu} \Gamma \left( \frac{\nu}{2} \right)} {}_2F_1 \left( \frac{1}{2}, \frac{\nu+1}{2}; \frac{3}{2}; -\frac{x^2}{\nu} \right), \tag{35}$$
>
> *and ${}_2F_1 (a, b; c; x)$ is the hypergeometric function. Above, function $\Xi(x)$ and the square are applied component-wise, and $\Sigma = \mathrm{diag} \left( \sigma_1^2, \cdots, \sigma_d^2 \right)$.*

*Proof of Corollary C.8.* First of all, we observe that

$$1 - 2\mathbb{P}\left( \nabla f_\gamma(x) < 0 \right) = 1 - 2\mathbb{P}\left( \nabla f(x) + \Sigma^{\frac{1}{2}} U < 0 \right) = 1 - 2F_\nu \left( -\Sigma^{-\frac{1}{2}} \nabla f(x) \right), \tag{36}$$

where $F_\nu (x)$ is the cumulative function of a $t$ distribution with $\nu$ degrees of freedom. Remembering that

$$F_\nu (x) = \frac{1}{2} + \Xi(x), \tag{37}$$

we have that

$$1 - 2\mathbb{P}\left( \nabla f_\gamma(x) < 0 \right) = 1 - 2 \left( \frac{1}{2} + \Xi(x) \right) = -2\Xi(x). \tag{38}$$

Similarly, one can prove that $\bar\Sigma$ defined in (17) becomes

$$\bar\Sigma = I_d - 4 \mathrm{diag} \left( \Xi \left( \Sigma^{-\frac{1}{2}} \nabla f(X_t) \right) \right)^2. \tag{39}$$

$\square$

**Lemma C.9.** *Under the assumptions of Corollary C.7 and signal-to-noise ratio $Y_t := \frac{\Sigma^{-\frac{1}{2}} \nabla f(X_t)}{\sqrt{2}}$,*

    *1. **Phase 1:** If $|Y_t| > \frac{3}{2}$, the SDE coincides with the ODE of SignGD:*

$$dX_t = -sign(\nabla f(X_t))dt; \tag{40}$$

    *2. **Phase 2:** If $1 < |Y_t| < \frac{3}{2}$:*

        *(a) $mY_t + \mathbf{q}^- \leq \frac{d\mathbb{E}[X_t]}{dt} \leq mY_t + \mathbf{q}^+$;*

658  (b) $\mathbb{P}\left[\|X_t - \mathbb{E}\left[X_t\right]\|_2^2 > a\right] \leq \frac{\eta}{a}\left(d - \|mY_t + \mathbf{q}^-\|_2^2\right);$

659  3. **Phase 3:** If $|Y_t| < 1$, the SDE is

$$dX_t = -\sqrt{\frac{2}{\pi}}\Sigma^{-\frac{1}{2}}\nabla f(X_t)dt + \sqrt{\eta}\sqrt{I_d - \frac{2}{\pi}\operatorname{diag}\left(\Sigma^{-\frac{1}{2}}\nabla f(X_t)\right)^2}dW_t. \quad (41)$$

660  *Proof of Lemma C.9.* Exploiting the regularity of the Erf function, we approximate the SDE in (29)
661  in three different regions:

662  1. **Phase 1:** If $|x| > \frac{3}{2}$, $\operatorname{Erf}(x) \sim \operatorname{sign}(x)$. Therefore, if $\left|\frac{\Sigma^{-\frac{1}{2}}\nabla f(X_t)}{\sqrt{2}}\right| > \frac{3}{2}$,

663  (a) $\operatorname{Erf}\left(\frac{\Sigma^{-\frac{1}{2}}\nabla f(X_t)}{\sqrt{2}}\right) \sim \operatorname{sign}\left(\frac{\Sigma^{-\frac{1}{2}}\nabla f(X_t)}{\sqrt{2}}\right) = \operatorname{sign}\left(\nabla f(X_t)\right);$

664  (b) $\operatorname{Erf}\left(\frac{\Sigma^{-\frac{1}{2}}\nabla f(X_t)}{\sqrt{2}}\right)^2 \sim \operatorname{sign}\left(\frac{\Sigma^{-\frac{1}{2}}\nabla f(X_t)}{\sqrt{2}}\right)^2 = (1, \ldots, 1).$

665  Therefore,

$$dX_t = -\operatorname{Erf}\left(\frac{\Sigma^{-\frac{1}{2}}\nabla f(X_t)}{\sqrt{2}}\right)dt + \sqrt{\eta}\sqrt{I_d - \operatorname{diag}\left(\operatorname{Erf}\left(\frac{\Sigma^{-\frac{1}{2}}\nabla f(X_t)}{\sqrt{2}}\right)\right)^2}dW_t$$
$$\sim -\operatorname{sign}(\nabla f(X_t)); \quad (42)$$

666  2. **Phase 2:** Let $m$ and $q_1$ are the slope and intercept of the line secant to the graph of $\operatorname{Erf}(x)$
667  between the points $(1, \operatorname{Erf}(1))$ and $\left(\frac{3}{2}, \operatorname{Erf}\left(\frac{3}{2}\right)\right)$, while $q_2$ is the intercept of the line tangent
668  to the graph of $\operatorname{Erf}(x)$ and slope $m$. If $1 < x < \frac{3}{2}$, we have that

$$mx + q_1 < \operatorname{Erf}(x) < mx + q_2. \quad (43)$$

669  Analogously, if $-\frac{3}{2} < x < -1$

$$mx - q_2 < \operatorname{Erf}(x) < mx - q_1. \quad (44)$$

670  Therefore, we have that if $1 < \left|\frac{\Sigma^{-\frac{1}{2}}\nabla f(X_t)}{\sqrt{2}}\right| < \frac{3}{2}$, then

(a)

$$\frac{m}{\sqrt{2}}\Sigma^{-\frac{1}{2}}\nabla f(X_t) + \mathbf{q}^- < \operatorname{Erf}\left(\frac{\Sigma^{-\frac{1}{2}}\nabla f(X_t)}{\sqrt{2}}\right) < \frac{m}{\sqrt{2}}\Sigma^{-\frac{1}{2}}\nabla f(X_t) + \mathbf{q}^+, \quad (45)$$

671  where

$$(\mathbf{q}^+)_i := \begin{cases} q_2 & \text{if } \partial_i f(x) > 0 \\ -q_1 & \text{if } \partial_i f(x) < 0, \end{cases} \quad (46)$$

672  and

$$(\mathbf{q}^-)_i := \begin{cases} q_1 & \text{if } \partial_i f(x) > 0 \\ -q_2 & \text{if } \partial_i f(x) < 0, \end{cases} \quad (47)$$

673  Therefore,

$$\frac{m}{\sqrt{2}}\Sigma^{-\frac{1}{2}}\nabla f(X_t) + \mathbf{q}^- \leq \frac{d\mathbb{E}\left[X_t\right]}{dt} \leq \frac{m}{\sqrt{2}}\Sigma^{-\frac{1}{2}}\nabla f(X_t) + \mathbf{q}^+; \quad (48)$$

674  (b) Similar to the above,

$$\left(\frac{m}{\sqrt{2}}\Sigma^{-\frac{1}{2}}\nabla f(X_t) + \mathbf{q}^-\right)^2 \leq \operatorname{Erf}\left(\frac{\Sigma^{-\frac{1}{2}}\nabla f(X_t)}{\sqrt{2}}\right)^2 \leq \left(\frac{m}{\sqrt{2}}\Sigma^{-\frac{1}{2}}\nabla f(X_t) + \mathbf{q}^+\right)^2.$$

Therefore,

$$\mathbb{P}\left[\|X_t - \mathbb{E}[X_t]\|_2^2 > a\right] \leq \mathbb{P}\left[\exists i \text{ s.t. } |X_t^i - \mathbb{E}[X_t^i]|^2 > a\right] \tag{49}$$

$$\leq \sum_i \mathbb{P}\left[|X_t^i - \mathbb{E}[X_t^i]| > \sqrt{a}\right]$$

$$\leq \frac{\eta}{a} \sum_i \left(1 - \text{Erf}\left(\frac{\Sigma_i^{-\frac{1}{2}} \partial_i f(X_t)}{\sqrt{2}}\right)^2\right) \tag{50}$$

$$< \frac{\eta}{a}\left(d - \|\frac{m}{\sqrt{2}}\Sigma^{-\frac{1}{2}}\nabla f(X_t) + \mathbf{q}^-\|_2^2\right). \tag{51}$$

3. **Phase 3:** If $|x| < 1$, $\text{Erf}(x) \sim \frac{2}{\sqrt{\pi}}$. Therefore, if $\left|\frac{\Sigma^{-\frac{1}{2}}\nabla f(X_t)}{\sqrt{2}}\right| < 1$,

    (a) $\text{Erf}\left(\frac{\Sigma^{-\frac{1}{2}}\nabla f(X_t)}{\sqrt{2}}\right) \sim \sqrt{\frac{2}{\pi}}\Sigma^{-\frac{1}{2}}\nabla f(X_t)$;

    (b) $\left(\text{Erf}\left(\frac{\Sigma^{-\frac{1}{2}}\nabla f(X_t)}{\sqrt{2}}\right)\right)^2 \sim \frac{2}{\pi}\left(\Sigma^{-\frac{1}{2}}\nabla f(X_t)\right)^2$.

Therefore,

$$dX_t = -\text{Erf}\left(\frac{\Sigma^{-\frac{1}{2}}\nabla f(X_t)}{\sqrt{2}}\right)dt + \sqrt{\eta}\sqrt{I_d - \text{diag}\left(\text{Erf}\left(\frac{\Sigma^{-\frac{1}{2}}\nabla f(X_t)}{\sqrt{2}}\right)\right)^2}dW_t$$

$$\sim -\sqrt{\frac{2}{\pi}}\Sigma^{-\frac{1}{2}}\nabla f(X_t)dt + \sqrt{\eta}\sqrt{I_d - \frac{2}{\pi}\text{diag}\left(\Sigma^{-\frac{1}{2}}\nabla f(X_t)\right)^2}dW_t. \tag{52}$$

$\square$

**Lemma C.10** (Dynamics of Expected Loss). *Let $f$ be $\mu$-strongly convex, $Tr(\nabla^2 f(x)) \leq \mathcal{L}_\tau$, and $S_t := f(X_t) - f(X_*)$. Then, during*

    *1. Phase 1, the dynamics will stop before $t_* = 2\sqrt{\frac{S_0}{\mu}}$ because $S_t \leq \frac{1}{4}\left(\sqrt{\mu}t - 2\sqrt{S_0}\right)^2$;*

    *2. Phase 2 with $\Delta := \left(\frac{m}{\sqrt{2}\sigma_{max}} + \frac{\eta\mu m^2}{4\sigma_{max}^2}\right)$: $\mathbb{E}[S_t] \leq S_0 e^{-2\mu\Delta t} + \frac{\eta}{2}\frac{(\mathcal{L}_\tau - \mu d\hat{q}^2)}{2\mu\Delta}\left(1 - e^{-2\mu\Delta t}\right)$;*

    *3. Phase 3 with $\Delta := \left(\sqrt{\frac{2}{\pi}}\frac{1}{\sigma_{max}} + \frac{\eta}{\pi}\frac{\mu}{\sigma_{max}^2}\right)$: $\mathbb{E}[S_t] \leq S_0 e^{-2\mu\Delta t} + \frac{\eta}{2}\frac{\mathcal{L}_\tau}{2\mu\Delta}\left(1 - e^{-2\mu\Delta t}\right)$.*

*Proof of Lemma C.10.* We prove each point by leveraging the shape of the law of $X_t$ derived in Lemma C.9:

    1. **Phase 1:**

$$d(f(X_t) - f(X_*)) = -\nabla f(X_t)\text{sign}(\nabla f(X_t)) = -\|\nabla f(X_t)\|_1 \leq -\|\nabla f(X_t)\|_2 \tag{53}$$

    Since $f$ is $\mu - PL$, we have that $-\|\nabla f(X_t)\|_2^2 < -2\mu(f(X_t) - f(X_*))$, which implies that

$$f(X_t) - f(X_*) \leq \frac{1}{4}\left(\sqrt{\mu}t - 2\sqrt{f(X_0) - f(X_*)}\right)^2, \tag{54}$$

    meaning that the dynamics will stop before $t_* = 2\sqrt{\frac{f(X_0) - f(X_*)}{\mu}}$;

    2. **Phase 2:** By applying the Itô Lemma to $f(X_t) - f(X_*)$ and that

$$\frac{m}{\sqrt{2}}\Sigma^{-\frac{1}{2}}\nabla f(X_t) + \mathbf{q}^- < \text{Erf}\left(\frac{\Sigma^{-\frac{1}{2}}\nabla f(X_t)}{\sqrt{2}}\right) < \frac{m}{\sqrt{2}}\Sigma^{-\frac{1}{2}}\nabla f(X_t) + \mathbf{q}^+, \tag{55}$$

we have that if $\hat{q} := \max(q_1, q_2)$,

$$d(f(X_t) - f(X_*)) \leq - \left( \frac{m}{\sqrt{2}} \Sigma^{-\frac{1}{2}} \nabla f(X_t) + \mathbf{q}^- \right)^\top \nabla f(X_t) dt + \mathcal{O}(\text{Noise}) \tag{56}$$

$$+ \frac{\eta}{2} \text{Tr} \left[ \nabla^2 f(X_t) \left( I_d - \text{diag} \left( \frac{m}{\sqrt{2}} \Sigma^{-\frac{1}{2}} \nabla f(X_t) + \mathbf{q}^- \right)^2 \right) \right] \tag{57}$$

$$\leq - \frac{m}{\sqrt{2}} \frac{1}{\sigma_{\max}} \|\nabla f(X_t)\|_2^2 dt - \hat{q} \|\nabla f(X_t)\|_1 dt + \frac{\eta \mathcal{L}_\tau}{2} dt \tag{58}$$

$$- \frac{\eta \mu}{2} \| \frac{m}{\sqrt{2}} \Sigma^{-\frac{1}{2}} \nabla f(X_t) + \mathbf{q}^- \|_2^2 dt + \mathcal{O}(\text{Noise}) \tag{59}$$

$$\leq - \frac{m}{\sqrt{2}} \frac{1}{\sigma_{\max}} \|\nabla f(X_t)\|_2^2 dt - \hat{q} \|\nabla f(X_t)\|_1 dt + \frac{\eta \mathcal{L}_\tau}{2} dt \tag{60}$$

$$- \frac{\eta \mu m^2}{4 \sigma_{\max}^2} \|\nabla f(X_t)\|_2^2 dt - \frac{\eta \mu d \hat{q}^2}{2} dt - \frac{\sqrt{2} m \hat{q}}{\sigma_{\max}} \|\nabla f(X_t)\|_1 dt \tag{61}$$

$$+ \mathcal{O}(\text{Noise}) \tag{62}$$

$$\leq - 2\mu \left( \frac{m}{\sqrt{2}\sigma_{\max}} + \frac{\eta \mu m^2}{4\sigma_{\max}^2} \right) (f(X_t) - f(X_*)) dt \tag{63}$$

$$+ \frac{\eta}{2} \left( \mathcal{L}_\tau - \mu d \hat{q}^2 \right) dt + \mathcal{O}(\text{Noise}), \tag{64}$$

which implies that if $k := 2\mu \left( \frac{m}{\sqrt{2}\sigma_{\max}} + \frac{\eta \mu m^2}{4\sigma_{\max}^2} \right)$,

$$\mathbb{E}[f(X_t) - f(X_*)] \leq (f(X_0) - f(X_*)))e^{-kt} + \frac{\eta \left( \mathcal{L}_\tau - \mu d \hat{q}^2 \right)}{2k} \left( 1 - e^{-kt} \right). \tag{65}$$

3. **Phase 3:** By applying the Itô Lemma to $f(X_t) - f(X_*)$, we have that:

$$d(f(X_t) - f(X_*)) = - \sqrt{\frac{2}{\pi}} \nabla f(X_t)^\top \Sigma^{-\frac{1}{2}} \nabla f(X_t) dt + \mathcal{O}(\text{Noise}) \tag{66}$$

$$+ \frac{\eta}{2} \text{Tr} \left( \left( I_d - \frac{2}{\pi} \text{diag} \left( \Sigma^{-\frac{1}{2}} \nabla f(X_t) \right)^2 \right) \nabla^2 f(X_t) \right) dt \tag{67}$$

$$\leq - \sqrt{\frac{2}{\pi}} \frac{1}{\sigma_{\max}} \|\nabla f(X_t)\|_2^2 dt + \mathcal{O}(\text{Noise}) \tag{68}$$

$$+ \frac{\eta}{2} \text{Tr} \left( \nabla^2 f(X_t) \right) dt - \frac{\eta}{\pi} \frac{\mu}{\sigma_{\max}^2} \|\nabla f(X_t)\|_2^2 dt \tag{69}$$

$$\leq - \left( \sqrt{\frac{2}{\pi}} \frac{1}{\sigma_{\max}} + \frac{\eta}{\pi} \frac{\mu}{\sigma_{\max}^2} \right) \|\nabla f(X_t)\|_2^2 dt \tag{70}$$

$$+ \frac{\eta}{2} Tr(\nabla^2 f(X_t)) dt + \mathcal{O}(\text{Noise}) \tag{71}$$

Since $f$ is $\mu$-Strongly Convex, $f$ is also $\mu$-PL. Therefore, we have

$$d(f(X_t) - f(X_*)) \leq - 2\mu \left( \sqrt{\frac{2}{\pi}} \frac{1}{\sigma_{\max}} + \frac{\eta}{\pi} \frac{\mu}{\sigma_{\max}^2} \right) (f(X_t) - f(X_*)) dt \tag{72}$$

$$+ \frac{\eta}{2} Tr(\nabla^2 f(X_t)) dt + \mathcal{O}(\text{Noise}). \tag{73}$$

Therefore,

$$d\mathbb{E}[f(X_t) - f(X_*)] \leq -2\mu \left( \sqrt{\frac{2}{\pi}} \frac{1}{\sigma_{\max}} + \frac{\eta}{\pi} \frac{\mu}{\sigma_{\max}^2} \right) (\mathbb{E}[f(X_t) - f(X_*)]) dt + \frac{\eta}{2} \mathcal{L}_\tau dt, \tag{74}$$

which implies that if $k := 2\mu\left(\sqrt{\frac{2}{\pi}}\frac{1}{\sigma_{\max}} + \frac{\eta}{\pi}\frac{\mu}{\sigma_{\max}^2}\right)$,

$$\mathbb{E}[f(X_t) - f(X_*)] \le (f(X_0) - f(X_*)))e^{-kt} + \frac{\eta\mathcal{L}_\tau}{2k}\left(1 - e^{-kt}\right). \tag{75}$$

 $\qquad\qquad\qquad\qquad\qquad\qquad\qquad\qquad\qquad\qquad\qquad\qquad\qquad\qquad\qquad\qquad\square$

 **Lemma C.11.** *Under the assumptions of Lemma 3.5, for any step size scheduler $\eta_t$ such that*

$$\int_0^\infty \eta_s ds = \infty \text{ and } \lim_{t\to\infty}\eta_t = 0 \implies \mathbb{E}[f(X_t) - f(X_*)] \stackrel{t\to\infty}{\to} 0. \tag{76}$$

 *Proof of Lemma C.11.* For any scheduler $\eta_k$ used in

$$x_{k+1} = x_k - \eta\eta_k \text{sign}\left(f_{\gamma_k}(x_k)\right), \tag{77}$$

 the SDE of Phase 3 is

$$dX_t = -\sqrt{\frac{2}{\pi}}\Sigma^{-\frac{1}{2}}\nabla f(X_t)\eta_t dt + \sqrt{\eta}\eta_t\sqrt{I_d - \frac{2}{\pi}\text{diag}\left(\Sigma^{-\frac{1}{2}}\nabla f(X_t)\right)^2}dW_t. \tag{78}$$

 Therefore, analogously to the calculations in Lemma C.10, we have that

$$\mathbb{E}[f(X_t) - f(X_*)] \le \frac{f(X_0) - f(X_*) + \frac{\eta\mathcal{L}_\tau}{2}\int_0^t e^{2\mu\int_0^s\left(\sqrt{\frac{2}{\pi}}\frac{1}{\sigma_{\max}}\eta_l + \frac{\eta}{\pi}\frac{\mu}{\sigma_{\max}^2}\eta_l^2\right)dl}\eta_s^2 ds}{e^{2\mu\int_0^t\left(\sqrt{\frac{2}{\pi}}\frac{1}{\sigma_{\max}}\eta_s + \frac{\eta}{\pi}\frac{\mu}{\sigma_{\max}^2}\eta_s^2\right)ds}}. \tag{79}$$

 Therefore, using l'Hôpital's rule we have that

$$\int_0^\infty \eta_s ds = \infty \text{ and } \lim_{t\to\infty}\eta_t = 0 \implies \mathbb{E}[f(X_t) - f(X_*)] \stackrel{t\to\infty}{\to} 0. \tag{80}$$

 $\qquad\qquad\qquad\qquad\qquad\qquad\qquad\qquad\qquad\qquad\qquad\qquad\qquad\qquad\qquad\qquad\square$

 **Lemma C.12.** *Let $H = \text{diag}(\lambda_1,\ldots,\lambda_d)$ and $M_t := e^{-2\left(\sqrt{\frac{2}{\pi}}\Sigma^{-\frac{1}{2}}H + \frac{\eta}{\pi}\Sigma^{-\frac{1}{2}}H^2\right)t}$. Then,*

 *1. $\mathbb{E}\left[X_t\right] = e^{-\sqrt{\frac{2}{\pi}}\Sigma^{-\frac{1}{2}}Ht}X_0$;*

 *2. $Var\left[X_t\right] = \left(M_t - e^{-2\sqrt{\frac{2}{\pi}}\Sigma^{-\frac{1}{2}}Ht}\right)X_0^2 + \frac{\eta}{2}\left(\sqrt{\frac{2}{\pi}}I_d + \frac{\eta}{\pi}H\right)^{-1}H^{-1}\Sigma^{\frac{1}{2}}\left(I_d - M_t\right)$.*

 *Proof of Lemma C.12.* The proof is banal: The expected value derivation leverages the martingale
 property of the Brownian motion while that of the variance uses the Ito Isomerty. $\qquad\square$

 **Lemma C.13.** *Let $H = \text{diag}(\lambda_1,\ldots,\lambda_d)$. Then, $\mathbb{E}\left[\frac{X_t^\top H X_t}{2}\right]$ is equal to*

$$\sum_{i=1}^d \frac{\lambda_i(X_0^i)^2}{2}e^{-2\lambda_i\left(\sqrt{\frac{2}{\pi}}\frac{1}{\sigma_i} + \frac{\lambda_i\eta}{\pi\sigma_i^2}\right)t} + \frac{\eta}{4\left(\sqrt{\frac{2}{\pi}}\frac{1}{\sigma_i} + \frac{\lambda_i\eta}{\pi\sigma_i^2}\right)}\left(1 - e^{-2\lambda_i\left(\sqrt{\frac{2}{\pi}}\frac{1}{\sigma_i} + \frac{\lambda_i\eta}{\pi\sigma_i^2}\right)t}\right). \tag{81}$$

 *Proof of Lemma C.13.* Since the matrix $H$ is diagonal, we focus on a single component. We apply
 the Ito Lemma to $\frac{\lambda_i(X_t^i)^2}{2}$:

$$d\left(\frac{\lambda_i(X_t^i)^2}{2}\right) = -2\sqrt{\frac{2}{\pi}}\frac{\lambda_i}{\sigma_i}\frac{\lambda_i(X_t^i)^2}{2}dt + \frac{\eta\lambda_i}{2}dt - \frac{2\lambda_i^2\eta}{\pi\sigma_i^2}\frac{\lambda_i(X_t^i)^2}{2} + \mathcal{O}(\text{Noise}), \tag{82}$$

 which implies that

$$\mathbb{E}\left[\frac{\lambda_i(X_t^i)^2}{2}\right] = \frac{\lambda_i(X_0^i)^2}{2}e^{-2\left(\sqrt{\frac{2}{\pi}}\frac{\lambda_i}{\sigma_i} + \frac{\lambda_i^2\eta}{\pi\sigma_i^2}\right)t} + \frac{\eta}{4\left(\sqrt{\frac{2}{\pi}}\frac{1}{\sigma_i} + \frac{\lambda_i\eta}{\pi\sigma_i^2}\right)}\left(1 - e^{-2\left(\sqrt{\frac{2}{\pi}}\frac{\lambda_i}{\sigma_i} + \frac{\lambda_i^2\eta}{\pi\sigma_i^2}\right)t}\right).$$

$$\tag{83}$$

Therefore,

$$\mathbb{E}\left[\frac{X_t^\top H X_t}{2}\right] = \sum_{i=1}^d \frac{\lambda_i (X_0^i)^2}{2} e^{-2\lambda_i\left(\sqrt{\frac{2}{\pi}}\frac{1}{\sigma_i} + \frac{\lambda_i \eta}{\pi \sigma_i^2}\right)t} + \frac{\eta}{4\left(\sqrt{\frac{2}{\pi}}\frac{1}{\sigma_i} + \frac{\lambda_i \eta}{\pi \sigma_i^2}\right)}\left(1 - e^{-2\lambda_i\left(\sqrt{\frac{2}{\pi}}\frac{1}{\sigma_i} + \frac{\lambda_i \eta}{\pi \sigma_i^2}\right)t}\right).$$
(84)

□

**Lemma C.14.** *Under the assumptions of Corollary C.8, where $\nabla f_\gamma(x) = \nabla f(x) + \sqrt{\Sigma} U$, we have that the dynamics of SignSGD in **Phase 3** is:*

$$dX_t = -\sqrt{\frac{1}{2}}\Sigma^{-\frac{1}{2}}\nabla f(X_t)dt + \sqrt{\eta}\sqrt{I_d - \frac{1}{2}\operatorname{diag}\left(\Sigma^{-\frac{1}{2}}\nabla f(X_t)\right)^2} dW_t.$$
(85)

*Proof of lemma C.14.* We apply Eq. (34) with $\nu = 2$ and linearly approximate $\Xi(x)$ as $|x| < 1$, where $2\Xi(x) \sim \frac{x}{\sqrt{2}}$. □

## C.3 Formal derivation - RMSprop

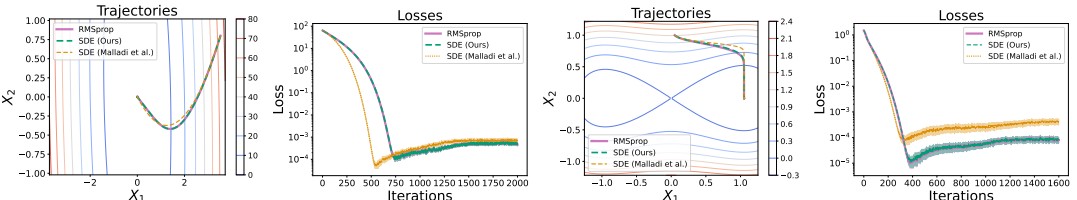

Figure 7: The first two subfigures on the left compare our SDE, that from Malladi et al. (2022), and RMSprop in terms of trajectories and $f(x)$, respectively, for a convex quadratic function. The others subfigures do the same for an embedded saddle and one clearly observes that our derived SDE better matches RMSprop.

In this subsection, we provide our formal derivation of an SDE model for RMSprop. Let us consider the stochastic process $L_t := (X_t, V_t) \in \mathbb{R}^d \times \mathbb{R}^d$ defined as the solution of

$$dX_t = -P_t^{-1}(\nabla f(X_t)dt + \sqrt{\eta}\Sigma(X_t)^{\frac{1}{2}}dW_t)$$
(86)
$$dV_t = \rho((\nabla f(X_t))^2 + \operatorname{diag}(\Sigma(X_t)) - V_t)dt,$$
(87)

where $\beta = 1 - \eta\rho$, $\rho = \mathcal{O}(1)$, and $P_t := \operatorname{diag}(V_t)^{\frac{1}{2}} + \epsilon I_d$.

*Remark* C.15. We observe that the term in blue is the only difference w.r.t. the SDE derived in (Malladi et al., 2022) (see Theorem D.2): This is extremely relevant when the gradient size is not negligible. Figure 7 shows the comparison between our SDE, the one derived in (Malladi et al., 2022), and RMSprop itself: It is clear that even on simple landscapes, our SDE matches the algorithm much better. Importantly, one can observe that the SDE derived in (Malladi et al., 2022) is only slightly worse than ours at the end of the dynamics: As we show in Lemma C.17, Theorem D.2 is a corollary of Theorem C.16 when $\nabla f(x) = \mathcal{O}(\sqrt{\eta})$: It only describes the dynamics where the gradient is vanishing. In Figure 8, we compare the two SDEs in question with RMSprop on an MLP, a CNN, a ResNet, and a Transformer: Our SDE exhibits a superior description of the dynamics.

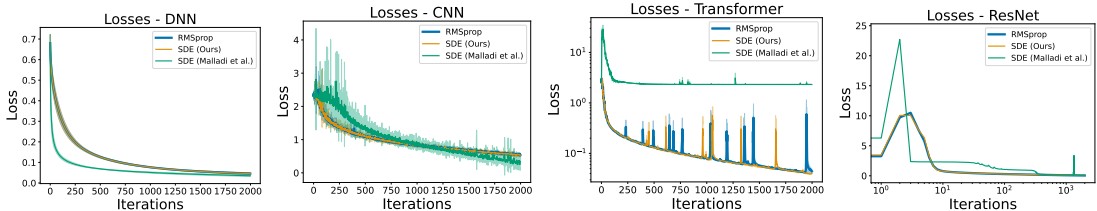

Figure 8: We compare our SDE, that from Malladi et al. (2022), and RMSprop in terms of $f(x)$: The first is an MLP on the Breast Cancer dataset, the second a CNN on MNIST, the third a Transformer on MNIST, and the last a ResNet on CIFAR-10: Ours match the algorithms better.

The following theorem guarantees that such a process is a 1-order SDE of the discrete-time algorithm of RMSprop

$$x_{k+1} = x_k - \eta \frac{\nabla f_{\gamma_k}(x_k)}{\sqrt{v_{k+1}} + \epsilon I_d} \tag{88}$$

$$v_{k+1} = \beta v_k + (1 - \beta) \left(\nabla f_{\gamma_k}(x_k)\right)^2 \tag{89}$$

with $(x_0, v_0) \in \mathbb{R}^d \times \mathbb{R}^d$, $\eta \in \mathbb{R}^{>0}$ is the step size, $\beta = 1 - \rho\eta$ for $\rho = \mathcal{O}(1)$, the mini-batches $\{\gamma_k\}$ are modelled as i.i.d. random variables uniformly distributed on $\{1, \cdots, N\}$, and of size $B \geq 1$.

> **Theorem C.16** (Stochastic modified equations). *Let $0 < \eta < 1, T > 0$ and set $N = \lfloor T/\eta \rfloor$. Let $l_k := (x_k, v_k) \in \mathbb{R}^d \times \mathbb{R}^d, 0 \leq k \leq N$ denote a sequence of RMSprop iterations defined by Eq. (88). Consider the stochastic process $L_t$ defined in Eq. (86) and fix some test function $g \in G$ and suppose that $g$ and its partial derivatives up to order 6 belong to $G$. Then, under Assumption C.2 and $\rho = \mathcal{O}(1)$ there exists a constant $C > 0$ independent of $\eta$ such that for all $k = 0, 1, \ldots, N$, we have*
>
> $$|\mathbb{E}g\left(L_{k\eta}\right) - \mathbb{E}g\left(l_k\right)| \leq C\eta.$$
>
> *That is, the SDE (86) is an order 1 weak approximation of the RMSprop iterations (88).*

*Proof.* The proof is virtually identical to that of Theorem C.5. Therefore, we only report the key steps necessary to conclude the thesis. First of all, we observe that since $\beta = 1 - \eta\rho$

$$v_{k+1} - v_k = -\eta\rho \left(v_k - \left(\nabla f_{\gamma_k}(x_k)\right)^2\right). \tag{90}$$

Then,

$$\frac{1}{\sqrt{v_{k+1}}} = \sqrt{\frac{v_k}{v_{k+1}} \frac{1}{v_k}} = \sqrt{\frac{v_{k+1} + \mathcal{O}(\eta)}{v_{k+1}} \frac{1}{v_k}} = \sqrt{1 + \frac{\mathcal{O}(\eta)}{v_{k+1}}} \sqrt{\frac{1}{v_k}} \sim \sqrt{\frac{1}{v_k}}(1 + \mathcal{O}(\eta)). \tag{91}$$

Therefore, we work with the following algorithm as all the approximations below only carry an additional error of order $\mathcal{O}(\eta^2)$, which we can ignore. Therefore, we have that

$$x_{k+1} - x_k = -\eta \frac{\nabla f_{\gamma_k}(x_k)}{\sqrt{v_k} + \epsilon I_d} \tag{92}$$

$$v_k - v_{k-1} = -\eta\rho \left(v_{k-1} - \left(\nabla f_{\gamma_{k-1}}(x_{k-1})\right)^2\right). \tag{93}$$

Therefore, if $\nabla f_{\gamma_j}(x_j) = \nabla f(x_j) + Z_j(x_j)$, $\mathbb{E}[Z_j(x_j)] = 0$, and $Cov(Z_j(x_j)) = \Sigma(x_j)$

1. $\mathbb{E}[x_{k+1} - x_k] = -\eta \operatorname{diag}(v_k + \epsilon I_d)^{-\frac{1}{2}} \nabla f(x_k)$ ;

2. $\mathbb{E}[v_k - v_{k-1}] = \eta\rho \left[\left(\nabla f(x_{k-1})\right)^2 + \operatorname{diag}(\Sigma(x_k)) - v_{k-1}\right]$ .

747 Then, we have that if $\Phi_k := \frac{\nabla f(x_k)}{\sqrt{v_k}+\epsilon I_d} - \frac{\nabla f_{\gamma_k}(x_k)}{\sqrt{v_k}+\epsilon I_d}$

1.

$$\mathbb{E}[(x_{k+1} - x_k)(x_{k+1} - x_k)^\top] = \mathbb{E}[(x_{k+1} - x_k)]\mathbb{E}[(x_{k+1} - x_k)]^\top \tag{94}$$

$$+ \eta^2 \mathbb{E}\left[(\Phi_k)(\Phi_k)^\top\right] \tag{95}$$

$$= \mathbb{E}[(x_{k+1} - x_k)]\mathbb{E}[(x_{k+1} - x_k)]^\top \tag{96}$$

$$+ \eta^2 (\text{diag}(v_k) + \epsilon I_d)^{-1}\Sigma(x_k); \tag{97}$$

748  2. $\mathbb{E}[(v_k - v_{k-1})(v_k - v_{k-1})^\top] = \mathbb{E}[(v_k - v_{k-1})]\mathbb{E}[(v_k - v_{k-1})]^\top + \mathcal{O}(\rho\eta^2);$

749  3. $\mathbb{E}[(x_{k+1} - x_k)(v_k - v_{k-1})^\top] = \mathbb{E}[(x_{k+1} - x_k)]\mathbb{E}[(v_k - v_{k-1})^\top] + 0.$

750 Therefore

$$dX_t = -P_t^{-1}(\nabla f(X_t)dt + \sqrt{\eta}\Sigma(X_t)^{\frac{1}{2}}dW_t) \tag{98}$$

$$dV_t = \rho(((\nabla f(X_t))^2 + \text{diag}(\Sigma(X_t)) - V_t))dt. \tag{99}$$

751 $\qquad\qquad\qquad\qquad\qquad\qquad\qquad\qquad\qquad\qquad\qquad\qquad\qquad\qquad\qquad\qquad\qquad\qquad\qquad\qquad\qquad\qquad$ □

752 **Lemma C.17.** *If $(\nabla f(x))^2 = \mathcal{O}(\eta)$, Theorem D.2 is a Corollary of Theorem C.16.*

753 *Proof.* In the proof of Theorem C.16, one drops the term $\eta(\nabla f(x))^2$ as it is of order $\eta^2$. □

754 **Corollary C.18.** *Under the assumptions of Theorem C.16 with $\Sigma(x) = \sigma^2 I_d$, $\tilde{\eta} = \kappa\eta$, $\tilde{B} = B\delta$, and*
755 *$\tilde{\rho} = \alpha\rho$,*

$$dX_t = \kappa \, \text{diag}(V_t)^{-\frac{1}{2}}\left(-\nabla f(X_t)dt + \frac{1}{\sqrt{\delta}}\sqrt{\frac{\eta}{B}}\sigma I_d dW_t\right) \tag{100}$$

$$dV_t = \frac{\alpha}{\kappa}\rho\left((\nabla f(X_t))^2 + \frac{\sigma^2}{B\delta}\mathbf{1} - V_t\right)dt. \tag{101}$$

756 **Lemma C.19** (Scaling Rule at Convergence)**.** *Under the assumptions of Corollary C.18, $f$ is $\mu$-*
757 *strongly convex, $\mathcal{L}_\tau := \text{Tr}(\nabla^2 f(x))$, and $(\nabla f(x))^2 = \mathcal{O}(\eta)$, the asymptotic dynamics of the iterates*
758 *of RMSprop satisfies the classic scaling rule $\kappa = \sqrt{\delta}$ because*

$$\mathbb{E}[f(X_t) - f(X_*)] \overset{t\to\infty}{\leq} \frac{\eta\sigma\mathcal{L}_\tau}{4\mu\sqrt{B}}\frac{\kappa}{\sqrt{\delta}}. \tag{102}$$

759 *By enforcing that the speed of $V_t$ matches that of $X_t$, one needs $\tilde{\rho} = \kappa^2\rho$, which implies $\tilde{\beta} =$*
760 *$1 - \kappa^2(1 - \beta)$.*

761 *Proof of Lemma C.19.* In order to recover the scaling of $\beta$, we enforce that the rate at which $V_t$
762 converges to its limit matches the speed of $X_t$: We need $\tilde{\rho} = \kappa^2\rho$, which recovers the classic scaling
763 $\tilde{\beta} = 1 - \kappa^2(1 - \beta)$. Additionally, since $(\nabla f(x))^2 = \mathcal{O}(\eta)$ we have that

$$dX_t = \kappa \, \text{diag}(V_t)^{-\frac{1}{2}}\left(-\nabla f(X_t)dt + \frac{1}{\sqrt{\delta}}\sqrt{\frac{\eta}{B}}\sigma I_d dW_t\right) \tag{103}$$

$$dV_t = \kappa\rho\left(\frac{\sigma^2}{B\delta}\mathbf{1} - V_t\right)dt. \tag{104}$$

764 Therefore, $V_t \overset{t\to\infty}{\to} \frac{\sigma^2}{B\delta}\mathbf{1}$, meaning that under these conditions:

$$dX_t = -\frac{\sqrt{B\delta}\kappa}{\sigma}\nabla f(X_t)dt + \kappa\sqrt{\eta}I_d dW_t, \tag{105}$$

765 which satisfies the following for $\mu$-strongly convex functions

$$d\mathbb{E}[f(X_t) - f(X_*)] \leq -2\kappa\mu\frac{\sqrt{B\delta}}{\sigma}\mathbb{E}[f(X_t) - f(X_*)]dt + \frac{\kappa^2\eta\mathcal{L}_\tau}{2}dt, \tag{106}$$

766    meaning that $\mathbb{E}[f(X_t) - f(X_*)] \overset{t \to \infty}{\lesssim} \frac{\eta \sigma \mathcal{L}_\tau}{4\mu\sqrt{B}} \frac{\kappa}{\sqrt{\delta}}$.

767    Since the asymptotic the loss is $\frac{\eta}{2} \frac{\mathcal{L}_\tau \sigma}{2\mu\sqrt{B}} \frac{\kappa}{\sqrt{\delta}}$ does not depend on $\kappa$ and $\delta$ if $\frac{\kappa}{\sqrt{\delta}} = 1$, we recover the

768    classic scaling rule.              $\square$

769    **Remark:** Under the same conditions, SGD satisfies

$$dX_t = -\kappa \nabla f(X_t)dt + \kappa \frac{1}{\sqrt{\delta}} \sqrt{\frac{\eta}{B}} \sigma I_d dW_t \tag{107}$$

770    and therefore

$$\mathbb{E}[f(X_t) - f(X_*)] \leq (f(X_0) - f(X_*))e^{-2\mu\kappa t} + \frac{\eta}{2} \frac{\mathcal{L}_\tau \sigma^2}{2\mu B} \frac{\kappa}{\delta} \left(1 - e^{-2\mu\kappa t}\right), \tag{108}$$

771    meaning that asymptotically the loss is $\frac{\eta}{2} \frac{\mathcal{L}_\tau \sigma^2}{2\mu B} \frac{\kappa}{\delta}$ which does not depend on $\kappa$ and $\delta$ if $\frac{\kappa}{\delta} = 1$.

772    **Lemma C.20.** *For $f(x) := \frac{x^\top H x}{2}$, the stationary distribution of RMSprop is $(\mathbb{E}[X_\infty]], Cov(X_\infty)) =$*

773    $\left(0, \frac{\eta}{2}\Sigma^{\frac{1}{2}}H^{-1}\right)$.

774    *Proof.* As $(\nabla f(x))^2 = \mathcal{O}(\eta)$ and $t \to \infty$, we have

$$dX_t = -\Sigma^{-\frac{1}{2}}HX_t dt + \sqrt{\eta}I_d dW_t \tag{109}$$

775    which implies that

$$X_t = e^{-\Sigma^{-\frac{1}{2}}Ht} \left(X_0 + \sqrt{\eta} \int_0^t e^{\Sigma^{-\frac{1}{2}}Hs} dW_s\right). \tag{110}$$

776    The thesis follows from the martingale property of Brownian motion and the Itô isometry.    $\square$

777    ## C.4    RMSpropW

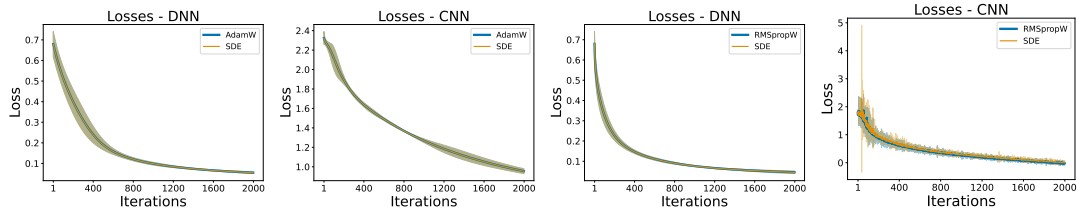

Figure 9: The first two represent the comparison between AdamW and its SDE in terms of $f(x)$. The other two do the same for RMSpropW. In both cases, the first is an MLP on the Breast Cancer Dataset and the second a CNN on MNIST: Our SDEs match the respective optimizers.

778    In this subsection, we derive the SDE of RMSpropW defined as

$$x_{k+1} = x_k - \eta \frac{\nabla f_{\gamma_k}(x_k)}{\sqrt{v_{k+1}} + \epsilon I_d} - \eta\gamma x_k \tag{111}$$

$$v_{k+1} = \beta v_k + (1 - \beta)\left(\nabla f_{\gamma_k}(x_k)\right)^2 \tag{112}$$

779    with $(x_0, v_0) \in \mathbb{R}^d \times \mathbb{R}^d$, $\eta \in \mathbb{R}^{>0}$ is the step size, $\beta = 1 - \rho\eta$ for $\rho = \mathcal{O}(1)$, $\gamma > 0$, the mini-batches

780    $\{\gamma_k\}$ are modelled as i.i.d. random variables uniformly distributed on $\{1, \cdots, N\}$, and of size $B \geq 1$.

781    **Theorem C.21.** *Under the same assumptions as Theorem C.16, the SDE of RMSpropW is*

$$dX_t = -P_t^{-1}(\nabla f(X_t)dt + \sqrt{\eta}\Sigma(X_t)^{\frac{1}{2}}dW_t) - \gamma X_t dt \tag{113}$$

$$dV_t = \rho((\nabla f(X_t))^2 + \text{diag}(\Sigma(X_t)) - V_t))dt, \tag{114}$$

782    *where $\beta = 1 - \eta\rho$, $\rho = \mathcal{O}(1)$, $\gamma > 0$, and $P_t := \text{diag}(V_t)^{\frac{1}{2}} + \epsilon I_d$.*

*Proof.* The proof is the same as the of Theorem C.16 and the only difference is that $\eta\gamma x_k$ is approximated with $\gamma X_t dt$. $\qquad\square$

Figure 4 and Figure 9 validate this result on a variety of architectures and datasets.

**Corollary C.22.** *Under the assumptions of Theorem C.21 with $\Sigma(x) = \sigma^2 I_d$, $\tilde{\eta} = \kappa\eta$, $\tilde{B} = B\delta$, and $\tilde{\rho} = \alpha\rho$, and $\tilde{\gamma} = \xi\gamma$,*

$$dX_t = \kappa \operatorname{diag}(V_t)^{-\frac{1}{2}} \left( -\nabla f(X_t)dt + \frac{1}{\sqrt{\delta}}\sqrt{\frac{\eta}{B}}\sigma I_d dW_t \right) - \xi\gamma\kappa X_t dt \tag{115}$$

$$dV_t = \frac{\alpha}{\kappa}\rho \left( (\nabla f(X_t))^2 + \frac{\sigma^2}{B\delta}\mathbf{1} - V_t \right) dt. \tag{116}$$

**Lemma C.23** (Scaling Rule at Convergence). *Under the assumptions of Corollary C.22, $f$ is $\mu$-strongly convex and L-smooth, $\mathcal{L}_\tau := Tr(\nabla^2 f(x))$, and $(\nabla f(x))^2 = \mathcal{O}(\eta)$, the asymptotic dynamics of the iterates of RMSpropW satisfies the novel scaling rule if $\kappa = \sqrt{\delta}$ and $\xi = \kappa$ because*

$$\mathbb{E}[f(X_t) - f(X_*)] \overset{t\to\infty}{\leq} \frac{\eta\mathcal{L}_\tau\sigma L}{2} \frac{\kappa}{2\mu\sqrt{B\delta}L + \sigma\xi\gamma(L+\mu)}. \tag{117}$$

*By enforcing that the speed of $V_t$ matches that of $X_t$, one needs $\tilde{\rho} = \kappa^2\rho$, which implies $\tilde{\beta} = 1 - \kappa^2(1-\beta)$.*

*Proof of Lemma C.23.* In order to recover the scaling of $\beta$, we enforce that the rate at which $V_t$ converges to its limit matches the speed of $X_t$: We need $\tilde{\rho} = \kappa^2\rho$, which recovers the classic scaling $\tilde{\beta} = 1 - \kappa^2(1-\beta)$. Additionally, since $(\nabla f(x))^2 = \mathcal{O}(\eta)$ we have that

$$dX_t = \kappa \operatorname{diag}(V_t)^{-\frac{1}{2}} \left( -\nabla f(X_t)dt + \frac{1}{\sqrt{\delta}}\sqrt{\frac{\eta}{B}}\sigma I_d dW_t \right) - \kappa\xi\gamma X_t dt \tag{118}$$

$$dV_t = \kappa\rho \left( \frac{\sigma^2}{B\delta}\mathbf{1} - V_t \right) dt. \tag{119}$$

Therefore, $V_t \overset{t\to\infty}{\to} \frac{\sigma^2}{B\delta}\mathbf{1}$, meaning that under these conditions:

$$dX_t = -\frac{\sqrt{B\delta}\kappa}{\sigma}\nabla f(X_t)dt + \kappa\sqrt{\eta}I_d dW_t - \kappa\xi\gamma X_t dt, \tag{120}$$

which satisfies the following for $\mu$-strongly convex and $L$-smooth functions

$$d\mathbb{E}[f(X_t) - f(X_*)] \leq \kappa \left( 2\mu\frac{\sqrt{B\delta}}{\sigma} + \xi\gamma\left(1 + \frac{\mu}{L}\right) \right) \mathbb{E}[f(X_t) - f(X_*)]dt + \frac{\kappa^2\eta\mathcal{L}_\tau}{2}dt, \tag{121}$$

meaning that $\mathbb{E}[f(X_t) - f(X_*)] \overset{t\to\infty}{\leq} \frac{\eta\mathcal{L}_\tau\sigma L}{2}\frac{\kappa}{2\mu\sqrt{B\delta}L+\sigma\xi\gamma(L+\mu)}$.

Since the asymptotic the loss $\frac{\eta\mathcal{L}_\tau\sigma L}{2}\frac{\kappa}{2\mu\sqrt{B\delta}L+\sigma\xi\gamma(L+\mu)}$ does not depend on $\kappa$ and $\delta$ and $\xi$ if $\kappa = \xi = \sqrt{\delta}$, we recover the novel scaling rule. $\qquad\square$

**Lemma C.24.** *For $f(x) := \frac{x^\top Hx}{2}$, the stationary distribution of RMSpropW is $(\mathbb{E}[X_\infty]], Cov(X_\infty)) = \left( 0, \frac{\eta}{2}(H\Sigma^{-\frac{1}{2}} + \gamma I_d)^{-1} \right)$.*

*Proof.* As $(\nabla f(x))^2 = \mathcal{O}(\eta)$ and $t \to \infty$, we have

$$dX_t = -\Sigma^{-\frac{1}{2}}HX_t dt + \sqrt{\eta}I_d dW_t - \gamma X_t dt \tag{122}$$

which implies that

$$X_t = e^{-(\Sigma^{-\frac{1}{2}}H+\gamma I_d)t} \left( X_0 + \sqrt{\eta}\int_0^t e^{(\Sigma^{-\frac{1}{2}}H+\gamma I_d)s}dW_s \right). \tag{123}$$

The thesis follows from the martingale property of Brownian motion and the Itô isometry. $\qquad\square$

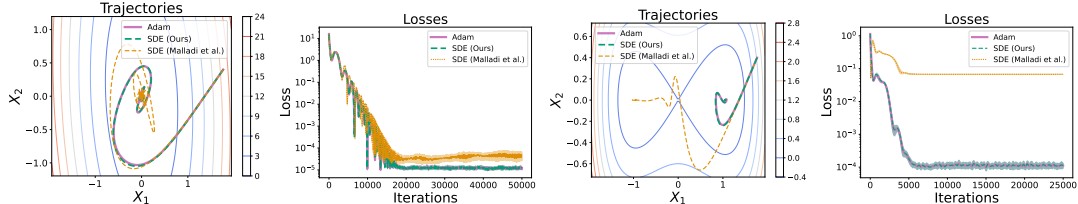

Figure 10: The first two on the left compare our SDE, that from Malladi et al. (2022), and Adam in terms of trajectories and $f(x)$, respectively, for a convex quadratic function. The others do the same for an embedded saddle: Ours clearly matches Adam better.

## C.5 Formal derivation - Adam

In this subsection, we provide our formal derivation of an SDE model for Adam. Let us consider the stochastic process $L_t := (X_t, M_t, V_t) \in \mathbb{R}^d \times \mathbb{R}^d \times \mathbb{R}^d$ defined as the solution of

$$dX_t = -\frac{\sqrt{\gamma_2(t)}}{\gamma_1(t)} P_t^{-1}(M_t + \eta\rho_1\left(\nabla f\left(X_t\right) - M_t\right))dt \tag{124}$$

$$dM_t = \rho_1\left(\nabla f\left(X_t\right) - M_t\right)dt + \sqrt{\eta}\rho_1\Sigma^{1/2}\left(X_t\right)dW_t \tag{125}$$

$$dV_t = \rho_2\left(\left(\nabla f(X_t)\right)^2 + \operatorname{diag}\left(\Sigma\left(X_t\right)\right) - V_t\right)dt, \tag{126}$$

where $\beta_i = 1 - \eta\rho_i$, $\gamma_i(t) = 1 - e^{-\rho_i t}$, $\rho_1 = \mathcal{O}(\eta^{-\zeta})$ s.t. $\zeta \in (0,1)$, $\rho_2 = \mathcal{O}(1)$, and $P_t = \operatorname{diag}\sqrt{V_t} + \epsilon\sqrt{\gamma_2(t)}I_d$.

*Remark* C.25. The terms in purple and in blue are the two differences w.r.t. that of (Malladi et al., 2022) which is reported in Theorem D.5. The first appears because we assume realistic values of $\beta_1$ while the second appears because we allow the gradient size to be non-negligible. For two simple landscapes, Figure 10 compares our SDE and that of Malladi et al. (2022) with Adam: In both cases, the first part of the dynamics is perfectly represented only by our SDE. While the discrepancy between the SDE of (Malladi et al., 2022) and Adam is asymptotically negligible in the convex setting, we observe that in the nonconvex case, it converges to a different local minimum than ours and of Adam. Finally, Theorem D.5 is a corollary of ours when $(\nabla f(x))^2 = \mathcal{O}(\eta)$ and $\rho_1 = \mathcal{O}(1)$: It only describes the dynamics where the gradient to noise ratio is vanishing and only for unrealistic values of $\beta_1 = 1 - \eta\rho_1$. In Figure 11, we compare the dynamics of our SDE, that of Malladi et al. (2022), and Adam on an MLP, a CNN, a ResNet, and a Transformer. One can clearly see that our SDE more accurately captures the dynamics. Details on these experiments are available in Appendix F.

The following theorem guarantees that such a process is a 1-order SDE of the discrete-time algorithm of Adam

$$v_{k+1} = \beta_2 v_k + (1 - \beta_2)\left(\nabla f_{\gamma_k}(x_k)\right)^2 \tag{127}$$

$$m_{k+1} = \beta_1 m_k + (1 - \beta_1)\nabla f_{\gamma_k}(x_k) \tag{128}$$

$$\hat{m}_k = m_k\left(1 - \beta_1^k\right)^{-1} \tag{129}$$

$$\hat{v}_k = v_k\left(1 - \beta_2^k\right)^{-1} \tag{130}$$

$$x_{k+1} = x_k - \eta\frac{\hat{m}_{k+1}}{\sqrt{\hat{v}_{k+1}} + \epsilon I_d}, \tag{131}$$

with $(x_0, m_0, v_0) \in \mathbb{R}^d \times \mathbb{R}^d \times \mathbb{R}^d$, $\eta \in \mathbb{R}^{>0}$ is the step size, $\beta_i = 1 - \rho_i\eta$ for $\rho_1 = \mathcal{O}(\eta^{-\zeta})$ s.t. $\zeta \in (0,1)$, $\rho_2 = \mathcal{O}(1)$, the mini-batches $\{\gamma_k\}$ are modelled as i.i.d. random variables uniformly distributed on $\{1, \cdots, N\}$, and of size $B \geq 1$.

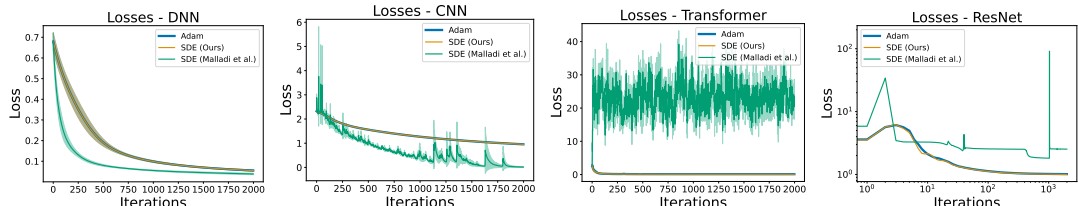

Figure 11: We compare our SDE, that from Malladi et al. (2022), and Adam in terms of $f(x)$: The first is an MLP on the Breast Cancer dataset, the second a CNN on MNIST, the third a Transformer on MNIST, and the last a ResNet on CIFAR-10: Ours match the algorithms better.

---

**Theorem C.26** (Stochastic modified equations). *Let $0 < \eta < 1, T > 0$ and set $N = \lfloor T/\eta \rfloor$. Let $l_k := (x_k, m_k, v_k) \in \mathbb{R}^d \times \mathbb{R}^d \times \mathbb{R}^d, 0 \le k \le N$ denote a sequence of Adam iterations defined by Eq. (127). Consider the stochastic process $L_t$ defined in Eq. (124) and fix some test function $g \in G$ and suppose that $g$ and its partial derivatives up to order 6 belong to $G$. Then, under Assumption C.2 $\rho_1 = \mathcal{O}(\eta^{-\zeta})$ s.t. $\zeta \in (0,1)$, while $\rho_2 = \mathcal{O}(1)$, there exists a constant $C > 0$ independent of $\eta$ such that for all $k = 0, 1, \ldots, N$, we have*

$$|\mathbb{E}g\left(L_{k\eta}\right) - \mathbb{E}g\left(l_k\right)| \le C\eta.$$

*That is, the SDE (124) is an order 1 weak approximation of the Adam iterations (127).*

829

*Proof.* The proof is virtually identical to that of Theorem C.5. Therefore, we only report the key steps necessary to conclude the thesis. First of all, we observe that since $\beta_1 = 1 - \eta\rho_1$

$$v_{k+1} - v_k = -\eta\rho_1 \left( v_k - (\nabla f_{\gamma_k}(x_k))^2 \right). \tag{132}$$

Then,

$$\frac{1}{\sqrt{v_{k+1}}} = \sqrt{\frac{v_k}{v_{k+1}} \frac{1}{v_k}} = \sqrt{\frac{v_{k+1} + \mathcal{O}(\eta)}{v_{k+1}} \frac{1}{v_k}} = \sqrt{1 + \frac{\mathcal{O}(\eta)}{v_{k+1}}} \sqrt{\frac{1}{v_k}} \sim \sqrt{\frac{1}{v_k}}(1 + \mathcal{O}(\eta)). \tag{133}$$

Therefore, we work with the following algorithm as all approximations only carry an additional error of order $\mathcal{O}(\eta^2)$, which we can ignore. Therefore, we have that

$$v_k - v_{k-1} = -\eta\rho_2 \left( v_{k-1} - \left( \nabla f_{\gamma_{k-1}}(x_{k-1}) \right)^2 \right) \tag{134}$$

$$m_{k+1} - m_k = -\eta\rho_1 \left( m_k - \nabla f_{\gamma_k}(x_k) \right) \tag{135}$$

$$\hat{m}_k = m_k \left( 1 - \beta_1^k \right)^{-1} \tag{136}$$

$$\hat{v}_k = v_k \left( 1 - \beta_1^k \right)^{-1} \tag{137}$$

$$x_{k+1} - x_k = -\frac{\eta}{\sqrt{v_k} + \epsilon I_d} \frac{\sqrt{1 - (1 - \eta\rho_2)^k}}{1 - (1 - \eta\rho_1)^{k+1}} (m_k + \eta\rho_1(\nabla f_{\gamma_k}(x_k) - m_k)). \tag{138}$$

Therefore, if $\nabla f_{\gamma_j}(x_j) = \nabla f(x_j) + Z_j(x_j)$ and $\mathbb{E}[Z_j(x_j)] = 0$, and $Cov(Z_j(x_j)) = \Sigma(x_j)$, we have that

1. $\mathbb{E}[v_k - v_{k-1}] = \eta\rho_2 \left[ (\nabla f(x_{k-1}))^2 + \text{diag}(\Sigma(x_k)) - v_{k-1} \right]$ ;

2. $\mathbb{E}[m_{k+1} - m_k] = \eta\rho_1 \left[ \nabla f(x_k) - m_k \right]$ ;

3. $\mathbb{E}[x_{k+1} - x_k] = -\frac{\eta}{\sqrt{v_k} + \epsilon I_d} \frac{\sqrt{1 - (1 - \eta\rho_2)^k}}{1 - (1 - \eta\rho_1)^{k+1}} (m_k + \eta\rho_1(\nabla f(x_k) - m_k))$ .

Then, we have

 1. $\mathbb{E}[(x_{k+1} - x_k)(x_{k+1} - x_k)^\top] = \mathbb{E}[(x_{k+1} - x_k)]\mathbb{E}[(x_{k+1} - x_k)]^\top + \mathcal{O}(\eta^4 \rho_1^2)$;

 2. $\mathbb{E}[(x_{k+1} - x_k)(m_k - m_{k-1})^\top] = \mathbb{E}[(x_{k+1} - x_k)]\mathbb{E}[(m_k - m_{k-1})]^\top + 0$;

 3. $\mathbb{E}[(x_{k+1} - x_k)(v_k - v_{k-1})^\top] = \mathbb{E}[(x_{k+1} - x_k)]\mathbb{E}[(v_k - v_{k-1})]^\top + 0$;

 4. $\mathbb{E}[(v_k - v_{k-1})(v_k - v_{k-1})^\top] = \mathbb{E}[(v_k - v_{k-1})]\mathbb{E}[(v_k - v_{k-1})]^\top + \mathcal{O}(\eta^2 \rho_2^2)$;

 5. $\mathbb{E}[(m_k - m_{k-1})(m_k - m_{k-1})^\top] = \mathbb{E}[(m_k - m_{k-1})]\mathbb{E}[(m_k - m_{k-1})]^\top + \eta^2 \rho_1^2 \Sigma(x_{k-1})$;

 6. $\mathbb{E}[(v_k - v_{k-1})(m_k - m_{k-1})^\top] = \mathbb{E}[(v_k - v_{k-1})]\mathbb{E}[(m_k - m_{k-1})]^\top + \mathcal{O}(\eta^2 \rho_1 \rho_2)$.

 Since in real-world applications, $\rho_1 = \mathcal{O}(\eta^{-\zeta})$ s.t. $\zeta \in (0,1)$, while $\rho_2 = \mathcal{O}(1)$, we have

$$dX_t = -\frac{\sqrt{\gamma_2(t)}}{\gamma_1(t)} P_t^{-1}(M_t + \eta \rho_1 (\nabla f(X_t) - M_t))dt \tag{139}$$

$$dM_t = \rho_1 (\nabla f(X_t) - M_t) dt + \sqrt{\eta}\rho_1 \Sigma^{1/2}(X_t) dW_t \tag{140}$$

$$dV_t = \rho_2 \left((\nabla f(X_t))^2 + \mathrm{diag}(\Sigma(X_t)) - V_t\right) dt. \tag{141}$$

 where $\beta_i = 1 - \eta \rho_i$, $\gamma_i(t) = 1 - e^{-\rho_i t}$, and $P_t = \mathrm{diag}\sqrt{V_t} + \epsilon\sqrt{\gamma_2(t)} I_d$. $\qquad\square$

 **Corollary C.27.** *Under the assumptions of Theorem C.26 with $\Sigma(x) = \sigma^2 I_d$, $\tilde{\eta} = \kappa\eta$, $\tilde{B} = B\delta$,*
 *$\tilde{\rho}_1 = \alpha_1 \rho_1$, and $\tilde{\rho}_2 = \alpha_2 \rho_2$*

$$dX_t = -\kappa\frac{\sqrt{\gamma_2(t)}}{\gamma_1(t)} P_t^{-1}(M_t + \eta\alpha_1 \rho_1 (\nabla f(X_t) - M_t))dt \tag{142}$$

$$dM_t = \frac{\alpha_1 \rho_1}{\kappa} (\nabla f(X_t) - M_t) dt + \sqrt{\eta}\frac{\alpha_1 \rho_1}{\kappa}\frac{\sigma}{\sqrt{B\delta}} I_d dW_t \tag{143}$$

$$dV_t = \frac{\alpha_2 \rho_2}{\kappa} \left((\nabla f(X_t))^2 + \frac{\sigma^2}{B\delta} I_d - V_t\right) dt. \tag{144}$$

 **Lemma C.28.** *Under the assumptions of Corollary C.27, $f$ is $\mu$-strongly convex, $\mathcal{L}_\tau := Tr(\nabla^2 f(x))$,*
 *and $(\nabla f(x))^2 = \mathcal{O}(\eta)$, the asymptotic dynamics of the iterates of Adam satisfies the classic scaling*
 *rule $\kappa = \sqrt{\delta}$ because $\mathbb{E}[f(X_t)] \overset{t\to\infty}{\leq} \frac{\eta\sigma\mathcal{L}_\tau}{4\sqrt{B}}\frac{\kappa}{\sqrt{\delta}}$. To enforce that the speed of $M_t$ and $V_t$ match that of*
 *$X_t$, one needs $\tilde{\rho}_i = \kappa^2 \rho_i$, which implies $\tilde{\beta}_i = 1 - \kappa^2(1 - \beta_i)$.*

 *Proof.* First of all, we need to ensure that the relative speeds of $X_t$, $M_t$, and $V_t$ match. Therefore,
 we select $\alpha_i = \kappa^2$, which recovers the scaling rules for $\tilde{\beta}_i = 1 - \kappa^2(1 - \beta_i)$. Then, recalling that
 $(\nabla f(x))^2 = \mathcal{O}(\eta)$, we have that as $t \to \infty$, $V_t \to \frac{\sigma^2}{B\delta}$, and $M_t \to \nabla f(X_t)$ with high probability.
 Therefore,

$$dX_t = -\kappa\frac{\sqrt{B\delta}}{\sigma}\nabla f(X_t)dt \tag{145}$$

$$dM_t = \kappa\sqrt{\eta}\rho_1 \frac{\sigma}{\sqrt{B\delta}} dW_t \tag{146}$$

$$dV_t = 0. \tag{147}$$

 Therefore, if $H(X_t, V_t) := f(X_t) + \frac{\mathcal{L}_\tau \delta B}{\rho^2 \sigma^2}\frac{\|M_t\|_2^2}{2}$ and $\xi \in (0,1)$ we have that by Itô's lemma,

$$dH(X_t, V_t) = -(\nabla f(X_t))^\top \left( \kappa \frac{\sqrt{B\delta}}{\sigma} \nabla f(X_t) \right) dt + \left( \frac{\mathcal{L}_\tau \delta B}{\rho^2 \sigma^2} M_t \right) \kappa \sqrt{\eta} \rho_1 \frac{\sigma}{\sqrt{B\delta}} dW_t \qquad (148)$$

$$+ \frac{1}{2} \left( \frac{\mathcal{L}_\tau \delta B}{\rho^2 \sigma^2} \right) \kappa^2 \eta \rho^2 \frac{\sigma^2}{B\delta} dt \qquad (149)$$

$$= - \left( \kappa \frac{\sqrt{B\delta}}{\sigma} \right) \|\nabla f(X_t)\|_2^2 dt + \text{Noise} + \frac{\kappa^2 \eta \lambda}{2} dt \qquad (150)$$

$$= - \left( \kappa \frac{\sqrt{B\delta}}{\sigma} \right) \left( \xi \|\nabla f(X_t)\|_2^2 + (1-\xi) \|\nabla f(X_t)\|_2^2 \right) dt + \text{Noise} + \frac{\kappa^2 \eta \lambda}{2} dt \qquad (151)$$

$$\leq -2\kappa\mu \frac{\sqrt{B\delta}}{\sigma} \xi \left( f(X_t) + \frac{1-\xi}{\mu\xi} \frac{\|\nabla f(X_t)\|_2^2}{2} \right) dt + \text{Noise} + \frac{\kappa^2 \eta \lambda}{2} dt. \qquad (152)$$

Let us now select $\xi$ such that $\frac{1-\xi}{\mu\xi} = \frac{\mathcal{L}_\tau \delta B}{\rho^2 \sigma^2}$, this means that $\xi = \frac{\sigma^2 \rho^2}{\sigma^2 \rho^2 + \mu \mathcal{L}_\tau \sigma B} \in (0,1)$ and $\frac{1}{\xi} = 1 + \mu \frac{\mathcal{L}_\tau \delta B}{\rho^2 \sigma^2}$. Since $M_t \to \nabla f(X_t)$, we have that

$$dH(X_t, V_t) \leq -2\kappa\mu \frac{\sqrt{B\delta}}{\sigma} \xi H(X_t, V_t) dt + \frac{\kappa^2 \eta \lambda}{2} dt + \text{Noise}. \qquad (153)$$

Therefore,

$$\frac{\mathbb{E}[f(X_t)]}{\xi} = \left( 1 + \mu \frac{\mathcal{L}_\tau \delta B}{\rho^2 \sigma^2} \right) \mathbb{E}[f(X_t)] \leq \mathbb{E}[H(X_t, V_t)] \overset{t\to\infty}{\leq} \frac{1}{\xi} \frac{\eta \sigma \mathcal{L}_\tau}{4\mu\sqrt{B}} \frac{\kappa}{\sqrt{\delta}}, \qquad (154)$$

which implies that

$$\mathbb{E}[f(X_t)] \overset{t\to\infty}{\leq} \frac{\eta \sigma \mathcal{L}_\tau}{4\mu\sqrt{B}} \frac{\kappa}{\sqrt{\delta}}. \qquad (155)$$

Analogously,

$$\mathbb{E}[f(X_t) - f(X_*)] \overset{t\to\infty}{\leq} \frac{\eta \sigma \mathcal{L}_\tau}{4\mu\sqrt{B}} \frac{\kappa}{\sqrt{\delta}}. \qquad (156)$$

which gives the square root scaling rule. $\qquad \square$

**Lemma C.29.** *Under the assumptions of Corollary C.27, $f(x) = \frac{x^\top H x}{2}$ s.t. $H = \text{diag}(\lambda_1, \cdots, \lambda_d)$ and $(\nabla f(x))^2 = \mathcal{O}(\eta)$, the dynamics of Adam implies that $f(X_t) \to \frac{\eta \sigma d}{4\sqrt{B}} \frac{\kappa}{\sqrt{\delta}}$.*

*Proof.* Recalling that $(\nabla f(x))^2 = \mathcal{O}(\eta)$, we have that as $t \to \infty$, $V_t \to \frac{\sigma^2}{B\delta}$, and $M_t \to \lambda X_t$ with high probability. Therefore, in the one-dimensional case

$$dX_t = -\kappa \frac{\sqrt{B\delta}}{\sigma} \lambda X_t dt \qquad (157)$$

$$dM_t = \kappa \sqrt{\eta} \rho_1 \frac{\sigma}{\sqrt{B\delta}} dW_t \qquad (158)$$

$$dV_t = 0. \qquad (159)$$

Therefore, if $H(X_t, V_t) := \frac{\lambda X_t^2}{2} + \frac{\lambda \delta B}{\rho^2 \sigma^2} \frac{M_t^2}{2}$,[5] we have that by Itô's lemma,

---

[5]Inspired by (Barakat and Bianchi, 2021)

$$dH(X_t, V_t) = -(\lambda X_t) \left( \kappa \frac{\sqrt{B\delta}}{\sigma} \lambda X_t \right) dt + \left( \frac{\lambda \delta B}{\rho^2 \sigma^2} M_t \right) \kappa \sqrt{\eta} \rho_1 \frac{\sigma}{\sqrt{B\delta}} dW_t \tag{160}$$

$$+ \frac{1}{2} \left( \frac{\lambda \delta B}{\rho^2 \sigma^2} \right) \kappa^2 \eta \rho^2 \frac{\sigma^2}{B\delta} dt \tag{161}$$

$$= -2\kappa\lambda \frac{\sqrt{B\delta}}{\sigma} f(X_t) dt + \frac{\kappa^2 \eta \rho^2 \sigma^2}{2B\delta} \frac{\lambda \delta B}{\rho^2 \sigma^2} dt + \text{Noise.} \tag{162}$$

$$= -2\kappa\lambda \frac{\sqrt{B\delta}}{\sigma} f(X_t) dt + \frac{\kappa^2 \eta \lambda}{2} dt + \text{Noise.} \tag{163}$$

871  Once again, since $M_t \to \lambda X_t$, we have that

$$H(X_t, V_t) = \frac{\lambda X_t^2}{2} + \frac{\lambda \delta B}{\rho^2 \sigma^2} \frac{M_t^2}{2} \to \frac{\lambda X_t^2}{2} + \lambda \frac{\lambda \delta B}{\rho^2 \sigma^2} \frac{\lambda X_t^2}{2} = \left( 1 + \lambda \frac{\lambda \delta B}{\rho^2 \sigma^2} \right) \frac{\lambda X_t^2}{2} =: K f(X_t). \tag{164}$$

872  Therefore,

$$K d\mathbb{E}[f(X_t)] = -2\kappa\lambda \frac{\sqrt{B\delta}}{\sigma} \mathbb{E}[f(X_t)] dt + \frac{\kappa^2 \eta \lambda}{2} dt, \tag{165}$$

873  which implies that $\mathbb{E}[f(X_t)] \to \frac{\eta \sigma}{4\sqrt{B}} \frac{\kappa}{\sqrt{\delta}}$, which also gives the square root scaling rule. The general-
874  ization to $d$ dimension is analogous and one needs to sum across all the dimensions. $\qquad \square$

875  **Lemma C.30.** *Let* $f(x) := \frac{x^\top H x}{2}$ *where* $H = \text{diag}(\lambda_1, \ldots, \lambda_d)$. *The stationary distribution of*
876  *Adam is* $(\mathbb{E}[X_\infty]], Cov(X_\infty)) = \left( 0, \frac{\eta}{2} \Sigma^{\frac{1}{2}} H^{-1} \right)$.

877  *Proof.* The expected value follows immediately from the fact that

$$dX_t = -\Sigma^{-\frac{1}{2}} X_t dt \tag{166}$$

878  For the covariance, we focus on the one-dimensional case. We define $H(X_t, V_t) := \frac{X_t^2}{2} + \frac{\lambda^2}{2\sigma^2 \rho^2} \frac{M_t^2}{2}$.
879  With the same arguments as Lemma C.29, we have

$$d(X_t)^2 = -\frac{\lambda}{\sigma} X_t^2 dt + \frac{\eta}{2} dt + \text{Noise,} \tag{167}$$

880  which implies that

$$\mathbb{E}[X_t^2] \overset{t \to 0}{\Rightarrow} \frac{\eta}{2} \frac{\sigma}{\lambda}. \tag{168}$$

881  The thesis follows by applying the same logic to multiple dimensions. $\qquad \square$

## C.6  AdamW

883  In this subsection, we derive the SDE of AdamW defined as defined as

$$v_{k+1} = \beta_2 v_k + (1 - \beta_2) \left( \nabla f_{\gamma_k}(x_k) \right)^2 \tag{169}$$

$$m_{k+1} = \beta_1 m_k + (1 - \beta_1) \nabla f_{\gamma_k}(x_k) \tag{170}$$

$$\hat{m}_k = m_k \left( 1 - \beta_1^k \right)^{-1} \tag{171}$$

$$\hat{v}_k = v_k \left( 1 - \beta_2^k \right)^{-1} \tag{172}$$

$$x_{k+1} = x_k - \eta \frac{\hat{m}_{k+1}}{\sqrt{\hat{v}_{k+1}} + \epsilon I_d} - \eta \gamma x_k \tag{173}$$

884  with $(x_0, m_0, v_0) \in \mathbb{R}^d \times \mathbb{R}^d \times \mathbb{R}^d$, $\eta \in \mathbb{R}^{>0}$ is the step size, $\beta_i = 1 - \rho_i \eta$ for $\rho_1 = \mathcal{O}(\eta^{-\zeta})$
885  s.t. $\zeta \in (0, 1)$, $\rho_2 = \mathcal{O}(1)$, $\gamma > 0$, the mini-batches $\{\gamma_k\}$ are modelled as i.i.d. random variables
886  uniformly distributed on $\{1, \cdots, N\}$, and of size $B \geq 1$.

**Theorem C.31.** *Under the same assumptions as Theorem C.26, the SDE of AdamW is*

$$dX_t = -\frac{\sqrt{\gamma_2(t)}}{\gamma_1(t)} P_t^{-1}(M_t + \eta\rho_1 \left(\nabla f\left(X_t\right) - M_t\right))dt - \gamma X_t dt \tag{174}$$

$$dM_t = \rho_1 \left(\nabla f\left(X_t\right) - M_t\right) dt + \sqrt{\eta}\rho_1 \Sigma^{1/2} \left(X_t\right) dW_t \tag{175}$$

$$dV_t = \rho_2 \left((\nabla f(X_t))^2 + \mathrm{diag}\left(\Sigma\left(X_t\right)\right) - V_t\right) dt. \tag{176}$$

*where $\beta_i = 1 - \eta\rho_i$, $\gamma > 0$, $\gamma_i(t) = 1 - e^{-\rho_i t}$, and $P_t = \mathrm{diag}\sqrt{V_t} + \epsilon\sqrt{\gamma_2(t)}I_d$.*

*Proof.* The proof is the same as the of Theorem C.26 and the only difference is that $\eta\gamma x_k$ is approximated with $\gamma X_t dt$. □

Figure 4 and Figure 9 validate this result on a variety of architectures and datasets.

**Corollary C.32.** *Under the assumptions of Theorem C.31 with $\Sigma(x) = \sigma^2 I_d$, $\tilde{\eta} = \kappa\eta$, $\tilde{B} = B\delta$, $\tilde{\rho}_1 = \alpha_1\rho_1$, $\tilde{\gamma} : \xi\gamma$, and $\tilde{\rho}_2 = \alpha_2\rho_2$*

$$dX_t = -\kappa\frac{\sqrt{\gamma_2(t)}}{\gamma_1(t)} P_t^{-1}(M_t + \eta\alpha_1\rho_1 \left(\nabla f\left(X_t\right) - M_t\right))dt - \kappa\xi\gamma X_t dt \tag{177}$$

$$dM_t = \frac{\alpha_1\rho_1}{\kappa} \left(\nabla f\left(X_t\right) - M_t\right) dt + \sqrt{\eta}\frac{\alpha_1\rho_1}{\kappa}\frac{\sigma}{\sqrt{B\delta}}I_d dW_t \tag{178}$$

$$dV_t = \frac{\alpha_2\rho_2}{\kappa} \left((\nabla f(X_t))^2 + \frac{\sigma^2}{B\delta}I_d - V_t\right) dt. \tag{179}$$

**Lemma C.33** (Scaling Rule at Convergence). *Under the assumptions of Corollary C.32, $f$ is $\mu$-strongly convex and $L$-smooth, $\mathcal{L}_\tau := Tr(\nabla^2 f(x))$, and $(\nabla f(x))^2 = \mathcal{O}(\eta)$, the asymptotic dynamics of the iterates of AdamW satisfies the novel scaling rule if $\kappa = \sqrt{\delta}$ and $\xi = \kappa$ because*

$$\mathbb{E}[f(X_t) - f(X_*)] \overset{t\to\infty}{\leq} \frac{\eta\mathcal{L}_\tau\sigma L}{2}\frac{\kappa}{2\mu\sqrt{B\delta}L + \sigma\xi\gamma(L+\mu)} \tag{180}$$

*By enforcing that the speed of $V_t$ matches that of $X_t$, one needs $\tilde{\rho} = \kappa^2\rho$, which implies $\tilde{\beta}_i = 1 - \kappa^2(1 - \beta_i)$.*

*Proof.* The proof is the same as Lemma C.28 where we also use $L$-smoothness as in Lemma C.23. □

**Lemma C.34.** *For $f(x) := \frac{x^\top H x}{2}$, the stationary distribution of AdamW is $(\mathbb{E}[X_\infty]], Cov(X_\infty)) = \left(0, \frac{\eta}{2}(H\Sigma^{-\frac{1}{2}} + \gamma I_d)^{-1}\right)$.*

*Proof.* The proof is the same as Lemma C.30. □

# D    SDEs from the literature

**Theorem D.1** (Original Malladi's Statement). *Let $\sigma_0 := \sigma\eta$, $\epsilon_0 := \epsilon\eta$, and $c_2 := \frac{1-\beta}{\eta^2}$. Define the state of the SDE as $L_t = (X_t, u_t)$ and the dynamics as*

$$dX_t = -P_t^{-1} \left(\nabla f\left(X_t\right) dt + \sigma_0\Sigma^{1/2} \left(X_t\right) dW_t\right) \tag{181}$$

$$du_t = c_2 \left(\mathrm{diag}\left(\Sigma\left(X_t\right)\right) - u_t\right) dt \tag{182}$$

*where $P_t := \sigma_0 \mathrm{diag}\left(u_t\right)^{1/2} + \epsilon_0 I_d$.*

**Theorem D.2** (Informal Statement of Theorem C.2 Malladi et al. (2022)). *Under sufficient regularity conditions and $\nabla f(x) = \mathcal{O}(\sqrt{\eta})$, the following SDE is an order 1 weak approximation of RMSprop:*

$$dX_t = -P_t^{-1}(\nabla f(X_t)dt + \sqrt{\eta}\Sigma(X_t)^{\frac{1}{2}} dW_t) \tag{183}$$

$$dV_t = \rho(\mathrm{diag}(\Sigma(X_t)) - V_t)dt, \tag{184}$$

*where $\beta = 1 - \eta\rho$, $\rho = \mathcal{O}(1)$, and $P_t := \mathrm{diag}\left(V_t\right)^{\frac{1}{2}} + \epsilon I_d$.*

**Lemma D.3.** *Theorem D.1 and Theorem D.2 are equivalent.*

*Proof.* It follows applying time rescaling $t := \eta\xi$ and observing that $W_t = W_{\eta\xi} = \sqrt{\eta}W_\xi$. □

**Theorem D.4** (Original Malladi's Statement). *Let $c_1 := (1 - \beta_1)/\eta^2$, $c_2 := (1 - \beta_2)/\eta^2$ and define $\sigma_0, \epsilon_0$ in Theorem D.1. Let $\gamma_1(t) := 1 - \exp(-c_1 t)$ and $\gamma_2(t) := 1 - \exp(-c_2 t)$. Define the state of the SDE as $L_t = (X_t, m_t, u_t)$ and the dynamics as*

$$dX_t = -\frac{\sqrt{\gamma_2(t)}}{\gamma_1(t)} P_t^{-1} m_t dt \tag{185}$$

$$dm_t = c_1 \left( \nabla f(X_t) - m_t \right) dt + \sigma_0 c_1 \Sigma^{1/2}(X_t) dW_t, \tag{186}$$

$$du_t = c_2 \left( \text{diag}(\Sigma(X_t)) - u_t \right) dt, \tag{187}$$

*where $P_t := \sigma_0 \,\text{diag}(u_t)^{1/2} + \epsilon_0\sqrt{\gamma_2(t)} I_d$.*

**Theorem D.5** (Informal Statement of Theorem D.2 Malladi et al. (2022)). *Under sufficient regularity conditions and $\nabla f(x) = \mathcal{O}(\sqrt{\eta})$, the following SDE is an order 1 weak approximation of Adam:*

$$dX_t = -\frac{\sqrt{\gamma_2(t)}}{\gamma_1(t)} P_t^{-1} M_t dt \tag{188}$$

$$dM_t = \rho_1 \left( \nabla f(X_t) - M_t \right) dt + \sqrt{\eta}\rho_1 \Sigma^{1/2}(X_t) dW_t \tag{189}$$

$$dV_t = \rho_2 \left( \text{diag}(\Sigma(X_t)) - V_t \right) dt. \tag{190}$$

*where $\beta_i = 1 - \eta\rho_i$, $\gamma_i(t) = 1 - e^{-\rho_i t}$, $\rho_i = \mathcal{O}(1)$, and $P_t = \text{diag}\sqrt{V_t} + \epsilon\sqrt{\gamma_2(t)} I_d$.*

**Lemma D.6.** *Theorem D.4 and Theorem D.5 are equivalent.*

*Proof.* It follows applying time rescaling $t := \eta\xi$ and observing that $W_t = W_{\eta\xi} = \sqrt{\eta}W_\xi$. □

# E SDE cannot be derived nor used naively

In this section, we provide a gentle introduction to the meaning of deriving an SDE model for an optimizer and discuss how SDEs have been used to derive scaling rules. To aid the intuition of the reader, we informally derive an SDE for SGD with learning rate $\eta$, mini-batches $\gamma_B$ of size $B$, and starting point $x_0 = x$, which we dub $\text{SGD}^{(\eta,B)}$. The iterates are given by:

$$x_{k+1} = x_k - \eta \nabla f_{\gamma_k^B}(x_k) \tag{191}$$

which for $U_k := \sqrt{\eta}(\nabla f(x_k) - \nabla f_{\gamma_k^B}(x_k))$, we rewrite as

$$x_k - \eta \nabla f(x_k) + \sqrt{\eta} U_k, \tag{192}$$

where $\mathbb{E}[U_k] = 0$ and $Cov(U_k) = \frac{\eta}{B}\Sigma(x_k) = \frac{\eta}{B}\frac{1}{n}\sum_{i=0}^{B}(\nabla f(x_k) - \nabla f_i(x_k))(\nabla f(x) - \nabla f_i(x_k))^\top$. If we now consider the SDE

$$dX_t = -\nabla f(X_t)dt + \sqrt{\frac{\eta}{B}}\Sigma(X_t)^{\frac{1}{2}}dW_t, \tag{193}$$

its Euler-Maruyama discretization with pace $\Delta t = \eta$ and $Z_k \sim \mathcal{N}(0, I_d)$ is

$$X_{k+1} = X_k - \eta \nabla f(X_k) + \sqrt{\eta}\sqrt{\frac{\eta}{B}}\Sigma(X_k)^{\frac{1}{2}}Z_k. \tag{194}$$

Since the Eq. (191) and Eq. (194) share the first two moments, it is reasonable that by identifying $t = k\eta$, the SDE in Eq. (193) is a good model to describe the iterates of SGD in Eq. (191).

Informally, we need a "good model", which is an SDE that is close to the real optimizer. This is formalized in the following definition which comes from the field of numerical analysis of SDEs (see Mil'shtein (1986)) and bounds the disparity between the the discrete and the continuous process.

**Definition E.1** (Weak Approximation). A continuous-time stochastic process $\{X_t\}_{t \in [0,T]}$ is an order $\alpha$ weak approximation (or $\alpha$-order SDE) of a discrete stochastic process $\{x_k\}_{k=0}^{\lfloor T/\eta \rfloor}$ if for every polynomial growth function $g$, there exists a positive constant $C$, independent of the stepsize $\eta$, such that $\max_{k=0,\dots,\lfloor T/\eta \rfloor} |\mathbb{E}g(x_k) - \mathbb{E}g(X_{k\eta})| \leq C\eta^\alpha$.

To see if an SDE satisfies such a definition, one has to check that for $\bar{\Delta} = x_1 - x$ and $\Delta = X_\eta - x$,

    1. $\left| \mathbb{E}\Delta_i - \mathbb{E}\bar{\Delta}_i \right| = \mathcal{O}(\eta^2), \quad \forall i = 1, \ldots, d;$

    2. $\left| \mathbb{E}\Delta_i \Delta_j - \mathbb{E}\bar{\Delta}_i \bar{\Delta}_j \right| = \mathcal{O}(\eta^2), \quad \forall i, j = 1, \ldots, d.$

**Example:** Let us prove that the SDE in Eq. (193) is a valid approximation of $\text{SGD}^{(\eta,B)}$: The first condition is easily verified. Coming to the second condition we have that

    1. $\mathbb{E}\Delta_i \Delta_j = \eta^2 \partial_i f(x) \partial_j f(x) + \frac{\eta^2}{B}\Sigma(x);$

    2. $\mathbb{E}\bar{\Delta}_i \bar{\Delta}_j = \eta^2 \partial_i f(x) \partial_j f(x) + \frac{\eta^2}{B}\Sigma(x) + \mathcal{O}(\eta^3);$

whose difference is of order $\eta^3$ and thus satisfies the condition. However, we observe that if the scale of the noise is too small w.r.t $\eta$, i.e. $\Sigma(x) = \mathcal{O}(\eta^\alpha)$ for $\alpha \geq 0$, then the **simplest** SDE model describing $\text{SGD}^{(\eta,B)}$ is the ODE $dX_t = -\nabla f(X_t)dt$ as in that case

    1. $\mathbb{E}\Delta_i \Delta_j = \eta^2 \partial_i f(x) \partial_j f(x) + \mathcal{O}(\eta^{2+\alpha});$

    2. $\mathbb{E}\bar{\Delta}_i \bar{\Delta}_j = \eta^2 \partial_i f(x) \partial_j f(x) + \mathcal{O}(\eta^2),$

whose difference is also of order $\eta^2$. Much differently, if $\Sigma(x) = \mathcal{O}(\eta^{-\alpha})$ for $\alpha > 0$, the simplest model is the SDE in Eq. (193). We highlight that *simplest* does not mean *best*: The SDE is more accurate than the ODE even in a regime with low noise, but this observation serves as a provocation. One has to pay attention when deriving SDEs: Some models are more realistic than others.

Let us dig deeper into this thought as we derive **two** SDEs for SGD with learning rate $\tilde{\eta} := \kappa\eta$ and batch size $\tilde{B} := \delta B$ for $\kappa > 1$ and $\delta > 1$, which we dub $\text{SGD}^{(\tilde{\eta},\tilde{B})}$. The first is derived considering that the learning rate is $\tilde{\eta}$ and carries an error of order $\mathcal{O}(\tilde{\eta})$ w.r.t. $\text{SGD}^{(\tilde{\eta},\tilde{B})}$

$$dX_t = -\nabla f(X_t)dt + \sqrt{\frac{\tilde{\eta}}{\tilde{B}}}\Sigma(X_t)^{\frac{1}{2}}dW_t = -\nabla f(X_t)dt + \sqrt{\frac{\eta\kappa}{B\delta}}\Sigma(X_t)^{\frac{1}{2}}dW_t. \tag{195}$$

The second one instead is derived considering $\eta$ as the learning rate and $\kappa$ as a constant "scheduler". Consistently with (Li et al., 2017), the SDE which carries an error of order $\mathcal{O}(\eta)$ w.r.t $\text{SGD}^{(\tilde{\eta},\tilde{B})}$ is

$$dX_t = -\kappa\nabla f(X_t)dt + \kappa\sqrt{\frac{\eta}{B\delta}}\Sigma(X_t)^{\frac{1}{2}}dW_t. \tag{196}$$

While they both are valid models, there are three reasons why one should prefer the latter:

    1. It fully reflects the fact that a larger learning rate results in a faster and noisier dynamics

    2. It has intrinsically less error than the other;

    3. It is consistent with the optimizer in that there is no combination of $\kappa$ and $\delta$ that can ever leave the dynamics unchanged.

## E.1 Deriving scaling rules

Jastrzebski et al. (2018) observed that only the ratio between $\eta$ and $B$ matters in determining the dynamics of Eq. (194). Therefore, they argue that for $\kappa = \delta$ the SDE for $\text{SGD}^{(\kappa\eta,\delta B)}$ coincides with that of $\text{SGD}^{(\eta,B)}$ and that this implies that the path properties of the optimizers are the same. On the contrary, the path of $\text{SGD}^{(\eta,B)}$ strongly depends on the hyperparameters: The speed and volatility of the dynamics are driven by $\eta$, and no choice of $B$ can undo this. We remind the reader that the goal of these rules is not to keep the dynamics of the optimizers unaltered, but rather to give a practical way to change a hyperparameter, e.g. $\eta$, and have a principled way to adjust the others, e.g. $B$, such that the performance of the optimizer is preserved. Therefore, we propose deriving scaling rules as we preserve certain relevant quantities of the dynamics such as the convergence bound on the expected loss or the speed. To show this quantitative, we use this rationale to derive the scaling rule of SGD as we aim at preserving the asymptotic loss level.

**Lemma E.2.** *If $f$ is a $\mu$ strongly convex function, $\mathcal{L}_\tau \leq Tr(\nabla^2 f(x))$ and $\Sigma(x) = \sigma^2 I_d$, then:*

*1. Under the dynamics of Eq. (193) we have:*

$$\mathbb{E}[f(X_t) - f(X_*)] \le (f(X_0) - f(X_*))e^{-2\mu t} + \frac{\eta}{2}\frac{\mathcal{L}_\tau \sigma^2}{2\mu B}\left(1 - e^{-2\mu t}\right); \qquad (197)$$

*2. Under the dynamics of Eq. (195) we have:*

$$\mathbb{E}[f(X_t) - f(X_*)] \le (f(X_0) - f(X_*))e^{-2\mu t} + \frac{\eta}{2}\frac{\mathcal{L}_\tau \sigma^2}{2\mu B}\frac{\kappa}{\delta}\left(1 - e^{-2\mu t}\right); \qquad (198)$$

*3. Under the dynamics of Eq. (196) we have:*

$$\mathbb{E}[f(X_t) - f(X_*)] \le (f(X_0) - f(X_*))e^{-2\mu\kappa t} + \frac{\eta}{2}\frac{\mathcal{L}_\tau \sigma^2}{2\mu B}\frac{\kappa}{\delta}\left(1 - e^{-2\mu\kappa t}\right). \qquad (199)$$

The first bound implies that the asymptotic limit of the expected loss for $\text{SGD}^{(\eta,B)}$ is $\frac{\eta}{2}\frac{\mathcal{L}_\tau \sigma^2}{2\mu B}$. The last two bounds predict that the asymptotic loss level for $\text{SGD}^{(\tilde\eta,\tilde B)}$ is $\frac{\eta}{2}\frac{\mathcal{L}_\tau \sigma^2}{2\mu B}\frac{\kappa}{\delta}$. Since the objective of the scaling rule is to find $\kappa$ and $\delta$ such that $\text{SGD}^{(\tilde\eta,\tilde B)}$ achieves the same loss level as $\text{SGD}^{(\eta,B)}$, we recover the linear scaling rule setting $\kappa = \delta$. However, only the last bound can correctly capture the fact that the dynamics of $\text{SGD}^{(\tilde\eta,\tilde B)}$ is $\kappa$ times faster than that of $\text{SGD}^{(\eta,B)}$.

We conclude the discussion with a simple sample of how deriving a scaling rule from the SDE itself inevitably leads to the wrong conclusion. We define the following algorithm which is inspired by AdamW and which we dub SGDW:

$$x_{k+1} = x_k - \eta \nabla f_{\gamma_k}(x_k) - \eta\gamma x_k. \qquad (200)$$

**Lemma E.3.** *The SDE of SGDW is*

$$dX_t = -\nabla f(X_t)dt + \sqrt{\frac{\eta}{B}}\Sigma(X_t)^{\frac{1}{2}}dW_t - \gamma X_t dt. \qquad (201)$$

Therefore, one would naively deduce that to keep the SDE unchanged, one can simply use the linear scaling rule of SGD and leave $\gamma$ unaltered. However, one can easily derive the upper bound on the expected loss for a convex quadratic function and observe that to preserve that, it is imperative to scale $\gamma$ by $\kappa$ as well.

We thus conclude that:

1. Eq. (196) is a better model for $\text{SGD}^{(\tilde\eta,\tilde B)}$ as it represents the dynamics more accurately;

2. Maintaining the shape of the SDE does not preserve the path properties of the optimizer;

3. Deriving a scaling rule uniquely from the SDE might lead to the wrong conclusions in the general case.

*Remark E.4.* We highlight that Theorem 5.3 of Malladi et al. (2022) claimed to have *formally* derived one for RMSprop: In line with (Jastrzebski et al., 2018), they argue that if they were to find a scaling rule that would leave their SDE unchanged, this would imply that even the dynamics of the iterates of RMSprop itself would be unchanged. First, we remind the reader that an SDE is formally defined as an *equation that drives the dynamics plus* an *initial condition* (See (Karatzas and Shreve, 2014), Section 5). While their scaling rule does leave the *equation unchanged*, it *alters the initial condition*, thus *changing the SDE* itself: This invalidates their claim and proof. Second, contrary to their claim, the rule is only valid near convergence as their SDE is only valid there. Third, Lemma E.2 offers a shred of concrete evidence that keeping the SDE unchanged does not imply that the path properties of the optimizers are preserved. Fourth, Lemma E.3 is a piece of concrete evidence that deriving scaling rules directly and naively from the SDE might lead to the wrong conclusions.

# F Experiments

In this section, we provide the modeling choices and instructions to replicate our experiments. All experiments we run on one NVIDIA GeForce RTX 3090 GPU. The code is implemented in Python 3 (Van Rossum and Drake, 2009) mainly using Numpy (Harris et al., 2020), scikit-learn (Pedregosa et al., 2011), and JAX (Bradbury et al., 2018).

### F.1 SignSGD: SDE validation (Figure 1)

In this subsection, we describe the experiments we run to produce Figure 1: The loss dynamics of SignSGD and that of our SDE match on average.

**DNN on Breast Cancer Dataset (Dua and Graff, 2017)**  This paragraph refers to the *left* of Figure 1. The DNN has 10 dense layers with 20 neurons each activated with a ReLu. We minimize the binary cross-entropy loss. We run SignSGD for $50000$ epochs as we calculate the full gradient and inject it with Gaussian noise $Z \sim \mathcal{N}(0, \sigma^2 I_d)$ where $\sigma = 1$. The learning rate is $\eta = 0.001$. Similarly, we integrate the SignSGD SDE (Eq. (7)) with Euler-Maruyama (Algorithm 1) with $\Delta t = \eta$. Results are averaged over 3 runs and the shaded areas are the average $\pm$ the standard deviation.

**CNN on MNIST (Deng, 2012)**  This paragraph refers to the *center-left* of Figure 1. The CNN has a $(3, 3, 32)$ convolutional layer with stride 1, followed by a ReLu activation, a $(2, 2)$ max pool layer with stride $(2, 2)$, a $(3, 3, 32)$ convolutional layer with stride 1, a ReLu activation, a $(2, 2)$ max pool layer with stride $(2, 2)$. Then the activations are flattened and passed through a dense layer that compresses them into $128$ dimensions, a final ReLu activation, and a final dense layer into the output dimension $10$. The output finally goes through a softmax as we minimize the cross-entropy loss. We run SignSGD for $40000$ epochs as we calculate the full gradient and inject it with Gaussian noise $Z \sim \mathcal{N}(0, \sigma^2 I_d)$ where $\sigma = 1$. The learning rate is $\eta = 0.001$. Similarly, we integrate the SignSGD SDE (Eq. (7)) with Euler-Maruyama (Algorithm 1) with $\Delta t = \eta$. Results are averaged over 3 run and the shaded areas are the average $\pm$ the standard deviation.

**Transformer on MNIST**  This paragraph refers to the *center-right* of Figure 1. The Architecture is a scaled-down version of (Dosovitskiy et al., 2021), where the hyperparameters are *patch size*=28, *out features*=10, *width*=48, *depth*=3, *num heads*=6, and *dim ffn*=192. We minimize the cross-entropy loss as we run SignSGD for $5000$ epochs as we calculate the full gradient and inject it with Gaussian noise $Z \sim \mathcal{N}(0, \sigma^2 I_d)$ where $\sigma = 1$. The learning rate is $\eta = 0.001$. Similarly, we integrate the SignSGD SDE (Eq. (7)) with Euler-Maruyama (Algorithm 1) with $\Delta t = \eta$. Results are averaged over 3 runs and the shaded areas are the average $\pm$ the standard deviation.

**ResNet on CIFAR-10 (Krizhevsky et al., 2009)**  This paragraph refers to the *right* of Figure 1. The ResNet has a $(3, 3, 128)$ convolutional layer with stride 1, followed by a ReLu activation, a second $(3, 3, 64)$ convolutional layer with stride 1, followed by a residual connection from the first convolutional layer, then a $(2, 2)$ max pool layer with stride $(2, 2)$. Then the activations are flattened and passed through a dense layer that compresses them into $128$ dimensions, a final ReLu activation, and a final dense layer into the output dimension $10$. The output finally goes through a softmax as we minimize the cross-entropy loss. We run SignSGD for $5000$ epochs as we calculate the full gradient and inject it with Gaussian noise $Z \sim \mathcal{N}(0, \sigma^2 I_d)$ where $\sigma = 1$. The learning rate is $\eta = 0.001$. Similarly, we integrate the SignSGD SDE (Eq. (7)) with Euler-Maruyama (Algorithm 1) with $\Delta t = \eta$. Results are averaged over 3 runs and the shaded areas are the average $\pm$ the standard deviation.

### F.2 SignSGD: insights validation (Figure 2)

In this subsection, we describe the experiments we run to produce Figure 2: We successfully validate them all.

**Phases: Lemma 3.4 and Lemma 3.5**  In this paragraph, we describe how we validated the existence of the phases of SignSGD as predicted in Lemma 3.4 and Lemma 3.5. To produce the *left* of Figure 2), we simulated the *full SDE* (Eq. (16)) and the one describing Phase 3 (Eq. (5)). The optimized function is $f(x) = \frac{x^\top H x}{2}$ for $H = \mathrm{diag}(1, 2)$, $x_0$ drawn (and fixed for all runs) from a normal distribution $\mathcal{N}(0, 0.01)$, $\eta = 0.001$, and $\Sigma = \sigma^2 I_d$ where $\sigma = 0.1$. We integrate the SDEs with Euler-Maruyama (Algorithm 1) with $\Delta t = \eta$ and for 3000 iterations. Results are averaged over $500$ runs and the shaded areas are the average $\pm$ the standard deviation. Clearly, the two SDEs share the same dynamics.

To produce the *center-left* of Figure 2, we repeat the above as $x_0$ drawn (and fixed for all runs) from a normal distribution $\mathcal{N}(0, 1)$. Then, we plot the average loss values together with the theoretical prediction of Phase 1 and Phase 3: They perfectly overlap.

**Stationary distribution: Lemma 3.7** In this paragraph, we describe how we validated the convergence behavior predicted in Lemma 3.7. To produce the *center-right* of Figure 2), we run SignSGD on $f(x) = \frac{x^\top H x}{2}$ for $H = \mathrm{diag}(1, 2)$, $x_0 = (0.001, 0.001)$, $\eta = 0.001$ and $\Sigma = \sigma^2 I_d$ where $\sigma = 0.1$. We run this for 5000 times and report the evolution of the moments. Then, we add lines representing the theoretical predictions derived in Lemma 3.7: They match.

**Schedulers: Lemma 3.9** In this paragraph, we describe how we validated the convergence behavior predicted in Lemma 3.9. To produce the *right* of Figure 2, we run SignSGD on $f(x) = \frac{x^\top H x}{2}$ for $H = \mathrm{diag}(1, 2)$, $x_0 = (0.01, 0.01)$, $\eta = 0.01$ and $\Sigma = \sigma^2 I_d$ where $\sigma = 0.1$. We used the scheduler $\eta_t^\gamma = \frac{1}{(t+1)^\gamma}$ for $\gamma \in \{0.1, 0.5, 1.5\}$. For the first two choices of $\gamma$, $\eta_t^\gamma$ satisfies our sufficient condition for the convergence of SignSGD: In the figure, we observe that indeed SignSGD converges to 0 with the same speed as the one predicted in the Lemma. For $\gamma = 1.5$, we observe that SignSGD does not converge following the theoretical curve because it does not satisfy our sufficient condition. Results are averaged over 500 runs.

### F.3 RMSprop: SDE validation (Figure 7 and Figure 8)

In this subsection, we describe the experiments we run to produce Figure 7 and Figure 8: The dynamics of our SDE matches that of RMSprop better than the SDE derived in (Malladi et al., 2022).

**Quadratic convex function** This paragraph refers to the *left* and *center-left* of Figure 7. We optimize the function $f(x) = \frac{x^\top H x}{2}$ where $H = \mathrm{diag}(10, 2)$. We run RMSprop for 2000 epochs as we calculate the full gradient and inject it with Gaussian noise $Z \sim \mathcal{N}(0, \sigma^2 I_d)$ where $\sigma = 0.1$. The learning rate is $\eta = 0.01$, $\beta = 0.99$. Similarly, we integrate our RMSprop SDE (Eq. (86)) and that of Malladi (Eq. (183)) with Euler-Maruyama (Algorithm 1) with $\Delta t = \eta$. Results are averaged over 500 runs and the shaded areas are the average $\pm$ the standard deviation: Our SDE matches RMSprop much better.

**Embedded saddle** This paragraph refers to the *center-right* and *right* of Figure 7. We optimize the function $f(x) = \frac{x^\top H x}{2} + \frac{1}{4}\lambda \sum_{i=1}^2 x_i^4 - \frac{\xi}{3} \sum_{i=1}^2 x_i^3$ where $H = \mathrm{diag}(-1, 2)$, $\lambda = 1$, and $\xi = 0.1$. We run RMSprop for 1600 epochs as we calculate the full gradient and inject it with Gaussian noise $Z \sim \mathcal{N}(0, \sigma^2 I_d)$ where $\sigma = 0.01$. The learning rate is $\eta = 0.01$, $\beta = 0.99$. Similarly, we integrate our RMSprop SDE (Eq. (86)) and that of Malladi (Eq. (183)) with Euler-Maruyama (Algorithm 1) with $\Delta t = \eta$. Results are averaged over 500 runs and the shaded areas are the average $\pm$ the standard deviation: Our SDE matches RMSprop much better.

**DNN on Breast Cancer Dataset** This paragraph refers to the *left* of Figure 8. The architecture and loss are the same as used above for SignSGD. We run RMSprop for 2000 epochs as we calculate the full gradient and inject it with Gaussian noise $Z \sim \mathcal{N}(0, \sigma^2 I_d)$ where $\sigma = 10^{-2}$. The learning rate is $\eta = 10^{-4}$, $\beta = 0.9995$. Similarly, we integrate our RMSprop SDE (Eq. (86)) and that of Malladi (Eq. (183)) with Euler-Maruyama (Algorithm 1) with $\Delta t = \eta$. Results are averaged over 3 runs and the shaded areas are the average $\pm$ the standard deviation: Our SDE matches RMSprop much better.

**CNN on MNIST** This paragraph refers to the *center-left* of Figure 8. The architecture and loss are the same as used above for SignSGD. We run RMSprop for 2000 epochs as we calculate the full gradient and inject it with Gaussian noise $Z \sim \mathcal{N}(0, \sigma^2 I_d)$ where $\sigma = 10^{-2}$. The learning rate is $\eta = 10^{-3}$, $\beta = 0.995$. Similarly, we integrate our RMSprop SDE (Eq. (86)) and that of Malladi (Eq. (183)) with Euler-Maruyama (Algorithm 1) with $\Delta t = \eta$. Results are averaged over 3 run and the shaded areas are the average $\pm$ the standard deviation: Our SDE matches RMSprop much better.

**Transformer on MNIST** This paragraph refers to the *center-right* of Figure 8. The architecture and loss are the same as used above for SignSGD. We run RMSprop for 2000 epochs as we calculate the full gradient and inject it with Gaussian noise $Z \sim \mathcal{N}(0, \sigma^2 I_d)$ where $\sigma = 10^{-2}$. The learning rate is $\eta = 10^{-3}$, $\beta = 0.995$. Similarly, we integrate our RMSprop SDE (Eq. (86)) and that of Malladi (Eq. (183)) with Euler-Maruyama (Algorithm 1) with $\Delta t = \eta$. Results are averaged over 3 runs and the shaded areas are the average $\pm$ the standard deviation: Our SDE matches RMSprop much better.

**ResNet on CIFAR-10** This paragraph refers to the *right* of Figure 8. The architecture and loss are the same as used above for SignSGD. We run RMSprop for 500 epochs as we calculate the full gradient and inject it with Gaussian noise $Z \sim \mathcal{N}(0, \sigma^2 I_d)$ where $\sigma = 10^{-4}$. The learning rate is $\eta = 10^{-4}$, $\beta = 0.9999$. Similarly, we integrate our RMSprop SDE (Eq. (86)) and that of Malladi (Eq. (183)) with Euler-Maruyama (Algorithm 1) with $\Delta t = \eta$. Results are averaged over 3 runs and the shaded areas are the average $\pm$ the standard deviation: Our SDE matches RMSprop much better.

### F.4   Adam: SDE validation (Figure 10 and Figure 11)

In this subsection, we describe the experiments we run to produce Figure 11 and Figure 10: The dynamics of our SDE matches that of Adam better than that derived in (Malladi et al., 2022).

**Quadratic convex function** This paragraph refers to the *left* and *center-left* of Figure 10. We optimize the function $f(x) = \frac{x^\top H x}{2}$ where $H = \mathrm{diag}(10, 2)$. We run Adam for 50000 epochs as we calculate the full gradient and inject it with Gaussian noise $Z \sim \mathcal{N}(0, \sigma^2 I_d)$ where $\sigma = 0.01$. The learning rate is $\eta = 0.001$, $\beta_1 = 0.9$, and $\beta_2 = 0.999$. Similarly, we integrate our Adam SDE (Eq. (124)) and that of Malladi (Eq. (188)) with Euler-Maruyama (Algorithm 1) with $\Delta t = \eta$. Results are averaged over 500 runs and the shaded areas are the average $\pm$ the standard deviation: Our SDE matches Adam much better.

**Embedded saddle** This paragraph refers to the *center-right* and *right* of Figure 10. We optimize the function $f(x) = \frac{x^\top H x}{2} + \frac{1}{4}\lambda \sum_{i=1}^{2} x_i^4 - \frac{\xi}{3} \sum_{i=1}^{2} x_i^3$ where $H = \mathrm{diag}(-1, 2)$, $\lambda = 1$, and $\xi = 0.1$. We run Adam as we calculate the full gradient and inject it with Gaussian noise $Z \sim \mathcal{N}(0, \sigma^2 I_d)$ where $\sigma = 0.1$. The learning rate is $\eta = 0.001$, $\beta_1 = 0.9$, and $\beta_2 = 0.999$. Similarly, we integrate our Adam SDE (Eq. (124)) and that of Malladi (Eq. (188)) with Euler-Maruyama (Algorithm 1) with $\Delta t = \eta$. Results are averaged over 500 runs and the shaded areas are the average $\pm$ the standard deviation: Our SDE matches Adam much better.

**DNN on Breast Cancer Dataset** This paragraph refers to the *left* of Figure 11. The architecture and loss are the same as used above for SignSGD. We run Adam for 2000 epochs as we calculate the full gradient and inject it with Gaussian noise $Z \sim \mathcal{N}(0, \sigma^2 I_d)$ where $\sigma = 10^{-2}$. The learning rate is $\eta = 10^{-4}$, $\beta_1 = 0.99$, and $\beta_2 = 0.999$. Similarly, we integrate our Adam SDE (Eq. (124)) and that of Malladi (Eq. (188)) with Euler-Maruyama (Algorithm 1) with $\Delta t = \eta$. Results are averaged over 3 runs and the shaded areas are the average $\pm$ the standard deviation: Our SDE matches Adam much better.

**CNN on MNIST** This paragraph refers to the *center-left* of Figure 11. The architecture and loss are the same as used above for SignSGD. We run Adam for 2000 epochs as we calculate the full gradient and inject it with Gaussian noise $Z \sim \mathcal{N}(0, \sigma^2 I_d)$ where $\sigma = 10^{-2}$. The learning rate is $\eta = 10^{-2}$, $\beta_1 = 0.9$, and $\beta_2 = 0.99$. Similarly, we integrate our Adam SDE (Eq. (124)) and that of Malladi (Eq. (188)) with Euler-Maruyama (Algorithm 1) with $\Delta t = \eta$. Results are averaged over 3 runs and the shaded areas are the average $\pm$ the standard deviation: Our SDE matches Adam much better.

**Transformer on MNIST** This paragraph refers to the *center-right* of Figure 11. The architecture and loss are the same as used above for SignSGD. We run Adam for 2000 epochs as we calculate the full gradient and inject it with Gaussian noise $Z \sim \mathcal{N}(0, \sigma^2 I_d)$ where $\sigma = 10^{-2}$. The learning rate is $\eta = 10^{-2}$, $\beta_1 = 0.9$, and $\beta_2 = 0.99$. Similarly, we integrate our Adam SDE (Eq. (124)) and that of Malladi (Eq. (188)) with Euler-Maruyama (Algorithm 1) with $\Delta t = \eta$. Results are averaged over 3 runs and the shaded areas are the average $\pm$ the standard deviation: Our SDE matches Adam much better.

**ResNet on CIFAR-10** This paragraph refers to the *right* of Figure 11. The architecture and loss are the same as used above for SignSGD. We run Adam for 2000 epochs as we calculate the full gradient and inject it with Gaussian noise $Z \sim \mathcal{N}(0, \sigma^2 I_d)$ where $\sigma = 10^{-5}$. The learning rate is $\eta = 10^{-5}$, $\beta_1 = 0.99$, and $\beta_2 = 0.9999$. Similarly, we integrate our Adam SDE (Eq. (124)) and that of Malladi (Eq. (188)) with Euler-Maruyama (Algorithm 1) with $\Delta t = \eta$. Results are averaged over 3 runs and the shaded areas are the average $\pm$ the standard deviation: Our SDE matches Adam much better.

### F.5 RMSpropW & AdamW: SDE validation (Figure 3, Figure 4)

The settings are exactly the same as those for RMSprop and Adam. The regularization parameter used is always $\gamma = 0.01$. We observe that our SDEs match the respective algorithm with a good agreement.

### F.6 RMSpropW & AdamW: insights validation (Figure 5)

In this subsection, we describe the experiments we run to produce Figure 5: The theoretically predicted asymptotic loss value and moments of RMSpropW and AdamW match those empirically found.

**Asymptotic loss & scaling rule of AdamW**  This paragraph refers to the *left* of Figure 5. We optimize the function $f(x) = \frac{x^\top H x}{2}$ where $H = \mathrm{diag}(1, 3)$. We run AdamW for 20000 epochs as we calculate the full gradient and inject it with Gaussian noise $Z \sim \mathcal{N}(0, \sigma^2 I_d)$ where $\sigma = 1$. The learning rate is $\eta = 0.001$, $\beta_1 = 0.9$, and $\beta_2 = 0.999$. Experiments are run for both $\gamma = 1$ and $\gamma = 4$. The rescaled versions of the algorithms *AdamW R* follow the novel scaling rule with $\kappa = 2$. *AdamW NR* follows the scaling rule but not for $\gamma$ which is left unchanged. We plot the evolution of the loss values with the theoretical predictions of Lemma C.28: Results are averaged over 500 runs.

**Asymptotic loss & scaling rule of RMSpropW**  This paragraph refers to the *center-left* of Figure 5: The only difference with the previous paragraph is that we use RMSpropW with $\beta = 0.999$.

**AdamW: the role of the $\beta$s**  This paragraph refers to the *center-right* of Figure 5. We optimize the function $f(x) = \frac{x^\top H x}{2} + \frac{1}{4}\lambda \sum_{i=1}^2 x_i^4 - \frac{\xi}{3}\sum_{i=1}^2 x_i^3$ where $H = \mathrm{diag}(-1, 2)$, $\lambda = 1$, and $\xi = 0.1$. We run AdamW as we calculate the full gradient and inject it with Gaussian noise $Z \sim \mathcal{N}(0, \sigma^2 I_d)$ where $\sigma = 0.1$. The learning rate is $\eta = 0.001$, $\gamma = 0.1$, $\beta_1 \in \{0.99, 0.999\}$, and $\beta_2 \in \{0.992, 0.996, 0.998\}$: Clearly, three combinations go into a minimum and three go into the other. For each minimum, the three optimizers converge to the same asymptotic loss value independently on the values of $\beta_1$ and $\beta_2$. We argue that $\beta_1$, and $\beta_2$ select the basin and the speed of convergence, not the asymptotic loss value: This is consistent with Lemma 3.13.

**Stationary distribution**  This paragraph refers to the *right* of Figure 5. We optimize the function $f(x) = \frac{x^\top H x}{2}$ where $H = \mathrm{diag}(1, 3)$. We run Adam for 20000 epochs as we calculate the full gradient and inject it with Gaussian noise $Z \sim \mathcal{N}(0, \sigma^2 I_d)$ where $\sigma = 0.01$. The learning rate is $\eta = 0.001$, $\gamma = 4$, $\beta = 0.999$, $\beta_1 = 0.9$, and $\beta_2 = 0.999$. We plot the evolution of the average variances with the theoretical predictions of Lemma C.24 and Lemma 3.14: Results are averaged over 100 runs.

### F.7 Effect of noise - validation (Figure 6)

In this subsection, we describe the experiments run to produce Figure 6: All bounds on the asymptotic expected loss value for SGD, SignSGD, Adam, and AdamW are perfectly verified.

We optimize the loss $f(x) = \frac{x^\top H x}{2}$ where $H = \mathrm{diag}(1, 1)$ as we run each optimizer for 100000 iterations with $\eta = 0.01$. We repeat this procedure five times, one for each $\sigma \in \{0.01, 0.1, 1, 10, 100\}$. As we train, we inject noise on the gradient as distributed as $\mathcal{N}(0, \sigma^2 I_d)$. We plot the average loss together with the respective limits predicted by our Lemmas. For each optimizer and each $\sigma$, the average asymptotic loss matches the predicted limit. Therefore, we verify that the loss of SGD scales quadratically in $\sigma$, that of Adam and SignSGD scales linearly, and that of AdamW is limited in $\sigma$.

### F.8 Increasing weight decay with the batch size

The analysis of Malladi et al. (2022) suggests that, when scaling batch size $B$ by a factor $\kappa$ one has to scale up ($\uparrow$) the learning rate $\eta$ by a factor $\sqrt{\kappa}$ and scale down ($\downarrow$) $\beta_2$ to the value $1 - \kappa(1 - \beta_2)$. Our SDE analysis confirms similar rules (Lemma 3.13) but additionally suggests scaling up the decoupled weight decay parameter $\gamma$ by a factor $\sqrt{\kappa}$. We test this in two settings: VGG11 and ResNet34 (convolutional networks) on CIFAR-10 classification. We select a base batch size of 256,

and run AdamW with $\eta = 0.001$, $\beta_2 = 0.99$, and $\gamma = 0.1$. We consider scaling the batch by a factor 4: In Table 1, we show the effect of updating each hyperparameter with the proposed rule and we denote by a "·" the model parameters of the base run with $B = 256$. We train for 150 epochs the model with $B = 256$, and $150 \times 4$ the model with $B = 4 \times 256$. Experiments are repeated 3 times. We find that, while improvements are marginal, they are consistent with our theoretical results.

| $B$ | $\eta$ | $\beta_2$ | $\lambda$ | VGG11 (Test Acc ↑) | ResNet 34 (Test Acc ↑) |
|---|---|---|---|---|---|
| · | · | · | · | $90.581 \pm 0.295$ | $94.396 \pm 0.126$ |
| ↑ | · | · | · | $90.502 \pm 0.093$ | $94.296 \pm 0.220$ |
| ↑ | ↑ | · | · | $90.767 \pm 0.119$ | $94.507 \pm 0.148$ |
| ↑ | ↑ | ↓ | · | $90.703 \pm 0.271$ | $94.590 \pm 0.188$ |
| ↑ | ↑ | ↓ | ↑ | $\mathbf{90.966 \pm 0.252}$ | $\mathbf{94.639 \pm 0.192}$ |

Table 1: Scaling with the batch size: Effect of adapting AdamW hyperparameters.

