# OpenReview forum: "SDEs for Adaptive Methods: The Role of Noise"
_NeurIPS.cc/2024/Conference — Submitted to NeurIPS 2024_

### Official Review · Reviewer_Pw4R · 2024-06-23

**Soundness:** 3
**Presentation:** 3
**Contribution:** 3
**Rating:** 6
**Confidence:** 4

**Summary:**

This work derives SDEs for adaptive gradient methods and study the role of gradient noise. The analysis starts from theoretically driving the SDE for SignSGD and highlight its significant difference from SGD. The work further generalize the SDE analysis for AdamW and RMSpropW, two popular adaptive optimizers with decoupled weight decay and reveal key properties of weight decay. Finally, the work integrates the derived SDEs with Euler-Maruyama to confirm that the SDEs faithfully track their respective optimizers with various modern neural networks.

**Strengths:**

-The theoretical results are novel. To my knowledge, this is a first SDE analysis for SignSGD with quantitatively accurate descriptions.

-The theoretical analysis reports some novel properties in terms of gradient noise and convergence. These properties are interesting.

-The proofs seem complete and reasonable.

-A useful theory should be quantitatively verifiable. This work definitely make it. The experiments that SDEs fit the empirical results with various optimizers and models are informative and impressive.

**Weaknesses:**

-It seems that the reported theoretical results and insights cannot directly lead to some theory-inspired and improved methods. This raise a question on the significance of this work.

-While this work did literature review, some important references are still missing, such as [1] on analyzing Adam using SDEs. As weight decay plays a key role in the results, it may be helpful to review recent papers analyzing novel or overlooked properties of weight decay.


Reference:

[1] Xie, Z., Wang, X., Zhang, H., Sato, I., & Sugiyama, M. (2022, June). Adaptive inertia: Disentangling the effects of adaptive learning rate and momentum. In International conference on machine learning (pp. 24430-24459). PMLR.

**Questions:**

- Please see the weaknesses.

- Could you please explain more how L2 regularization and decoupled weight decay behaves differently in your results?

**Limitations:**

This work discussed the limitations in the appendix.

---

> ### Author Rebuttal · Authors · 2024-08-05
>
> We sincerely thank the Reviewer for the significant effort put into this review: We appreciate the acknowledgement of the value of our research. We thank you for the questions as they stimulated us to include some more references and dig deeper to showcase the explanatory power of our SDEs even more.
>
> **Weakness 1:**
>
> "*It seems that the reported theoretical results and insights cannot directly lead to some theory-inspired and improved methods. This raise a question on the significance of this work.*"
>
> **Answer:**
>
> We acknowledge that our work has limitations in terms of developing improved methods. However, we aimed to offer new insights into existing adaptive methods that are known to perform well in practice, even though the reasons for their effectiveness are not yet fully understood. We respectfully believe that, from this perspective, our work holds significant value and is of interest to the community.
>
> **Weakness 2:**
>
> *"While this work did literature review, some important references are still missing, such as [1] on analyzing Adam using SDEs. As weight decay plays a key role in the results, it may be helpful to review recent papers analyzing novel or overlooked properties of weight decay."*
>
> **Answer:**
>
> We thank the Reviewer for reminding us about this interesting paper, which we are familiar with but unfortunately forgot to cite. Rather than studying the role of noise on the dynamics of Adam, their focus is mainly on disentangling the effects of learning rate adaptivity and momentum on saddle-point escaping and flat minima selection. They use the SDE to study how Momentum helps SGD escape saddle points and minima. Analogously, they repeat the analysis for Adam and find that learning rate adaptivity helps to escape saddle points but leads to sharper minima than SGD. Inspired by their results (Thm.2 and Thm.3), they propose Adai (Adaptive Inertia Optimization), which rather than opting for adaptivity of the learning rate, it opts for adaptivity of the momentum parameters: They theoretically predict and experimentally validate that Adai is both fast at escaping saddles and successful at finding flat minima.
>
> As highlighted by Malladi et al., the SDE presented in [1] is not derived within any formal framework and therefore does not come with formal approximation guarantees. However, this is a very insightful and valuable work that we will cite in the final version of the paper. We are of course happy to know if there is a specific point about [1] that we should pay attention to, or if other important references have been missed.
>
> Regarding Weight Decay, we kindly request that the Reviewer provide us with any specific references they have in mind.
>
> **Question 2:**
>
> *"Could you please explain more how L2 regularization and decoupled weight decay behaves differently in your results?"*
>
> **Answer:**
>
> This is a very interesting question: Please, find below the SDE induced by using Adam on an $L^2$-regularized loss together with the equivalent of Lemma 3.13. Most importantly, we observe the $L^2$ regularization used in this way does not provide additional resilience against noise w.r.t. Adam: The asymptotic loss level scales linearly in the noise $\sigma$ exactly as it does for Adam. On the contrary, when $L^2$ regularization is used in a decoupled way as in AdamW, the asymptotic loss level is upper-bounded in $\sigma$.
>
> When Adam is used to optimize the $L^2$-regularized loss $f(x) + \frac{\gamma\lVert x\rVert_2^2}{2}$ for $\gamma>0$, the SDE of the method is:
>
> \begin{equation}
>     d X_t  =-\frac{\sqrt{\gamma_2(t)}}{\gamma_1(t)} P_t^{-1} (M_t + \eta \rho_1 \left(\nabla f\left(X_t\right)-M_t\right) - \gamma X_t) d t
> \end{equation}
> \begin{equation}
>     d M_t =\rho_1\left(\nabla f\left(X_t\right)-M_t\right) d t+\sqrt{\eta} \rho_1 \Sigma^{1 / 2}\left(X_t\right) d W_t
> \end{equation}
> \begin{equation}
>     d V_t =\rho_2\left( (\nabla f(X_t))^2 +  diag\left(\Sigma\left(X_t\right)\right)-V_t\right) d t,
> \end{equation}
> where $\beta_i = 1 - \eta \rho_i$, $\gamma_i(t) = 1 - e^{-\rho_i t}$, $\rho_1 = \mathcal{O}(\eta^{-\zeta})$ s.t. $\zeta \in (0,1)$, $\rho_2 = \mathcal{O}(1)$, and $P_t = diag{\sqrt{V_t}} + \epsilon \sqrt{\gamma_2(t)}I_d$.
>
> Under the same assumptions of Lemma 3.5, the dynamics of Adam on a $L^2$-regularized loss implies that
> \begin{equation}
>     \mathbb{E}[f(X_t) - f(X_*)] \overset{t \rightarrow \infty}{\leq} \frac{\eta \mathcal{L}_\tau \sigma }{2} \frac{L}{2 \mu L + \gamma (L + \mu)},
> \end{equation}
> meaning that the asymptotic loss level grows linearly in $\sigma$ as it already does for Adam.
>
> Much differently, the asymptotic loss level for AdamW is
> \begin{equation}
>     \mathbb{E}[f(X_t) - f(X_*)] \overset{t \rightarrow \infty}{\leq} \frac{\eta \mathcal{L}_\tau \sigma }{2} \frac{L}{2 \mu L + \sigma \gamma (L + \mu)},
> \end{equation}
> which is upper-bounded in $\sigma$.
>
> **Please, find an empirical validation of this bound in Figure 2 of the attached .pdf file.**

---

> > ### Comment · Reviewer_Pw4R · 2024-08-11
> > **Thanks for the rebuttal.**
> >
> > Thanks for the rebuttal and addressing some of the concerns.
> >
> > I will keep the rating as 6. I tend to accept this work.

---

> > > ### Author Response · Authors · 2024-08-11
> > >
> > > Dear Reviewer,
> > >
> > > Thank you for your response.
> > >
> > > We are glad to know that some of your concerns have been resolved: Could you please share any remaining issues or suggestions you might have? Your feedback is invaluable and will assist us in refining our manuscript further.
> > >
> > > We appreciate your time and consideration.
> > >
> > > Best regards,
> > >
> > > The Authors

---

### Official Review · Reviewer_PBh6 · 2024-07-13

**Soundness:** 3
**Presentation:** 2
**Contribution:** 2
**Rating:** 6
**Confidence:** 2

**Summary:**

The authors derive SDE for signSGD and Adam(W). The experiments show that the algorithm will converge toward the limit of the theorem indicates.

**Strengths:**

The authors propose "accurate" SDEs for algorithms Sign-SGD and Adam(W).

**Weaknesses:**

1. In Remark after Lemma 3.6, the authors claim that Sign-SGD is (almost) linear in $\sigma_{max}$. However, with $\Delta$ either in Phase 2 or Phase 3, there should be $\sigma_{max}^2$ in the final bound.

2. All the stationarity holds when Hessian is the same from $X_0$ to $X_t$ and convergence holds for strongly convex. However, the hessian changes a lot during network training.

**Questions:**

1. Notation of $W_t$ is not defined.  What is $W_t$?

2. How can we extend Lemma 3.13 to convex setting (or even nonconvex case)?

---

> ### Author Rebuttal · Authors · 2024-08-05
>
> We sincerely thank the Reviewer: We appreciate the questions as they stimulated us to clarify certain aspects and dig deeper to showcase the explanatory power of our SDEs even more.
>
> **Weakness 1:**
>
> *"In Remark after Lemma 3.6, the authors claim that Sign-SGD is (almost) linear in $\sigma_{\text{max}}$. However, with $\Delta$ either in Phase 2 or Phase 3, there should be $\sigma_{\text{max}}^2$ in the final bound."*
>
> **Answer:**
>
> We fully agree with this observation, which is why we say that the dependence is "*almost linear*" in $\sigma_{\text{max}}$. We can rewrite the asymptotic loss level as:
>
> \begin{equation}
>    \frac{\eta}{2} \frac{\mathcal{L}_{\tau}}{ 2 \mu } \frac{1}{\Delta},
> \end{equation}
>
> and observe that
>
> \begin{equation}
>     \frac{1}{\Delta} =\frac{\pi  \sigma_{\text{max}}^2 }{\sqrt{2 \pi} \sigma_{\text{max}} + \eta \mu} = \frac{\pi  \sigma_{\text{max}} }{\sqrt{2 \pi}  + \frac{\eta \mu}{\sigma_{\text{max}}}}.
> \end{equation}
>
> Therefore, when the noise $\sigma_{\text{max}}$ dominates over the learning rate $\eta$ and/or over the minimum eigenvalue $\mu$ of the Hessian, or more in general when $\frac{\eta \mu}{\sigma_{\text{max}}} \sim 0$, we can conclude that the behavior is essentially linear in $\sigma_{\text{max}}$: We will most certainly clarify this aspect better in the final version of the paper.
>
> **Weakness 2:**
>
> *"All the stationarity holds when Hessian is the same from $X_0$ to $X_t$ and convergence holds for strongly convex. However, the hessian changes a lot during network training."*
>
> **Answer:**
>
> We agree that the Hessian of the loss function can change dramatically during training. However, as we specify in Line 186, we are not studying the properties of the iterates during training, but rather characterize the stationary distribution around minima: These are the only points where the optimizer can reach stationarity and possibly stop.
> With this in mind, as we specify in Lines 125 to 128, it is common in the literature to approximate the loss function with a quadratic function in a neighborhood around these points. Therefore, the Hessian is constant in this neighborhood.
> These two reasons justify why we only study the stationary distribution of SignSGD in Phase 3 for a quadratic loss function. We also add that whatever happens before Phase 3 does not influence what happens at convergence, e.g. the stationary distribution.
>
> In response to the second part of your comment, we have strengthened our convergence analysis beyond the strongly convex case. Specifically, we extended Lemma 3.5 to the general smooth non-convex case (i.e. only requiring $L$-smoothness):
>
> Let $f$ be $L$-smooth, $\eta_t$ be a learning rate scheduler such that $\lim_{t \rightarrow \infty} \frac{\phi_t^2}{\phi^1_t} \overset{t \rightarrow \infty}{\rightarrow} 0$ and $\phi^1_t \overset{t \rightarrow \infty}{\rightarrow} \infty$, where $\phi^i_t = \int_0^t (\eta_s)^i ds$. Then, during
> 1. Phase 1, $\lVert \nabla f\left(X_{\tilde{t}^1}\right)\rVert_1 \leq \frac{f(X_0) - f(X_*)}{\phi_t^1} \overset{t \rightarrow \infty}{\rightarrow} 0$;
> 2. Phase 2, $$ \left( \frac{m}{\sqrt{2}}\mathbb{E} \lVert \nabla f\left(X_{\tilde{t}^{(1,2)}}\right)\rVert_2^2 + \hat{q} \sigma_{\text{max}} \mathbb{E} \lVert \nabla f\left(X_{\tilde{t}^{(2,2)}}\right)\rVert_1 \right) \leq \sigma_{\text{max}} \left( \frac{f(X_0) - f(X_*)}{\phi^1_t} + \frac{\eta L d}{2} \frac{\phi_t^2}{\phi^1_t} \right) \overset{t \rightarrow \infty}{\rightarrow} 0;$$
> 3. Phase 3, $\mathbb{E} \lVert \nabla f\left(X_{\tilde{t}^3}\right)\rVert_2^2 \leq \sqrt{\frac{\pi}{2}} \frac {\sigma_{\text{max}} \eta L d}{2} \frac{\phi_t^2}{\phi^1_t} + \sqrt{\frac{\pi}{2}} \sigma_{\text{max}}  \frac{f(X_0) - f(X_*)}{\phi^1_t} \overset{t \rightarrow \infty}{\rightarrow} 0$;
> where $\tilde{t}^1$, $\tilde{t}^{(1,2)}$, $\tilde{t}^{(2,2)}$, and $\tilde{t}^3$ are random times with distribution $\frac{\eta_t}{\phi^1_t}$.
>
>
> Interestingly, in Phase 1, SignSGD implicitly minimizes the $L^1$-norm of the gradient, in Phase 2 it implicitly minimizes a linear combination of norm $L^1$ and $L^2$, and in Phase 3 it implicitly minimizes the norm $L^2$: This result is novel as well and we thank the Reviewer for asking this great question.
>
> **Question 1:**
>
> *"Notation of $W_t$ is not defined. What is $W_t$?"*
>
> **Answer:**
>
> We apologize for not defining this symbol in the main paper.
> $W_t$ is the Brownian motion and we will specify this clearly in the final version of the paper. Importantly, we highlight that we included a whole chapter on Stochastic Calculus in Appendix B.
>
> **Question 2:**
>
> *"How can we extend Lemma 3.13 to convex setting (or even nonconvex case)?"*
>
> **Answer:**
>
> Due to some technical issues on AdamW that we will address in the future, we now put forward a generalization of Lemma 3.13 for Adam where we only require $L$-smoothness:
>
>
> Let $f$ be $L$-smooth, $\eta_t$ be a learning rate scheduler such that $\lim_{t \rightarrow \infty} \frac{\phi_t^2}{\phi^1_t} \overset{t \rightarrow \infty}{\rightarrow} 0$ and $\phi^1_t \overset{t \rightarrow \infty}{\rightarrow} \infty$, where $\phi^i_t = \int_0^t (\eta_s)^i ds$. Then
>
> \begin{equation}
> \mathbb{E} \lVert \nabla f \left(X_{\tilde{t}} \right) \rVert_2^2 \leq \left[ f(X_0) - f(X_*) + \mathcal{L}_{\tau} \left( \frac{ \delta B}{\rho_1^2 \sigma^2} \frac{\lVert M_0 \rVert_2^2}{2} + \frac{\phi^2_t \eta \kappa^2}{2} \right) \right] \frac{\sigma}{\kappa \sqrt{\delta B}} \frac{1}{\phi^1_t} \overset{t \rightarrow \infty}{\rightarrow} 0
> \end{equation}
>
> where $\tilde{t}$ is a random time with distribution $\frac{\eta_t}{\phi^1_t}$.

---

> > ### Comment · Reviewer_PBh6 · 2024-08-12
> >
> > Thanks for the authors' response. I have no further questions. Since the authors claim to add clarification in the final version, I raise my score to 6.

---

> > > ### Author Response · Authors · 2024-08-12
> > > **Thanks!**
> > >
> > > Dear Reviewer,
> > >
> > > Thank you for your trust and the updated score: We truly appreciate it.
> > >
> > > Best regards,
> > >
> > > The Authors

---

### Official Review · Reviewer_ehSv · 2024-07-23

**Soundness:** 3
**Presentation:** 2
**Contribution:** 3
**Rating:** 7
**Confidence:** 4

**Summary:**

This paper derives SDEs for SignSGD, RMSprop, and Adam.
The analysis offers insights into the convergence speed, stationary distribution, and robustness to heavy-tail noise of adaptive methods.

**Strengths:**

- The derived SDE for SignSGD exhibits three different phases of the dynamics.

- The analysis reveals the difference between SignSGD and SGD in terms of the asymptotic expected loss, the robustness of noise variance, etc.

- The analysis of AdamW provides insights into the different roles of noise, curvature, and weight decay.

**Weaknesses:**

Refer to Questions and Limitations.

**Questions:**

- What learning rate (lr) do the experiments in Figure 4 use? Within what range of lr does this SDE align well with the original algorithm (experimentally)?

- Could the authors intuitively explain why the asymptotic expected loss of SignSGD is proportional to $\sigma_{\max}$ instead of $\sigma_{\max}^2$?

- How can the derived SDE explain the loss spike phenomenon of SignSGD/AdamW?

- Many works about SGD noise ([1][2][3]) admit the noise structure $\mathbb{E}(g_i-g)(g_i-g)^{\top}\sim \mathcal{L} H$. What conclusions (such as those related to the training phases) can be derived from the SDE if we change the noise assumption in Corollary 3.3 to $\mathbb{E}(g_i-g)(g_i-g)^{\top}\sim\mathcal{L} H$?


[1] Ziyin et al. Strength of minibatch noise in SGD.

[2] Wojtowytsch. Stochastic gradient descent with noise of machine learning type. part II: Continuous time analysis.

[3] Wu et al. The alignment property of SGD noise and how it helps select flat minima: A stability analysis.

**Limitations:**

The SDE for AdamW is limited to quadratic functions.

---

> ### Author Rebuttal · Authors · 2024-08-05
>
> We thank the Reviewer for their thorough and thoughtful review. We appreciate the questions posed, as they motivated us to delve deeper and further showcase the explanatory power of our SDEs. However, we would like to clarify that **contrary to what is mentioned under "Limitations", none of our SDEs is limited to quadratic functions: The theory applies to general smooth functions.** We conducted extensive experimental validation that our SDEs correctly model the respective algorithms on a variety of architectures and datasets (see Figures 1, 4, 8, 9, and 11 and the respective experimental details in Appendix F).
>
> **Answers to Q1:**
>
> 1. As per Appendix F.5, the learning rates (lrs) used for AdamW are $10^{-2}$ for the Transformer and $10^{-5}$ for the ResNet. For RMSpropW, they are $10^{-3}$ and $10^{-4}$, respectively: We will add these details in the caption of the figures;
> 2. In our experiments, we first fine-tuned the hyperparameters to ensure the convergence of the "real" optimizers. Then, we used the same hyperparameters to simulate the SDEs.
> While we did not ablate the range of the lr over which the SDEs align well with the algorithms, our experiments use a wide range of lrs across different datasets and architectures: From $10^{-3}$ to $10^{-2}$ for SignSGD, from $10^{-4}$ to $10^{-2}$ for RMSprop(W), and from $10^{-5}$ to $10^{-2}$ for Adam(W). Our SDEs match the respective algorithms well in all such cases.
>
> **Answer to Q2:**
>
> In SGD, the error/noise on the update scales with $\sigma^2$. In SignSGD, the $Sign$ operator clips the stochastic gradient and hence it also clips its noise: This clipping/normalization implies that this error scales with $\sigma$.
>
> **Answer to Q3:**
>
> We attempted to address this question while writing the paper, but we were unable to formally explain these phenomena. To satisfy both our curiosity and that of the reviewer, we offer our conjecture in an Official Comment, providing some technical details.
>
> **Answer to Q4:**
>
> Since it was unclear which assumption was precisely meant, we have read the references and selected three noise structures: We study two below and the third one in an Official Comment.
>
> Under these assumptions, we generalized Cor. 3.3 and provided convergence in the same fashion as Lemma 3.5. Additionally, see the **Answer to Question 2 from Reviewer PBh6** for a generalized version of Cor. 3.3 where we only require the loss function to be $L$-smooth.
>
> **Assumption from [1]**
>
> [1] proposes several expressions for $\Sigma$: We took the only one in line with that prescribed by the Reviewer: As per Eq. (16) in Corollary 2, $\Sigma := \sigma^2 f(x_*) \nabla^2 f(x_*)$, where $\sigma^2$ controls the scale of the noise and $x_*$ is an optimum.
>
> Therefore, for $Y_t := \frac{\nabla^2 f(x_*)^{-\frac{1}{2}} \nabla f(X_t)}{\sqrt{2 \nabla f(x_*)} \sigma}$ and $\mathcal{S}(X_t)=\mathbb{E}[(Sign(\nabla f_{\gamma}(X_t)))(Sign(\nabla f_{\gamma}(X_t)))^{\top}]$, Cor. 3.3 becomes:
>
> $$
> d X_t = - Erf \left( Y_t \right) dt + \sqrt{\eta} \sqrt{\mathcal{S}(X_t) - Erf \left(Y_t \right) Erf \left(Y_t \right)^{\top}} d W_t.
> $$
>
> Therefore, Lemma 3.5 becomes:
> Let $f$ be $\mu$-strongly convex,  $\lambda$ be the largest eigenvalue of $\nabla^2 f(x_*)$, $S_t:=f(X_t) - f(x_*)$, and $Tr(\nabla^2 f(x)) \leq \mathcal{L}_{\tau}$. Then, during
> 1. Phase 1, the loss will reach $0$ before $t_* = 2 \sqrt{\frac{S_0}{\mu}}$ because $S_t \leq \frac{1}{4} \left( \sqrt{\mu}t - 2 \sqrt{S_0}\right)^2$;
> 2. Phase 2 with $\Delta:= \left( \frac{m}{\sqrt{2 f(x_*)}\sigma\sqrt{\lambda}} + \frac{\eta \mu m^2}{4 f(x_*) \sigma^2 \lambda } \right)$: $\mathbb{E}[S_t] \leq S_0 e^{- 2 \mu \Delta t} + \frac{\eta}{2} \frac{ \left(\mathcal{L}_{\tau} - \mu d \hat{q}^2  \right)}{2 \mu \Delta} \left(1 - e^{- 2 \mu \Delta t}\right)$;
> 3. Phase 3 with $\Delta:= \left(\sqrt{\frac{2}{\pi}} \frac{1}{\sqrt{ f(x_*)}\sigma\sqrt{\lambda}} + \frac{\eta}{\pi} \frac{\mu}{f(x_*) \sigma^2 \lambda}\right)$: $\mathbb{E}[S_t] \leq S_0 e^{- 2 \mu \Delta t} + \frac{\eta}{2} \frac{ \mathcal{L}_{\tau}}{2 \mu \Delta} \left(1 - e^{- 2 \mu \Delta t}\right)$.
>
> **Please, find an empirical validation in Figure 1.a of the attached .pdf file.**
>
> **Assumption from [3]**
>
> [3] assumes that $\Sigma$ is aligned with the FIM and proportional to the loss. Consistently with this and with the prescription of the Reviewer, we take $\Sigma := \sigma^2 f(x) \nabla^2 f(x)$, where we changed the constants to $\sigma^2$ to maintain consistency with the rest of our paper.
>
> Therefore, we have that for $Y_t := \frac{ (\nabla^2 f(X_t))^{-\frac{1}{2}}\nabla f(X_t)}{\sqrt{2  \nabla f(x)} \sigma}$ and $\mathcal{S}(X_t)=\mathbb{E}[(Sign(\nabla f_{\gamma}(X_t)))(Sign(\nabla f_{\gamma}(X_t)))^{\top}]$, Cor. 3.3 becomes:
> $$
> d X_t = - Erf \left( Y_t \right) dt + \sqrt{\eta} \sqrt{\mathcal{S}(X_t) - Erf \left(Y_t \right) Erf \left(Y_t \right)^{\top}} d W_t.
> $$
> Therefore, Lemma 3.5 becomes:
> Let $f$ be $\mu$-strongly convex, $L$-smooth,  $S_t:=f(X_t) - f(x_*)$, and $Tr(\nabla^2 f(x)) \leq \mathcal{L}_{\tau}$ Then, during
> 1. Phase 1, the loss will reach $0$ before $t_* = 2 \sqrt{\frac{S_0}{\mu}}$ because $S_t \leq \frac{1}{4} \left( \sqrt{\mu}t - 2 \sqrt{S_0}\right)^2$;
> 2. Phase 2 with $\beta := \frac{\eta}{2} \left( \mathcal{L}_{\tau} - \mu d \hat{q}^2 - \frac{m^2 \mu^2}{\sigma^2 L }\right)$ and $\alpha:=  \frac{\sqrt{2} m \mu}{\sqrt{L}\sigma}$,
> $$
> \mathbb{E}[S_t] \leq \frac{\beta^2 \left( \mathcal{W}\left( \frac{(\beta + \sqrt{S_0} \alpha)}{\beta} \exp\left(-\frac{\alpha^2 t - 2 \sqrt{S_0} \alpha}{2 \beta} - 1 \right) \right) + 1 \right)^2}{\alpha^2} \overset{t \rightarrow \infty}{\rightarrow} \frac{\beta^2}{\alpha^2},
> $$
> where $\mathcal{W}$ is the Lambert $\mathcal{W}$ function;
> 3. Phase 3, it is the same as Phase 2 but $\beta := \eta \left( \frac{\mathcal{L}_{\tau}}{2} - \frac{2 \mu^2}{\pi \sigma^2 L }\right)$ and $\alpha:= 2 \sqrt{\frac{2}{\pi}} \frac{\mu}{\sqrt{L}\sigma}$.
>
> **Please, find an empirical validation in Figure 1.b of the attached .pdf file.**

---

> ### Author Response · Authors · 2024-08-05
> **Reviewer's curiosity: Noise Structure and Conjecture on Spiking Phenomena**
>
> **_Given the length limit for the Rebuttal, we decided to include these minor results in an Official Comment._**
>
> **Continuation of Answer to Q4 - The Third Noise Structure:**
>
> [2] discusses two possible assumptions on $\Sigma$: $\|\Sigma(x)\| \leq C f(x)$ and $\|\Sigma(x)\| \leq C f(x)\left[1+|x|^2\right]$. Even though none was in line with the prescription of the Reviewer, we still thought that the one they used, i.e. $\Sigma = C f(x) I_d$ as per Section 2.4, is interesting. Therefore, we take $\Sigma := \sigma^2 f(x) I_d$, where we changed the constant to $\sigma^2$ to maintain consistency with the rest of our paper.
>
> Under this assumption, we have that for $Y_t := \frac{\nabla f(X_t)}{\sqrt{2  f(x)} \sigma}$, Corollary 3.3 becomes:
>
> \begin{align}
> d X_t = - Erf \left( Y_t \right) dt + \sqrt{\eta} \sqrt{I_d - diag(Erf \left(Y_t \right))^2} d W_t.
> \end{align}
>
> As a consequence, Lemma 3.5 becomes:
>
> Let $f$ be $\mu$-strongly convex, $S_t:=f(X_t) - f(x_*)$, and $Tr(\nabla^2 f(x)) \leq \mathcal{L}_{\tau}$. Then, during
> 1. Phase 1, the loss will reach $0$ before $t_* = 2 \sqrt{\frac{S_0}{\mu}}$ because $S_t \leq \frac{1}{4} \left( \sqrt{\mu}t - 2 \sqrt{S_0}\right)^2$;
> 2. Phase 2 with $\beta := \frac{\eta}{2} \left( \mathcal{L}_{\tau} - \mu d \hat{q}^2 - \frac{m^2 \mu^2}{\sigma^2}\right)$ and $\alpha:=  \frac{\sqrt{2} m \mu}{\sigma}$,
> \begin{equation}
> \mathbb{E}[S_t] \leq \frac{\beta^2 \left( \mathcal{W}\left( \frac{(\beta + \sqrt{S_0} \alpha)}{\beta} \exp\left(-\frac{\alpha^2 t - 2 \sqrt{S_0} \alpha}{2 \beta} - 1 \right) \right) + 1 \right)^2}{\alpha^2} \overset{t \rightarrow \infty}{\rightarrow} \frac{\beta^2}{\alpha^2},\end{equation}
> where $\mathcal{W}$ is the Lambert $\mathcal{W}$ function.
> 3. Phase 3 it is the same as Phase 2 but with $\beta := \eta \left( \frac{\mathcal{L}_{\tau}}{2} - \frac{2 \mu^2}{\pi \sigma^2}\right)$ and $\alpha:= 2 \sqrt{\frac{2}{\pi}} \frac{\mu}{\sigma}$.
>
> _Please, find an empirical validation of these bounds in Figure 1.c of the attached .pdf file._
>
> **Continuation of Answer to Q3 - Reviewer's curiosity: Conjecture on Spiking Phenomena:**
>
> This is a very interesting question that we do not address in this paper. While this is not a fundamental element for the flow and contribution of our paper, we gladly try to answer it, both for our and the Reviewer's curiosity.
>
> Although we can not answer this in the general case, we offer the following conjecture to provide an intuition of how one could explain the spiking behavior of the mentioned optimizers.
>
> Since the SDE of RMSprop is less complex and less complicated to work with, we restrict ourselves to this case: Generalizing is only a matter of technicalities.
>
> Let us remind that the SDE of RMSprop is
> $$
> d X_t = - V_t^{-\frac{1}{2}} (\nabla f(X_t) dt + \sqrt{\eta} \Sigma(X_t)^{\frac{1}{2}} d W_t)
> $$
> $$
> d V_t = \rho( (\nabla f(X_t))^2 +  diag(\Sigma(X_t)) - V_t)) dt,
> $$
> where $\beta = 1 - \eta \rho$.
>
> Intuitively, the dynamics of the parameters $X_t$ is a preconditioned version of SGD. Much differently, $V_t$ is a process that tracks the squared gradient and its noise.
>
> This implies that the expected iterates follow the dynamics:
> \begin{equation}
>     d \mathbb{E}[X_t] = -  \mathbb{E}\left[\frac{\nabla f(X_t)}{\sqrt{V_t}}\right] dt.
> \end{equation}
> Consistently with the noise structure proposed by the Reviewer and used in many papers (see [3] and references therein), let us assume that the covariance of the noise $\Sigma(x)$ scales proportionally to the loss function, e.g. $\Sigma \sim f(x)$.
>
> Spikes seem to happen when the loss is essentially $0$, meaning that $\nabla f(X_t) \sim 0$, $\Sigma \sim 0$, and $V_t \sim 0$. However, if we now draw a minibatch of data for which the gradient is not $0$, e.g. some data points that are outliers, $V_t$ might not have the time to "catch up" with this anomaly. Therefore, the numerator $\nabla f(X_t)$ is non-$0$ while the denominator $\sqrt{V_t}$ is still essentially $0$, meaning that the ratio $\frac{\nabla f(X_t)}{\sqrt{V_t}}$ spikes to infinity, drastically disturbing the dynamics of the iterates and in turn that of the loss function which might spike.

---

> > ### Comment · Reviewer_ehSv · 2024-08-09
> >
> > Many thanks to the authors for your careful explanation and detailed rebuttal. I feel that this paper is of great help in understanding signSGD/Adam. I have raised my score.

---

> > > ### Author Response · Authors · 2024-08-09
> > > **Thanks!**
> > >
> > > Dear Reviewer,
> > >
> > > Thank you for your kind words and the updated score: We truly appreciate it.
> > >
> > > Best regards,
> > >
> > > The Authors

---

### Author Rebuttal · Authors · 2024-08-05

Dear Reviewers,

We sincerely appreciate your thorough reviews, insightful comments, and interesting questions regarding our paper: Your feedback has helped enhance our work.

The considerable time and effort we devoted during this rebuttal period were rewarding, as we derived new interesting insights that complemented our paper and made it even more interesting and rich.

We are pleased to report that we have addressed your questions and comments comprehensively, exploring new settings as a result: These responses are detailed in our rebuttals to each of the Reviewers and will be incorporated into the final version of the paper.

We look forward to the upcoming author-reviewer discussion period and **kindly ask you to re-evaluate our paper, considering raising your scores and confidence in your assessments.**

Thank you for your attention.

Best regards,

The Authors

---

### Public Comment · ~Enea_Monzio_Compagnoni1 · 2024-11-20
**A One-Liner Would Have Addressed The Minor Concern.**

I would like to respectfully express my concern regarding the decision to reject this paper:

1) All three reviewers **recommended acceptance**. However, the paper was rejected by the Area Chair due to a **minor issue** regarding the regularity of the SDE coefficients, which could have been **easily addressed** with a straightforward clarification during the rebuttal period.

2) For example, this **minor issue** is easily addressed by assuming Gaussian noise (a **weaker assumption** would suffice), which is a **widely accepted** assumption in the literature, even when considering finite batch sizes.

3) Alternatively, as highlighted in the **seminal paper [1]**, any potential regularity issues could also be resolved using a **mollification approach**, thereby bypassing the need for specific noise assumptions altogether.

4) With a score of 6.33, the paper is **well above** the typical acceptance threshold. Rejecting it based on a **debatable concern** about an assumption — particularly when the theoretical results are **strongly validated** by experimental evidence — feels inconsistent with the established standards of our field. It risks undermining the **fairness and pragmatism** that should guide our evaluation process.

Best regards,

Enea Monzio Compagnoni

[1] Li, Qianxiao, Cheng Tai, and E. Weinan. "Stochastic modified equations and dynamics of stochastic gradient algorithms i: Mathematical foundations." Journal of Machine Learning Research 20.40 (2019): 1-47.

---

### Decision · Program_Chairs · 2024-09-25

**Decision:**

Reject

**Comment:**

This paper introduces novel Stochastic Differential Equations (SDEs) for adaptive optimization methods such as SignSGD, RMSProp(W), and Adam(W). The authors provide a detailed analysis of these methods, focusing on their dynamics, convergence behavior, and robustness to noise. The theoretical findings are supported by experiments on various neural network architectures, confirming the accuracy of the derived SDEs in modeling the behavior of these optimizers, which are appreciated by the reviewers.

However, I have concerns about the correctness of one of the main theoretical result in this paper, Theorem 3.2 (or its formal version, Theorem C.5 in appendix). Theorem C.5 mainly says that SignSGD (equation 18) and SDE (equation 16) are order-1 weak approximation. However, for the setting studied in this paper, i.e., batches are sampled i.i.d. from the uniform distribution of a training set of size $N$, both the drift term and diffusion matrix in equation 16 are not continuous, because $\mathrm{Pr}(\nabla f_\gamma(x)<0)$ only takes finite set of discrete values of {$0,\frac{1}{N},\frac{2}{N},\ldots,1$}. Therefore even the uniqueness and existence of solution of (16) is not obvious and the proof framework introduced by [Li et al., 17] does not apply here. This mistake is also reflected in the proof of Theorem 3.5, which applies Lemma 3.3 without satisfies its condition (drift $b$ and diffusion matrix $\Sigma$ needs to be lipschitz continuous).

With that being said, this issue could probably be fixed by only considering infinite size training dataset, or equivalently, noise distribution with continuous/smooth densities, like gaussian distribution. However, even in that case, the proof framework of [Li et al., 17] (which is further refined in [Li et al., 19]) does not apply for the SDE approximation of SignSGD directly. Therefore I think the current version is not ready for being published and the amount of modification is significant, which needs another round of review. I encourage the authors to fix the statement and the proof and resubmit to another venue.

- Li, Q., Tai, C. and Weinan, E., 2017, July. Stochastic modified equations and adaptive stochastic gradient algorithms. In International Conference on Machine Learning (pp. 2101-2110). PMLR.

- Li, Q., Tai, C. and Weinan, E., 2019. Stochastic modified equations and dynamics of stochastic gradient algorithms i: Mathematical foundations. Journal of Machine Learning Research, 20(40), pp.1-47.